# BYZANTINE-ROBUST FEDERATED LEARNING WITH LEARNABLE AGGREGATION WEIGHTS

**Javad Parsa**[*]
Uppsala University, Sweden

**Amir Hossein Daghestani**[*]
KTH, Sweden

**André M. H. Teixeira**
Uppsala University, Sweden

**Mikael Johansson**
KTH, Sweden

## ABSTRACT

Federated Learning (FL) enables clients to collaboratively train a global model without sharing their private data. However, the presence of malicious (Byzantine) clients poses significant challenges to the robustness of FL, particularly when data distributions across clients are heterogeneous. In this paper, we propose a novel Byzantine-robust FL optimization problem that incorporates adaptive weighting into the aggregation process. Unlike conventional approaches, our formulation treats aggregation weights as *learnable parameters*, jointly optimizing them alongside the global model parameters. To solve this optimization problem, we develop an alternating minimization algorithm with strong convergence guarantees under adversarial attack. We analyze the Byzantine resilience of the proposed objective. We evaluate the performance of our algorithm against state-of-the-art Byzantine-robust FL approaches across various datasets and attack scenarios. Experimental results demonstrate that our method consistently outperforms existing approaches, particularly in settings with highly heterogeneous data and a large proportion of malicious clients.

## 1    INTRODUCTION

Federated Learning (FL) is a distributed machine learning framework that enables multiple clients to collaboratively train a shared global model without transferring their private data to a central location (Li et al., 2020a; Bonawitz et al., 2019; Li et al., 2020b; McMahan et al., 2017). Instead of centralizing data, FL only exchanges model updates such as gradients or parameters, between the clients and the central server. This architecture mitigates privacy risks and supports applications with distributed data that is too costly or too sensitive to share (Li et al., 2020a). In this paper, we focus on cross-silo FL, where a moderate number of long-lived clients (e.g., hospitals or financial institutions) participate in each round.

In a typical FL workflow, a central server initializes a global model and sends it to the clients. Each client trains the model locally on its private dataset and transmits only the resulting updates back to the server (Li et al., 2020a; McMahan et al., 2017). The server aggregates these updates to improve the global model, and this cycle repeats across multiple communication rounds. Federated Averaging (FedAvg), one of the most widely adopted FL algorithms, computes a weighted average of client updates, accounting for the size of each client's dataset (McMahan et al., 2017).

A significant challenge in FL arises from the presence of malicious clients, often referred to as Byzantine clients. Previous studies (Baruch et al., 2019; Fang et al., 2020) have highlighted that the global model trained using FedAvg can be compromised when malicious clients deliberately send malicious model updates. Detecting such malicious behavior is inherently challenging due to the decentralized nature of FL, where the server has limited visibility into individual clients' local data and training processes. This challenge is further exacerbated in scenarios with heterogeneous data distributions, where each client's local dataset may differ significantly from others in terms of represented classes, feature distributions, or data volumes. These variations in data distributions make

---

[*]Equal contribution.

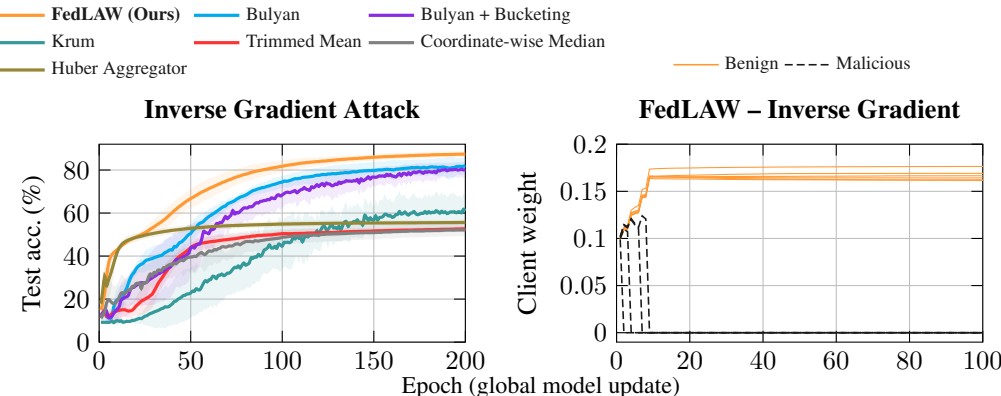

Figure 1: Test accuracy and weight evolution on MNIST under the *inverse gradient attack* (setting: $q = 0.9$, 40% malicious; see Section 5). **Left:** Average test accuracy $\pm 1$ std over 200 epochs and 5 runs, evaluated across multiple methods with 200 clients. **Right:** Aggregation weights of individual clients during the first 100 epochs (10 clients; MLP, batch size 64, 3 local epochs). Benign clients quickly converge to stable, non-trivial weights, while malicious clients are consistently suppressed.

it difficult to differentiate between benign updates influenced by data heterogeneity and corrupted updates sent by malicious clients (Cao et al., 2021; Liu et al., 2023b).

Various studies (Yin et al., 2018; Blanchard et al., 2017; Liu et al., 2023b; Guerraoui et al., 2018; Pillutla et al., 2022; Karimireddy et al., 2021) have proposed Byzantine-robust FL strategies to defend against malicious clients. Generally, robust aggregation methods can be clustered in three categories: distance-based (Blanchard et al., 2017), statistic-based (Yin et al., 2018; Farhadkhani et al., 2022; Liu et al., 2023a), and performance-based approaches (Xie et al., 2019). Krum, which is a distance-based method, filters outliers by selecting updates with the smallest cumulative Euclidean distance to their neighbors (Blanchard et al., 2017), while Median is a statistical method that replaces the mean operator with the median when aggregating local updates (Yin et al., 2018). Trimmedmean removes a fraction of the largest and smallest values for each parameter before computing the mean of the remaining values (Yin et al., 2018). Bulyan combines Krum to select consistent updates and Median to refine them (Guerraoui et al., 2018).

Defending against Byzantine attacks in heterogeneous settings presents substantial challenges (Karimireddy et al., 2020; Liu et al., 2023b). To the best of our knowledge, all existing Byzantine-robust FL methods typically follow a similar approach: after identifying and removing malicious clients, they assign uniform aggregation weights to the benign clients, akin to FedAvg (Blanchard et al., 2017; Shejwalkar & Houmansadr, 2021; Karimireddy et al., 2020; Liu et al., 2023b; Mhamdi et al., 2018). While this approach simplifies the aggregation process, it poses a significant drawback in heterogeneous data settings. Specifically, after removing the malicious clients, the distribution of data examples with specific labels may become imbalanced among the remaining benign clients. Assigning uniform weights in such a scenario fails to adapt to this imbalance, potentially degrading accuracy by not giving sufficient attention to all labels. This challenge becomes increasingly critical as the degree of data heterogeneity increases and label imbalance among benign clients becoming more pronounced. Addressing this challenge requires aggregation weights adjusted to the underlying data distribution. Figure 1 illustrates this phenomenon, showing how the aggregation weights of benign and malicious clients evolve under an inverse gradient attack and how adaptive weighting influences test accuracy.

We further demonstrate this relationship through empirical results in Section 5.

**Contributions.** In this paper, we propose a novel method for secure and robust FL that integrates adaptive weighting into the aggregation process. Unlike conventional methods, our approach treats aggregation weight selection as a part of the learning procedure, akin to global model parameters.

- First, we formulate the proposed optimization problem of jointly learning the global model parameters and the aggregation weights. Subsequently, we propose an algorithm to solve this problem using an alternating minimization approach that involves two key steps: first,

minimizing the objective with respect to aggregation weights, and second, minimizing it with respect to the model parameters.

- We provide theoretical analyses demonstrating both the Byzantine resilience and convergence properties of our method. Theorem 2 shows that the proposed objective is robust to malicious agents. In Theorem 1, we establish convergence guarantees for learning the aggregation weights step. In Theorem 3, we further prove that, under adversarial settings, the sequences generated by our algorithm, including both the aggregation weights and global model parameters, converge to a neighborhood of the optimum of the cost function.

- We evaluate the performance of our method against state-of-the-art Byzantine-robust FL approaches, considering five types of attacks and two different datasets under varying levels of heterogeneity.

## 2 NOTATION AND MODEL

### 2.1 NOTATION

Vectors and matrices are denoted by bold lowercase and uppercase letters, respectively. The support of a vector $\mathbf{w}$, denoted by $\mathrm{supp}(\mathbf{w})$, is the set of indices corresponding to the non-zero elements in $\mathbf{w}$. The symbols $\|\mathbf{w}\|_0$ and $\|\mathbf{w}\|_2$ denote the $\ell_0$ pseudo-norm (i.e., the number of non-zero elements in $\mathbf{w}$) and the $\ell_2$ norm of $\mathbf{w}$, respectively. The inner product of two vectors $\mathbf{x}$ and $\mathbf{y}$ is represented by $\langle \mathbf{x}, \mathbf{y} \rangle$. Additionally, $\nabla_w f$ denotes the gradient of the function $f$ with respect to $\mathbf{w}$. The vector $\mathbf{x}_\Lambda$ is a subset of the vector $\mathbf{x}$, which contains only the elements indexed by the entries in the vector $\Lambda$. The $i$-th element of the vector $\mathbf{x}$ is denoted by $x_i$. Finally, $\mathbf{v}_{k,i}$ denotes the vector associated with client $i$ at synchronous round $k$. Throughout the paper, $\mathbb{E}\{\cdot\}$ denotes expectation over the full-batch data distribution, while $\mathbb{E}_{\mathrm{batch}}\{\cdot\}$ denotes expectation with respect to mini-batch random sampling.

### 2.2 MODEL

We consider a federated learning system with a parameter server and $n$ clients, up to $b_f$ of which may be Byzantine (acting arbitrarily). In each synchronous round $k$, the server broadcasts the global model parameters $\boldsymbol{\theta}_k \in \mathbb{R}^d$ to all clients. Each honest client $i$ computes a mini-batch gradient $\tilde{\mathbf{v}}_{k,i} := \frac{1}{B} \sum_{z \in \xi_{k,i}} \nabla_\theta f_i(\boldsymbol{\theta}_k; z)$ where $\xi_{k,i} \subset D_i$ is a random sample of size $B$ from its local dataset $D_i$. This mini-batch gradient is an unbiased estimate of the client's full batch population gradient, $\mathbf{v}_{k,i}$, satisfying: $\mathbb{E}_{\xi_{k,i}}\{\tilde{\mathbf{v}}_{k,i}\} = \mathbf{v}_{k,i} = \nabla_\theta f_i(\boldsymbol{\theta}_k)$ with $\mathbb{E}_{D_i}\{\mathbf{v}_{k,i}\} = \mathbf{g}_{k,i}$. Each Byzantine client $j$ may submit an arbitrary gradient vector $\mathbf{b}_{k,j}$ and loss $\tilde{f}_{k,i}$. Attackers have full knowledge of the system and can collaborate (Lynch, 1996). The server receives $n$ gradient vectors, applies a robust aggregation rule to compute a single aggregated gradient $F_k$, and updates the model via the rule $\boldsymbol{\theta}_{k+1} = \boldsymbol{\theta}_k - \alpha F_k$.

## 3 BYZANTINE-ROBUST FEDERATED LEARNING WITH LEARNABLE WEIGHTS

### 3.1 PROBLEM FORMULATION

Traditional FL aims of finding a set of global model parameters $\boldsymbol{\theta} \in \mathbb{R}^d$ minimizing the training loss $f(\boldsymbol{\theta}) = \sum_{i=1}^{n} w_i f_i(\boldsymbol{\theta})$, where $n$ is the number of clients and $f_i : \mathbb{R}^d \mapsto \mathbb{R}$ represents the loss function of the $i$-th client. The aggregation weights $w_i$ are fixed and satisfy $w_i \geq 0$ and $\sum_{i=1}^{n} w_i = 1$. Byzantine resilience is typically introduced by adding functionality for detecting malicious clients and removing them from the learning process, effectively setting the corresponding $w_i$'s to zero.

The key idea in our approach is to transform the binary detection and removal of suspicious clients into a continuous weight optimization process, effectively embedding the Byzantine defense into the learning objective itself. We do so by treating the weights $\mathbf{w} = [w_1, \ldots, w_n]$ as decision variables and jointly optimize them with $\boldsymbol{\theta}$:

$$\min_{\boldsymbol{\theta} \in \mathbb{R}^d, \mathbf{w} \in \Delta_{t,\ell_0}^+} \sum_{i=1}^{n} w_i f_i(\boldsymbol{\theta}) \tag{1}$$

where

$$\Delta_{t,\ell_0}^+ = \{\mathbf{w} \in \mathbb{R}^n \mid \sum_{i=1}^n w_i = 1, w_i \geq 0, w_i \leq t, \|\mathbf{w}\|_0 \leq s\}. \tag{2}$$

Here $\Delta_{t,\ell_0}^+$ denotes a sparse unit-capped simplex, and the $\ell_0$ pseudo norm of $\mathbf{w}$ is utilized to achieve Byzantine robustness. Notably, if we set $t = 1/(n - b_f)$ and $s = n - b_f$, the only feasible weight vectors in $\Delta_{t,\ell_0}^+$ are those where $b_f$ clients are excluded and all others are weighted equally in the objective (see Proposition 2 in § A of the Supplementary).

## 3.2 PROPOSED ALGORITHM

Several techniques for solving (1) already exist, including the BSUM algorithm by Razaviyayn et al. (2013) and the prox-linear approach introduced in (Drusvyatskiy et al., 2019). These algorithms rely on alternating between updating the weights for fixed model parameters, and revising the model parameters for fixed weights. In our experience (see § B in the Supplementary), such updates tend to be too aggressive, and miss important couplings between the two variable blocks that are helpful for detecting malicious clients. To address these challenges, we propose a new algorithm based on rewriting (1) as a nested optimization problem (Dempe, 2002):

$$\min_{\mathbf{w} \in \Delta_{t,\ell_0}^+} \min_{\boldsymbol{\theta} \in \mathbb{R}^d} \sum_{i=1}^n w_i f_i(\boldsymbol{\theta}) \tag{3}$$

To solve the inner optimization problem, we use a quadratic approximation $\hat{f}_i(\boldsymbol{\theta})$ of $f_i(\boldsymbol{\theta})$

$$\hat{f}_i(\boldsymbol{\theta}; \boldsymbol{\theta}_k) = f_i(\boldsymbol{\theta}_k) + \langle \nabla_\theta f_i(\boldsymbol{\theta}_k), \boldsymbol{\theta} - \boldsymbol{\theta}_k \rangle + \frac{1}{2\alpha} \|\boldsymbol{\theta} - \boldsymbol{\theta}_k\|_2^2. \tag{4}$$

Substituting this quadratic approximation into the inner optimization problem in (3) leads to:

$$\boldsymbol{\theta}_{k+1}(\mathbf{w}) = \operatorname*{argmin}_{\boldsymbol{\theta} \in \mathbb{R}^d} \sum_{i=1}^n w_i \hat{f}_i(\boldsymbol{\theta}; \boldsymbol{\theta}_k) = \boldsymbol{\theta}_k - \alpha \sum_{i=1}^n w_i \nabla f_i(\boldsymbol{\theta}_k) = \boldsymbol{\theta}_k - \alpha \mathbf{G}_k \mathbf{w}, \tag{5}$$

where $\alpha$ is a step size and $\mathbf{G}_k = [\nabla_\theta f_1(\boldsymbol{\theta}_k), \cdots, \nabla_\theta f_n(\boldsymbol{\theta}_k)]$. Note that the model update depends on the aggregation weights. This dependence is accounted for in the outer optimization, which becomes

$$\mathbf{w}_{k+1} = \operatorname*{argmin}_{\mathbf{w} \in \Delta_{t,\ell_0}^+} \sum_{i=1}^n w_i f_i(\boldsymbol{\theta}_{k+1}(\mathbf{w})) = \operatorname*{argmin}_{\mathbf{w} \in \Delta_{t,\ell_0}^+} \sum_{i=1}^n w_i f_i(\boldsymbol{\theta}_k - \alpha \mathbf{G}_k \mathbf{w}), \tag{6}$$

The optimization in (6) minimizes the global objective by adjusting the weights $\mathbf{w} \in \Delta_{t,\ell_0}^+$ considering the effect that they have on the parameter update $\boldsymbol{\theta}_{k+1}(\mathbf{w}) = \boldsymbol{\theta}_k - \alpha \mathbf{G}_k \mathbf{w}$. In particular, the weighted combination of client gradients in $\mathbf{G}_k \mathbf{w}$ shapes the parameter update, coupling the choice of weights to the losses $f_i(\boldsymbol{\theta}_{k+1}(\mathbf{w}))$. This formulation prioritizes clients whose gradients, represented by the columns of $\mathbf{G}_k$, align with the descent direction of $f_i(\boldsymbol{\theta}_k - \alpha \mathbf{G}_k \mathbf{w})$, as benign client gradients typically form a coherent cluster in the parameter space. In contrast, Byzantine clients tend to submit updates that deviate from this direction or behave inconsistently due to adversarial perturbations, making them outliers. The sparsity constraint $\Delta_{t,\ell_0}^+$ ensures that up to $n - s$ misaligned clients, including Byzantines, are excluded, enhancing robustness of the learning process (see § B in the Supplementary).

While (5) is a simple gradient descent step, the optimization in (6) is more challenging since both its objective and constraint set are non-convex. Nevertheless, it can be approached by first approximating

$$\Phi_k(\mathbf{w}) = \sum_{i=1}^n w_i f_i(\boldsymbol{\theta} - \alpha \mathbf{G}_k \mathbf{w}) \tag{7}$$

by the following quadratic function

$$\hat{\Phi}_k(\mathbf{w}) = \Phi_k(\mathbf{w}_k) + \langle \nabla_\mathbf{w} \Phi_k(\mathbf{w}_k), \mathbf{w} - \mathbf{w}_k \rangle + \frac{1}{2\beta} \|\mathbf{w} - \mathbf{w}_k\|_2^2$$

for some positive step-size $\beta$. We then replace (6) by

$$\mathbf{w}_{k+1} = \operatorname*{argmin}_{\mathbf{w}} \hat{\Phi}_k(\mathbf{w}) + \delta_{\Delta_{t,\ell_0}^+}(\mathbf{w}) \tag{8}$$

where $\delta_{\Delta_{t,\ell_0}^+}(\mathbf{w})$ is the indicator function of the set $\Delta_{t,\ell_0}^+$. By completing the square and dropping constant terms, the above update is equivalent to

$$\mathbf{w}_{k+1} = \operatorname*{argmin}_{\mathbf{w}} \frac{1}{2}\|\mathbf{w} - \mathbf{h}_k\|_2^2 + \delta_{\Delta_{t,\ell_0}^+}(\mathbf{w}) = \operatorname{prox}_{\Delta_{t,\ell_0}^+}(\mathbf{h}_k), \tag{9}$$

where $\mathbf{h}_k = \mathbf{w}_k - \beta\nabla_{\mathbf{w}}\Phi_k(\mathbf{w}_k)$ and the final equality follows from the definition of the proximal mapping in (16) (see § A in the Supplementary). Thus, $\mathbf{w}_{k+1}$ is the projection of $\mathbf{h}_k$ onto the sparse unit capped simplex $\Delta_{t,\ell_0}^+$. Although this set is non-convex, the next result shows how the projection can be performed efficiently.

**Theorem 1.** *Denote $P_{L_s}(\mathbf{h}_k)$ as the operator selecting the $s$ largest elements of the vector $\mathbf{h}_k$, and let $\mathcal{P}_{\Delta_t^+}$ be the projection operator onto the unit-capped simplex $\Delta_t^+ = \{\mathbf{w} \in \mathbb{R}^n \mid \sum_{i=1}^n w_i = 1, w_i \geq 0, w_i \leq t\}$, which has an efficient solution provided in Algorithm 3 (see Section D in the Supplementary).*

*The problem (9) is exactly solved by the three-step projection method below:*

1. ***Sparsity enforcement*** $\mathbf{h}_\lambda = P_{L_s}(\mathbf{h}_k)$

2. ***Support selection:*** $\mathcal{S}^* = supp(\mathbf{h}_\lambda)$ *(10)*

3. ***Unit capped simplex projection:*** $\mathbf{w}_{k+1\mathcal{S}^*} = \mathcal{P}_{\Delta_t^+}(\mathbf{h}_{\lambda\mathcal{S}^*}), \quad \mathbf{w}_{k+1(\mathcal{S}^*)^{\complement}} = 0.$

*Proof.* See Section E in the Supplementary. $\quad\square$

After having updated the weights $\mathbf{w}_{k+1}$ based on (9), the parameter vector $\boldsymbol{\theta}_k$ can be updated by substituting $\mathbf{w} = \mathbf{w}_{k+1}$ into (5), yielding:

$$\boldsymbol{\theta}_{k+1} = \boldsymbol{\theta}_k - \alpha\mathbf{G}_k\mathbf{w}_{k+1}. \tag{11}$$

The pseudo-code of the final algorithm is summarized in Algorithm 2 (see Supplementary).

To translate our algorithm procedure in the FL framework, at epoch $k + 1$, the server broadcasts $\boldsymbol{\theta}_k$ to clients, who return $\nabla_\theta f_i(\boldsymbol{\theta}_k)$. The server constructs $\mathbf{G}_k$, computes $\tilde{\boldsymbol{\theta}}_{k+1} = \boldsymbol{\theta}_k - \alpha\mathbf{G}_k\mathbf{w}_k$, and sends $\tilde{\boldsymbol{\theta}}_{k+1}$ to clients. Clients respond with $\mathbf{f}_{k+1} = [f_1(\tilde{\boldsymbol{\theta}}_{k+1}), \dots, f_n(\tilde{\boldsymbol{\theta}}_{k+1})]^\top$ and $\tilde{\mathbf{G}}_{k+1} = [\nabla_\theta f_1(\tilde{\boldsymbol{\theta}}_{k+1}), \dots, \nabla_\theta f_n(\tilde{\boldsymbol{\theta}}_{k+1})]$. The server then updates $\mathbf{w}_{k+1}$ via (10), with $\mathbf{h}_k = \mathbf{w}_k + \alpha\beta\mathbf{G}_k^\top\tilde{\mathbf{G}}_{k+1}\mathbf{w}_k - \beta\mathbf{f}_{k+1}$, and computes $\boldsymbol{\theta}_{k+1}$ via (11). This requires two communication rounds: one for $\boldsymbol{\theta}_k$ and gradients, another for $\tilde{\boldsymbol{\theta}}_{k+1}$ and client responses.

**Remark 1.** *While our method requires two communication rounds per training epoch, this does not necessarily imply a doubling of total communication rounds compared to standard FL. We highlight a key factor that mitigate this cost. Introducing $\mathbf{w}$ as a decision variable offers additional flexibility in minimizing the loss (see (6)), which accelerates convergence of the global model parameters (see Figure 8 in the Supplementary). To fairly compare methods, we evaluate performance based on total communication rounds. Figure 8 shows that FedLAW achieves higher test accuracy within the same communication round. As a result, the total number of rounds required to reach a target accuracy by our method may be only marginally higher, or even lower than that of competing methods.*

We conclude this section by discussing the server-side overhead relative to standard FL. The only extra step in our method is the aggregation weight update (10), while the model update remains identical to that in standard FL. The following proposition quantifies the complexity of the extra step.

**Proposition 1.** *The memory complexity of server-side aggregation weight update based on (10) is $\mathcal{O}(dn)$ per round, and the computational complexity of the projection (10) in the server is $\mathcal{O}(n\min(s, \log n) + s^2)$, where $d$ is the number of model parameters, $n$ is the number of clients, and $s$ is the sparsity level.*

*Proof.* See Section I.5 in the Supplementary. $\quad\square$

## 4 THEORETICAL ANALYSES

### 4.1 BYZANTINE-RESILIENT ANALYSIS

The main method proposed in (6) is designed for Byzantine-resilient federated learning through optimized aggregation weights. The following theorem establishes its resilience property, as formalized in Definition 5 (Section A in the Supplementary).

The first step in our resilience analysis is to make the objective in (6) analytically tractable. We use Taylor's theorem with an exact remainder to reformulate it, revealing that it is governed by the quadratic form $\mathbf{w}^T \mathbf{G}_k^T \mathbf{G}_k \mathbf{w}$. By denoting $\mathbf{v}_{k,i} = \nabla_\theta f_i(\boldsymbol{\theta}_k)$, we can show this term is equivalent to an expression involving pairwise gradient distances:

$$\mathbf{w}^T \mathbf{G}_k^T \mathbf{G}_k \mathbf{w} = -\sum_{i=1}^n w_i \|\mathbf{v}_{k,i}\|_2^2 + \sum_{i=1}^n \sum_{j=1}^n w_i w_j \|\mathbf{v}_{k,i} - \mathbf{v}_{k,j}\|_2^2.$$

From here, the proof exploits the terms $\|\mathbf{v}_{k,i} - \mathbf{v}_{k,j}\|_2^2$ with a novel, especially tailored approach to establish Byzantine resilience. Although gradient differences are exploited in other Byzantine-resilient methods, here they appear naturally from the structure of (6) using our novel derivations. Note that the nested structure of (3), leading to (6), is an essential building block.

It is important to highlight that, unlike Definition 5, we consider a more general setting where population losses and gradients are non-iid. To improve practicality, our analysis also focuses on the case where the aggregator relies on mini-batch gradients rather than full-batch gradients, which is a more realistic scenario.

We now state the formal assumptions for our theoretical analysis.

**Assumption 1** (Formal Setup for Theoretical Analysis). *We analyze the setting where each honest client $i \in \mathcal{H}$ provides a mini-batch gradient $\tilde{\mathbf{v}}_{k,i}$ of size $B$. Let $\mathbf{v}_{k,i} = \nabla_\theta f_i(\boldsymbol{\theta}_k)$ be the full batch population gradient. We assume the following:*

*(A) Client loss functions*

    *(A1) (Smoothness) Each client loss function $f_i(\boldsymbol{\theta})$ is $L_i$-smooth, with $L_{\max} = \max_{1 \le i \le n} L_i$.*

    *(A2) The weight objective function $\Phi_k(\mathbf{w})$ in (7) with respect to $\mathbf{w}$ is $L_w$-smooth.*

*(B) Step-size Schedule*

    *(B1) Algorithm 2 employs a hybrid schedule where the step-size $\alpha$ is a fixed constant $0 < \alpha < 1/L_{\max}$, and the weight step-size $\{\beta_k\}$ is an adaptive schedule satisfying the standard Robbins-Monro conditions ($\beta_k > 0, \beta_k \to 0, \sum \beta_k = \infty, \sum \beta_k^2 < \infty$) and $\beta_k < 1/L_w$.*

*(C) Stochastic Gradient Model*

    *(C1) The deviation of any single-sample gradient from its full batch population is bounded by $\|\nabla f_i(\boldsymbol{\theta}_k; z) - \mathbf{v}_{k,i}\| \le R_k$ on one single data point $z$.*

*(D) Population Heterogeneity*

    *(D1) We assume the population gradients and losses at round $k$, $\{f_i(\boldsymbol{\theta}_k), \mathbf{v}_{k,i}\}_{i \in \mathcal{H}}$ is $\mathbb{E}\{\mathbf{v}_{k,i}\} = \mathbf{g}_{k,i}$ with $\mathbf{g} = \frac{1}{|\mathcal{H}|} \sum_{i \in \mathcal{H}} \mathbf{g}_{k,i}$, $m_{k,i} = \mathbb{E}\{f_i(\boldsymbol{\theta}_k)\}$ with $m_{avg} = \frac{1}{|\mathcal{H}|} \sum_{i \in \mathcal{H}} m_{k,i}$, and*

        • *Directional Heterogeneity: $\frac{1}{|\mathcal{H}|} \sum_{i \in \mathcal{H}} \|\mathbf{g}_{k,i} - \mathbf{g}\|^2 \le H_k^2$.*

        • *Magnitude Heterogeneity: $\left| \|\mathbf{g}_{k,i}\|^2 - \|\mathbf{g}\|^2 \right| \le K_k^2$.*

        • *Inter-Client Variance: $\mathbb{E}\{\|\mathbf{v}_{k,i} - \mathbf{g}_{k,i}\|_2^2\} \le d\sigma_k^2$.*

        • *Loss Heterogeneity: $\frac{1}{|\mathcal{H}|} \sum_{i \in \mathcal{H}} |m_{k,i} - m_{avg}| \le \varepsilon_k$.*

*(E) Byzantine Clients and Aggregator*

*(E1) Attackers submit arbitrary gradients $\mathbf{b}_{k,i}$ and non-negative losses $\tilde{f}_{k,i} \geq 0$, subject to the norm constraint $\|\mathbf{b}_{k,i}\| \leq \max_{j \in \mathcal{H}} \|\tilde{\mathbf{v}}_{k,j}\|_2$.*

**Theorem 2** (High-Probability Byzantine Resilience). *Under Assumption 1 (excluding A2 and B1) and let assume $b_f$ min-batch gradient updates $\tilde{\mathbf{v}}_{k,i}$ are replaced with their Byzantine counterparts $\mathbf{b}_{k,j}$ at any round $k$ with $2b_f + 2 \leq n$. Let the aggregator $\tilde{F}$ be computed as follows*

$$\tilde{F} = \sum_{i \in \mathcal{H}} w_{k,i} \tilde{\mathbf{v}}_{k,i} + \sum_{j \in \mathcal{H}^{\complement}} w_{k,j} \mathbf{b}_{k,j} \tag{12}$$

*where $\mathbf{w}_k$ represents the optimal weight vector obtained from (6), for a step-size $0 < \alpha \leq \alpha_{\max}$ in which*

$$\alpha_{\max} = \min \left\{ \frac{1}{L_{\max}}, \frac{C_{het}}{2L_{\max}\|\mathbf{g}\|^2} \right\}, \quad \varepsilon_S = \sqrt{\frac{2R_k^2 \log(2d/\delta)}{B}}, \tag{13}$$

$$C_{het} := \frac{4K_k^2 b_f}{n - b_f} + 2H_k^2 + \frac{2d\sigma_k^2(2n - b_f - 2)}{n - b_f} + \frac{2\varepsilon_S^2 n}{n - b_f}$$

*Then, with probability at least $1 - \delta$, the aggregator $\tilde{F}$ is Byzantine-resilient. Specifically, its expected bias with respect to the true global mean $\mathbf{g}$ is bounded by:*

$$\|\mathbb{E}\{\tilde{F}\} - \mathbf{g}\| \leq \eta_k, \tag{14}$$

*where the error bound $\eta_k$ is explicitly decomposed into the distinct sources of error:*

$$\eta_k = \sqrt{2b_f \left( \frac{\varepsilon_k}{\alpha} + C_{het} + L_{\max}\alpha \left( \frac{H_k^2}{2} + \frac{K_k^2}{2} + d\sigma_k^2 + \varepsilon_S^2 \right) \right)} + \frac{2b_f}{\sqrt{n - b_f}} H_k + \varepsilon_S \tag{15}$$

*Furthermore, if the signal-to-noise condition $\eta_k < \|\mathbf{g}\|$ holds, then the angle $\zeta$ between the expected aggregated gradient $\mathbb{E}\{\tilde{F}\}$ and the true global gradient $\mathbf{g}$ is bounded by $\sin \zeta \leq \frac{\eta_k}{\|\mathbf{g}\|}$.*

*Proof.* See Section F in the Supplementary. $\qquad\square$

Theorem 2 provides a comprehensive resilience guarantee for the proposed method. The key insight is the error decomposition in Eq. (15), which shows that the total error $\eta_k^2$ is a sum of four distinct and well-characterized sources: loss heterogeneity, gradient heterogeneity, inter-client variance, and mini-batch sampling noise. The bound explicitly shows that the sampling noise is controllable, as it vanishes when the mini-batch size $B$ increases.

The assumptions underpinning this result are standard and practically justifiable. The condition $\|\mathbf{b}_{k,i}\| \leq \max_{j \in \mathcal{H}} \|\tilde{\mathbf{v}}_{k,j}\|_2$ is enforced in practice by a standard gradient clipping mechanism on the server side: each incoming update is projected onto an $\ell_2$-ball of radius $C$. Honest updates remain untouched when $C$ exceeds their typical norm, whereas malicious updates cannot exceed the threshold. This adds no extra communication and is already commonplace in FedAvg, DP-Fed, and similar protocols. Similarly, assuming bounded heterogeneity $(H_k^2, K_k^2)$ is a standard prerequisite for any non-IID analysis. In addition, in practice $\varepsilon_k$ is typically very small even under a high heterogeneity, thereby mitigating the effect of the $1/\alpha$ scaling in (15). To support this empirically, we explicitly measured $\varepsilon_k$ under a large malicious fraction and extreme heterogeneity in Subsection I.6, which confirms this behavior. Additionally, Byzantine resilience under the scenario where both the losses and gradients are replaced with Byzantine versions is established separately in Theorem 8 (see Section G in the Supplementary).

**Remark 2.** *Theorem 2 is stated in a probabilistic form because, in practice, FedLAW operates on* mini-batch *gradients rather than full-batch expectations. Mini-batch sampling introduces an additional source of randomness beyond the data distribution, and in an adversarial setting the attacker interacts with* single realizations *of these samples, not their expectation. High-probability bounds therefore provide a more realistic Byzantine-resilience guarantee than purely deterministic, full-batch analyses.*

## 4.2 CONVERGENCE ANALYSIS

A key difficulty in Algorithm 2 is the arbitrary behavior of the adversary, which complicates the convergence analysis of our alternating minimization scheme to optimize $\boldsymbol{\theta}$ and $\mathbf{w}$ jointly. We tackle this by viewing the method through the lens of a hybrid step-size schedule. This perspective yields a strong stability guarantee: even under attack, the adaptive weights not only stabilize, but their limit is a critical point of the exact (non-approximated) objective in (6). Building on this result, the following theorem establishes convergence to a neighborhood of the optimum of problem (1) in both non-convex and strongly convex regimes under adversarial conditions.

**Theorem 3.** *Let the sequences $\{(\boldsymbol{\theta}_k, \mathbf{w}_k)\}_{k=1}^{\infty}$ be generated by Algorithm 2. Let the aggregator be $F_k(\mathbf{w}) = \mathbf{G}_k \mathbf{w}$ with bounded variance $\mathrm{Var}(F_k(\mathbf{w}_k)) \leq \sigma_{F,k}^2$, where $\mathbf{G}_k$ contain mini-batch honest gradients as well as arbitrary Byzantine gradients. Assume Assumption 1 holds and suppose:*

- *$\|\mathbb{E}\{F_k(\mathbf{w}_k)\} - \mathbf{g}_k\| \leq \zeta_k$ where $\mathbf{g}_k = \sum_{i=1}^{n} w_{k,i} \mathbb{E}\{\mathbf{v}_{k,i}\}$.*

- *As $k \to \infty$, we have $\zeta_k \to \zeta_\infty$, $\sigma_{F,k}^2 \to \sigma_{F,\infty}^2$, and $\sigma_k^2 \to \sigma_\infty^2$.*

*Then, the aggregation weights satisfy $\mathbf{w}_{k+1} \to \mathbf{w}_k$ as $k \to \infty$ with limits point $\mathbf{w}^\star$, which is a critical point of (6) for $k \to \infty$. Moreover, the following convergence guarantees hold:*

1. ***(L-smooth, Non-Convex Case)** The sequence generated by our algorithm converges to a neighborhood of a stationary point of the main loss function (1). In particular, with probability at least $1 - \delta$, the time-averaged squared gradient norm is bounded as:*

$$\limsup_{T \to \infty} \frac{1}{T} \sum_{k=0}^{T-1} \|\mathbf{g}_k\|^2 \leq \lim_{T \to \infty} \frac{\sum_{k=1}^{T} C_{2,k}}{T C_1} = \mathcal{O}\left(\zeta_\infty^2 + \sigma_{F,\infty}^2\right)$$

*where $C_1 = \frac{1}{4}(1 - \alpha L_{\max}) > 0$, $\varepsilon_S = \sqrt{\frac{2R_k^2 \log(2d/\delta)}{B}}$, and*

$$C_{2,k} = 2\zeta_{k+1}\sqrt{K_k^2 + d\sigma_k^2 + \varepsilon_S^2} + \zeta_k^2 + \sigma_{F,k}^2 + \sigma_k \sqrt{d}\sqrt{\zeta_k^2 + \sigma_{F,k}^2} + \frac{(2\zeta_{k+1} + \zeta_k + 4C_1\zeta_k)^2}{4C_1}.$$

2. ***(Strongly Convex and L-smooth Case)** If the client loss functions $f_i$ are $\mu$-strongly convex, then the algorithm converges to a neighborhood of the global optimum. Moreover, with probability at least $1 - \delta$, we have*

$$\limsup_{k \to \infty} \mathbb{E}\{Q(\boldsymbol{\theta}_k, \mathbf{w}_k) - Q^\star\} \leq d\sigma_\infty^2 + \frac{C_{2,\infty}}{2\mu C_1} = \mathcal{O}\left(\zeta_\infty^2 + \sigma_{F,\infty}^2 + \sigma_\infty^2\right).$$

*in which $Q(\boldsymbol{\theta}, \mathbf{w})$ denotes loss function in (1). Also, $Q^\star$ is its global minimum when $\mathbf{w} = \mathbf{w}^\star$, evaluated under honest gradients.*

*In addition, the bound $\zeta_\infty \leq \eta_\infty$ holds, where $\eta_\infty$ is the upper bound from (14) in Theorem 2, representing the Byzantine resilience guarantee of the update rule (6) as $k \to \infty$.*

*Proof.* See Section H in the Supplementary. $\qquad\square$

Theorem 3 establishes that our algorithm converges to a bounded neighborhood of the optimum, with the error radius scaling directly with the adversary's influence. The final error bound is determined by the asymptotic bias and variance of the aggregator ($\zeta_\infty, \sigma_{F,\infty}$), highlighting a powerful property: as the aggregator improves (e.g., $\zeta_k \to 0$), the algorithm converges to increasingly precise solutions.

The key insight of our work lies in the formal link between these two results. Theorem 2 provides a per-iteration resilience guarantee ($\eta_k$), while Theorem 3 strengthens this by showing that the asymptotic bias is bounded by the same guarantee, $\zeta_\infty \leq \eta_\infty$. Together, they establish that the long-term bias of the converged system is controlled by the same mechanism that ensures step-wise robustness. This unifies the static resilience of the aggregator with the dynamic convergence of the algorithm, demonstrating that robustness at the aggregation level directly translates into stability and convergence at the system level.

This theoretical foundation is strongly supported by our empirical results. Our experiments (e.g., Figs. 1 and 4) show the practical outcome of this theory: the algorithm effectively neutralizes attackers by learning to assign them zero weight, which is empirical evidence that the bias $\zeta_k$ converges to zero. Furthermore, the theoretical insights in Section B explain the mechanism by which our objective function enables the detection of malicious clients, providing a clear rationale for the algorithm's observed success.

Furthermore, in Appendix H.3, we show that if each $f_i$ is $L_i$-smooth and has bounded gradients with upper bound $C$, then $\nabla_w \Phi_k(\mathbf{w})$ is $L_w$-Lipschitz continuous with

$$L_w \leq \alpha C^2 \left( n^{3/2} + n + \alpha n L_{\max} + \frac{\alpha n^2 L_{\max} \varrho}{2} \right),$$

where $\varrho = \sqrt{kt^2 + r^2}$ with $k = \lfloor 1/t \rfloor$ and $r = 1 - kt \in [0, t)$.

## 5 NUMERICAL STUDY

### 5.1 EXPERIMENTAL SETUP

**Datasets and Models:** We conduct experiments on the MNIST (LeCun & Cortes, 2010) and CIFAR10 (Krizhevsky et al., 2009) datasets. We train a 3-layer fully connected network on MNIST, and a 4-layer convolutional neural network with group normalization on CIFAR10.

**Data Distribution and Client Configuration:** We simulate a federated setting with 200 clients and distribute the data in a non-IID fashion using the method from Cao et al. (2021). To control the degree of data heterogeneity, we introduce a concentration parameter $q$: each training example with label $l$ is assigned to the $l$-th group with probability $q$, and to the remaining $L - 1$ groups with probability $\frac{1-q}{L-1}$, where $L = 10$ is the total number of labels in our experiments. Within each group, data is uniformly distributed to clients. We consider $q \in \{0.6, 0.9\}$ to simulate moderate and high levels of data heterogeneity, respectively. The proportion of malicious clients varies across $\{0.1, 0.2, 0.3, 0.4\}$.

**Attack Types and Baselines:** We evaluate robustness under five adversarial attacks: label-flipping, inverse-gradient, backdoor, a combined (double) attack, and the LIE (Little Is Enough) attack. For comparison, we consider several baseline defenses: Krum, Trimmed Mean, Bulyan, Coordinate-wise Median (CwMed), CCLIP (Centered CLIPping (Karimireddy et al., 2020)), RFA (Robust Federated Averaging (Pillutla et al., 2022)), Huber aggregator (Zhao et al., 2024) and Bucketing combined with Bulyan, RFA, or CCLIP. We also include FedAvg, the standard aggregation without defense.

**Evaluation Protocol:** Each dataset is split into 80% training, 10% validation, and 10% testing. We report average test accuracy and malicious client detection accuracy over five independent runs. Standard deviations are provided in tables (not included here) for completeness. Additional experimental details, including computational complexity, hyperparameter settings, and sensitivity analysis, are provided in Section I of the Supplementary.

### 5.2 EXPERIMENTAL RESULTS

**Results on MNIST (see Figure 2.a and Table 3):** FedLAW delivers consistently strong results, typically surpassing robust baselines. The advantage is especially pronounced under severe contamination and heterogeneity; for example, under the inverse-gradient attack with 40% malicious clients, FedLAW attains a test accuracy 3.6% higher than the next-best defense. While several methods remain competitive at low attack rates, defenses such as RFA, RFA-bucketing, CClip, and CClip-bucketing deteriorate markedly as the attacker fraction grows, with some even diverging. For example, under the double attack, the accuracy of RFA and RFA-bucketing decreases by more than 31% with rising heterogeneity, and both CClip and CClip-bucketing diverge, whereas FedLAW maintains robustness.

A defining strength of FedLAW is its stability. Whereas competing defenses degrade sharply, Fed-LAW's accuracy remains notably consistent across attacker fractions, exhibiting graceful degradation. This robustness stems from a two-pronged design: it (i) identifies and removes malicious clients and (ii) adaptively reweights the remaining honest updates. Unlike traditional approaches that revert to uniform weights after filtering suspected clients, FedLAW continuously learns optimal weights, enhancing both robustness and representational fairness.

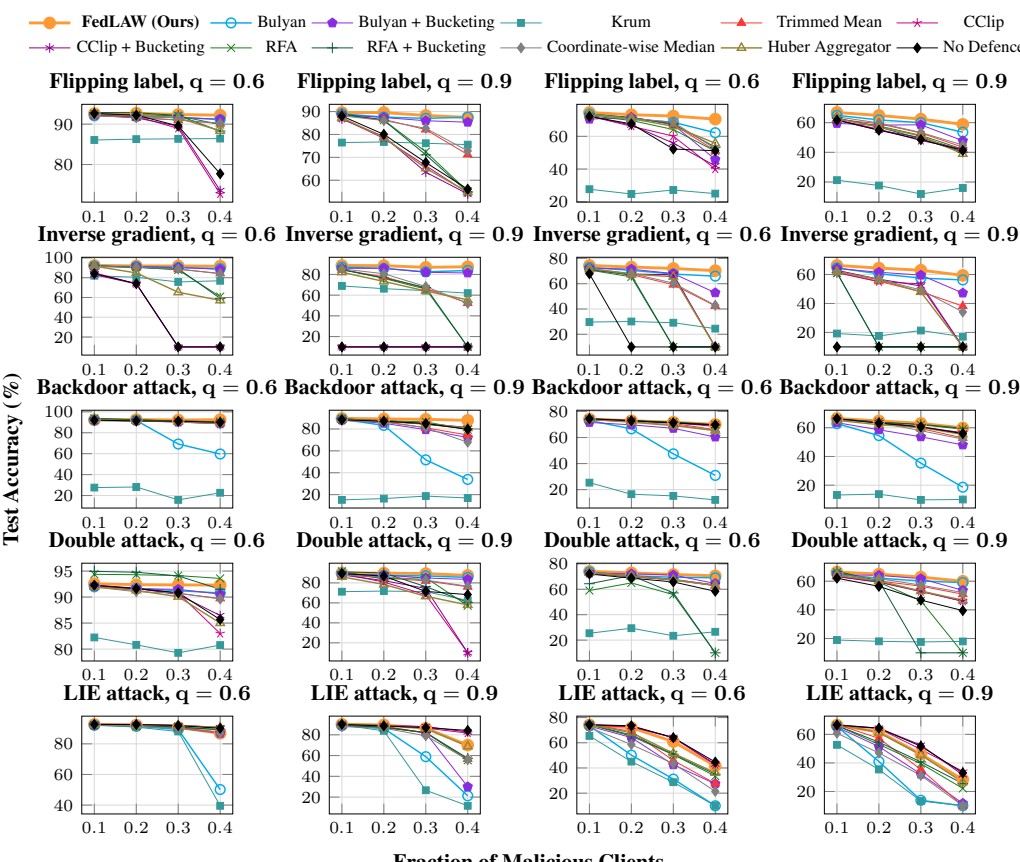

Figure 2: Defending against attacks on **MNIST** (left two columns) and **CIFAR10** (right two columns)

**Results on CIFAR10 (see Figure 2.b and Table 4):** The more complex CIFAR10 dataset presents additional challenges due to its higher dimensionality and visual diversity. Nevertheless, FedLAW consistently shows strong resilience across all attack types and configurations. Under label flipping with $q = 0.6$ and $40\%$ adversaries, FedLAW reaches $70.5\%$ accuracy, an $+8.3$ percentage-point gain over the best baseline (Bulyan at $62.2\%$). For inverse-gradient attacks with $q = 0.9$ and $40\%$ adversaries, it delivers a $+3.1$ percentage-point improvement over the best baseline ($59.38\%$ vs. $56.24\%$), while RFA and CClip variants diverge. In the challenging Double attack, FedLAW delivers the best results among state-of-the-art methods. Likewise, in Lie Attack, FedLAW performs on par with the strongest baselines, while substantially outperforming many others that suffer severe accuracy degradation. A key observation is the rapid convergence of the weights. The weights $\mathbf{w}$ typically stabilize within the first 20 rounds, after which further updates have negligible effect.

## 6 CONCLUSION

This paper introduces FedLAW, a Byzantine-robust Federated Learning framework that treats aggregation weights as learnable parameters, optimized alongside the global model. By enforcing a sparsity constraint, our method effectively neutralizes the influence of malicious clients while adaptively balancing contributions from benign clients. To solve the resulting joint optimization problem, we develop an alternating minimization algorithm that updates weights and model parameters in tandem. We prove convergence guarantees and establish theoretical resilience to adversarial behavior. Extensive empirical results on MNIST and CIFAR10 datasets under multiple attack scenarios demonstrate the robustness and effectiveness of FedLAW. Compared to existing Byzantine-robust algorithms, our method achieves consistently higher accuracy, especially in highly non-IID and adversarial environments. These findings highlight the benefits of integrating adaptive aggregation into the learning process, paving the way for more secure and equitable FL deployments.

ACKNOWLEDGEMENT

This work is supported by the Swedish Research Council under the grant 2023-05234, the Swedish Foundation for Strategic Research, the Knut and Alice Wallenberg Foundation, and Sweden's Innovation Agency. The computations and data handling were enabled by resources provided by the National Academic Infrastructure for Supercomputing in Sweden (NAISS), partially funded by the Swedish Research Council through grant agreement no. 2022-06725.

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

CONTENTS

---

**Algorithm 1** CLIENTUPDATE

---

**Require:** current global model $\boldsymbol{\theta}$; learning-rate $\alpha$; local epochs $E$; batch-size $B$; private data $\mathcal{D}_i$

1: $\psi_i \leftarrow \boldsymbol{\theta}$      ▷ initialize local model parameters
2: **for** $e = 1$ **to** $E$ **do**      ▷ full local epochs
3:      **for all** mini-batches $\mathcal{B} \subset \mathcal{D}_i$ of size $B$ **do**
4:          $\psi_i \leftarrow \psi_i - \alpha \nabla_\psi f_i(\psi_i, \mathcal{B})$
5:      **end for**
6: **end for**
7: $\mathbf{g}_i \leftarrow -\dfrac{\psi_i - \boldsymbol{\theta}}{\alpha}$      ▷ gradient at $\boldsymbol{\theta}$
8: $f_i \leftarrow f_i(\psi_i)$
9: **return** $(\mathbf{g}_i, f_i)$

---

**Algorithm 2** FedLAW (Federated Learning with Learnable Aggregation Weights)

---

**Require:** initial model $\boldsymbol{\theta}_0$; learning rates $\alpha, \beta$; total epochs $T$; local epochs $E$; batch-size $B$; sparsity level $s$; cap $t$; number of clients $n$

1: initialize weights $\mathbf{w}_0 \leftarrow \frac{1}{n}\mathbf{1}$
2: **for** $k = 0$ **to** $T - 1$ **do**
         // —— collect client updates —— //
3:      broadcast $\boldsymbol{\theta}_k$ to all clients
4:      **for all** clients $i = 1{:}n$ **in parallel do**
5:          $(\mathbf{g}_i, f_i) \leftarrow$ CLIENTUPDATE$(\boldsymbol{\theta}_k, \alpha, E, B)$
6:      **end for**
         // —— server step 1 —— //
7:      $\mathbf{g}_{\text{agg}} \leftarrow \sum_{i=1}^{n} w_{k,i}\, \mathbf{g}_i$
8:      $\tilde{\boldsymbol{\theta}}_{k+1} \leftarrow \boldsymbol{\theta}_k - \alpha\, \mathbf{g}_{\text{agg}}$
9:      broadcast $\tilde{\boldsymbol{\theta}}_{k+1}$ and collect $(\tilde{\mathbf{g}}_i, \tilde{f}_i)$ via CLIENTUPDATE
         // —— server step 2 (weight update) —— //
10:      $\mathbf{G} = [\mathbf{g}_1, \cdots, \mathbf{g}_n], \quad \tilde{\mathbf{G}} = [\tilde{\mathbf{g}}_1, \cdots, \tilde{\mathbf{g}}_n]$
11:      $\tilde{\mathbf{f}} = [\tilde{f}_1, \ldots, \tilde{f}_n]^\top$
12:      $\mathbf{h}_k \leftarrow \mathbf{w}_k + \alpha\beta\, \mathbf{G}^\top \tilde{\mathbf{G}}\, \mathbf{w}_k - \beta\, \tilde{\mathbf{f}}$
13:      $\mathbf{w}_{k+1} \leftarrow \text{Proj}_{\Delta_{t,\ell_0}^+}(\mathbf{h}_k)$ with $s$ via (10)
         // —— server step 3 (second model update) —— //
14:      $\mathbf{g}_{\text{agg}} \leftarrow \sum_{i=1}^{n} w_{k+1,i}\, \mathbf{g}_i$
15:      $\boldsymbol{\theta}_{k+1} \leftarrow \boldsymbol{\theta}_k - \alpha\, \mathbf{g}_{\text{agg}}$
16: **end for**
17: **return** $\boldsymbol{\theta}_T$

---

## A    MATHEMATICAL FOUNDATIONS

In this section, we first introduce key mathematical concepts used throughout the paper. We then present specific structural properties and feasibility results related to our proposed optimization framework.

We begin with the definition of proximal mapping.

**Definition 1.** *(Parikh & Boyd, 2014) Let $p\colon dom_p \to (-\infty, +\infty]$ be a proper and lower semi-continuous (PLSC) function. Then the proximal mapping of $p$ at $\mathbf{x} \in \mathbb{R}^n$ is defined as*

$$prox_p(\mathbf{x}) = \operatorname*{argmin}_{\mathbf{u} \in dom_p} \left\{ \frac{1}{2}\|\mathbf{x} - \mathbf{u}\|_2^2 + p(\mathbf{u}) \right\}. \tag{16}$$

**Definition 2.** *(Attouch et al., 2011) The subdifferential of a PLSC function $g$ at $\mathbf{x} \in \mathbb{R}^n$ is defined as*

$$\partial g(\mathbf{x}) \overset{\triangle}{=} \{\zeta \in \mathbb{R}^n \mid \exists \mathbf{x}_k \to \mathbf{x},\, g(\mathbf{x}_k) \to g(\mathbf{x}),\, \zeta_k \to \zeta \in \partial \hat{g}(\mathbf{x}_k)\}$$

where $\partial\hat{g}(\mathbf{x}_k)$ is the Fréchet subdifferential of $g$ at $\mathbf{x}_k \in \mathbb{R}^n$, defined as

$$\partial\hat{g}(\mathbf{x}_k) = \left\{ \zeta \in \mathbb{R}^n \mid \liminf_{\mathbf{v} \neq \mathbf{x}, \mathbf{v} \to \mathbf{x}} \frac{1}{\|\mathbf{v} - \mathbf{x}\|_2^2} \left[ g(\mathbf{v}) - g(\mathbf{x}) - \langle \mathbf{v} - \mathbf{x}, \zeta \rangle \right] \geq 0 \right\}. \tag{17}$$

**Definition 3.** *(Attouch et al., 2011) A point $\mathbf{x}^*$ is called a critical point of a PLSC function $f(\mathbf{x})$ if $0 \in \partial f(\mathbf{x}^*)$.*

**Definition 4** (Operator $P_{L_s}$). *The operator $P_{L_s}(\mathbf{w})$ leaves the $s$ largest elements (based on their values, not magnitudes) of the vector $\mathbf{w}$ unaltered and sets all other entries to zero.*

**Lemma 4** (Descent lemma (Bolte et al., 2014)). *Let $f : \mathbb{R}^n \to \mathbb{R}$ be a continuously differentiable function whose gradient $\nabla f$ is L-Lipschitz continuous. Then, for all $\mathbf{x}, \mathbf{y} \in \mathbb{R}^n$:*

$$f(\mathbf{y}) \leq f(\mathbf{x}) + \langle \nabla f(\mathbf{x}), \mathbf{y} - \mathbf{x} \rangle + \frac{L}{2} \|\mathbf{y} - \mathbf{x}\|_2^2. \tag{18}$$

**Definition 5** (($\zeta, b_f$)-Byzantine Resilience (Blanchard et al., 2017)). *Let $\zeta$ be any angular value in the interval $[0, \pi/2)$, and $b_f$ be any integer in $\{0, 1, \ldots, n\}$. Consider $n$ independent identically distributed (i.i.d.) random vectors $\mathbf{v}_1, \ldots, \mathbf{v}_n$ in $\mathbb{R}^d$ with $\mathbf{v}_i \sim G$ where $\mathbb{E}\{G\} = g$, and $b_f$ random vectors $\mathbf{b}_1, \ldots, \mathbf{b}_{b_f}$ in $\mathbb{R}^d$, possibly dependent on the $\mathbf{v}_i$'s. A choice function $F$ is said to be $(\zeta, b_f)$-Byzantine resilient if, for any $1 \leq j_1 < \cdots < j_{b_f} \leq n$, the vector*

$$F = F(\mathbf{v}_1, \ldots, \mathbf{b}_{j_1}, \ldots, \mathbf{b}_{j_{b_f}}, \ldots, \mathbf{v}_n) \tag{19}$$

*satisfies:*

1. *It maintains alignment with the expected gradient:*

$$\langle \mathbb{E}\{F\}, g \rangle \geq (1 - \sin \alpha) \|g\|^2 > 0.$$

2. $\mathbb{E}\{\|F\|^r\}$ *(for $r = 2, 3, 4$) satiates the following condition:*

$$\mathbb{E}\{\|F\|^r\} \leq C \sum \mathbb{E}\{\|G\|^{r_i}\}, \quad \text{where} \sum r_i = r.$$

**Definition 6** (Kurdyka-Łojasiewicz Property (Attouch & Bolte, 2009)). *A function $f : \mathbb{R}^d \to \mathbb{R} \cup \{+\infty\}$ satisfies the Kurdyka-Łojasiewicz (KL) property at a point $\bar{x} \in dom(\partial f)$ if there exist $\eta > 0$, a neighborhood $U$ of $\bar{x}$, and a continuous concave function $\phi : [0, \eta) \to [0, +\infty)$ with $\phi(0) = 0$, $\phi$ differentiable, and $\phi'(s) > 0$ for $s \in (0, \eta)$, such that for all $x \in U$ with $f(\bar{x}) < f(x) < f(\bar{x}) + \eta$,*

$$\phi'(f(x) - f(\bar{x})) \cdot dist(0, \partial f(x)) \geq 1,$$

*where $dist(0, \partial f(x)) = \inf\{\|v\| \mid v \in \partial f(x)\}$. A function satisfying the KL property at all points in $dom(\partial f)$ is called a KL function.*

**Proposition 2.** *Consider the sparse-unit capped simplex*

$$\Delta_{t,\ell_0}^+ = \left\{ \mathbf{w} \in \mathbb{R}^n \mid \sum_{i=1}^n w_i = 1, w_i \geq 0, w_i \leq t, \|\mathbf{w}\|_0 \leq s \right\} \tag{20}$$

*with $t = \frac{1}{n-b_f}$ and $s = n - b_f$. Then any weight vector $\mathbf{w} \in \Delta_{t,\ell_0}^+$ has exactly $n - b_f$ non-zero weights, each equal to $\frac{1}{n-b_f}$, and the remaining $b_f$ weights are zero.*

*Proof.* Let $\mathbf{w}$ have $k \leq n - b_f$ non-zero weights indexed by $\mathcal{S}$, so $\sum_{i \in \mathcal{S}} w_i = 1$, $0 < w_i \leq \frac{1}{n-b_f}$ for all $i \in \mathcal{S}$. If $k < n - b_f$, we have:

$$\sum_{i=1}^n w_i \leq k \cdot \frac{1}{n-b_f} < (n - b_f) \cdot \frac{1}{n-b_f} = 1,$$

which cannot satisfy $\sum_{i=1}^n w_i = 1$. Thus, $k = n - b_f$. Suppose at least one weight, say $w_1 < \frac{1}{n-b_f}$. Then:

$$\sum_{i=2}^{n-b_f} w_i = 1 - w_1 > 1 - \frac{1}{n-b_f} = \frac{n - b_f - 1}{n - b_f}.$$

The maximum sum for the remaining $n - b_f - 1$ weights, each capped at $\frac{1}{n-b_f}$, is $\frac{n-b_f-1}{n-b_f}$. Since the required sum $\sum_{i=2}^{n-b_f} w_i > \frac{n-b_f-1}{n-b_f}$ exceeds the maximum possible sum, this is impossible. Thus, $w_1 < \frac{1}{n-b_f}$ leads to a contradiction. Similarly, $w_i > \frac{1}{n-b_f}$ violates the upper bound $w_i \leq \frac{1}{n-b_f}$.

Hence, all $n - b_f$ non-zero weights must be $\frac{1}{n-b_f}$, and the remaining $b_f$ weights are zero, satisfying all constraints. $\qquad\square$

In the following lemma, we determine when the sparse, unit-sum, $t$-capped simplex is non-empty, a key condition for selecting valid sparsity and cap parameters in the proposed method (1). We also compute an upper bound of the Euclidean distance between any two vectors in this set that we will use in later results.

**Lemma 5** (Feasibility and Distance in Sparse Unit-Capped Simplex). *Consider the sparse unit-capped simplex* (20) *in which,* $n \geq 2$, $s \in \{1, \ldots, n\}$, *and* $t \in (0, 1]$. *Then* $\Delta_{t,\ell_0}^+ \neq \emptyset$ *iff* $st \geq 1$, *and for any* $\mathbf{w}_1, \mathbf{w}_2 \in \Delta_{t,\ell_0}^+$,

$$\|\mathbf{w}_1 - \mathbf{w}_2\|_2 \leq \sqrt{2\left(kt^2 + r^2\right)}. \tag{21}$$

*where* $k = \lfloor 1/t \rfloor$ *and* $r = 1 - kt \in [0, t)$. *If* $1/t \in \mathbb{N}$, *the bound simplifies to* $\|\mathbf{w}_1 - \mathbf{w}_2\|_2 \leq \sqrt{2t}$.

*Proof.* ($\Rightarrow$) If $\Delta_{t,\ell_0}^+ \neq \emptyset$, then there exists a $t$-capped, $s$-sparse vector summing to 1. Since each nonzero entry is at most $t$, we must have $st \geq 1$.

($\Leftarrow$) If $st \geq 1$, we can construct a feasible vector by setting $k = \lfloor 1/t \rfloor$, $r = 1 - kt$, and defining

$$\mathbf{w}^\star = [\underbrace{t, \ldots, t}_{k \text{ times}}, r, 0, \ldots, 0],$$

which has at most $s$ nonzero entries and sums to 1, so $\mathbf{w}^\star \in \Delta_{t,\ell_0}^+$.

The maximum $\ell_2$ norm in $\Delta_{t,\ell_0}^+$ is achieved by $\mathbf{w}^\star = [\underbrace{t, \ldots, t}_{k \text{ times}}, r, 0, \ldots, 0]$ with $\|\mathbf{w}^\star\|_2^2 = kt^2 + r^2$.

For any $\mathbf{w}_1, \mathbf{w}_2 \in \Delta_{t,\ell_0}^+$, we have

$$\|\mathbf{w}_1 - \mathbf{w}_2\|_2^2 = \|\mathbf{w}_1\|_2^2 + \|\mathbf{w}_2\|_2^2 - 2\langle \mathbf{w}_1, \mathbf{w}_2 \rangle \leq 2(kt^2 + r^2),$$

because $\langle \mathbf{w}_1, \mathbf{w}_2 \rangle \geq 0$ (all entries are non-negative). Taking square roots yields the result. $\qquad\square$

## B  OPTIMIZING THE JOINT LOSS FUNCTION: FEDLAW AND BSUM

A core contribution of this paper is to address Byzantine-robust federated learning by formulating and solving the following joint optimization problem over global model parameters $\boldsymbol{\theta} \in \mathbb{R}^d$ and aggregation weights $\mathbf{w} = [w_1, \ldots, w_n]$:

$$\min_{\boldsymbol{\theta} \in \mathbb{R}^d, \mathbf{w} \in \Delta_{t,\ell_0}^+} \sum_{i=1}^n w_i f_i(\boldsymbol{\theta}), \tag{22}$$

where $\Delta_{t,\ell_0}^+ = \{\mathbf{w} \in \mathbb{R}^n \mid \sum_{i=1}^n w_i = 1, w_i \geq 0, w_i \leq t, \|\mathbf{w}\|_0 \leq s\}$.

As detailed in Section 3.2, our primary solution to this problem is the FedLAW algorithm, whose pseudocode is provided in Algorithm 2. As another contribution, we adapt the *Block Successive Upper-bound Minimization (BSUM)* method (Razaviyayn et al., 2013) to solve (22), providing a baseline for our novel algorithm. By comparing FedLAW and BSUM, we demonstrate that FedLAW's integrated use of loss and gradient information results in more robust and efficient learning under adversarial conditions. Below, we detail both methods and their comparative performance.

## B.1 BSUM: A BASELINE FOR COMPARISON

BSUM alternates between updating the global model parameters $\boldsymbol{\theta}$ and client weights $\mathbf{w}$, minimizing a local upper bound approximation of the objective for each block while fixing the other. At iteration $k + 1$, it performs the following operations:

1. **Update $\boldsymbol{\theta}$**: With $\mathbf{w} = \mathbf{w}_k$, solve:

$$\min_{\boldsymbol{\theta} \in \mathbb{R}^d} \sum_{i=1}^n w_{k,i} f_i(\boldsymbol{\theta}). \tag{23}$$

Assuming differentiable $f_i(\boldsymbol{\theta})$, we provide a quadratic upper bound $\hat{f}_i(\boldsymbol{\theta})$ of $f_i(\boldsymbol{\theta})$

$$\hat{f}_i(\boldsymbol{\theta}; \boldsymbol{\theta}_k) = f_i(\boldsymbol{\theta}_k) + \langle \nabla_\theta f_i(\boldsymbol{\theta}_k), \boldsymbol{\theta} - \boldsymbol{\theta}_k \rangle + \frac{1}{2\alpha} \|\boldsymbol{\theta} - \boldsymbol{\theta}_k\|_2^2. \tag{24}$$

where $\alpha > 0$ is a step size. Substituting this quadratic upper bound into (23) gives

$$\tilde{\boldsymbol{\theta}}_{k+1} = \operatorname*{argmin}_{\boldsymbol{\theta} \in \mathbb{R}^d} \sum_{i=1}^n w_i \hat{f}_i(\boldsymbol{\theta}; \boldsymbol{\theta}_k) = \boldsymbol{\theta}_k - \alpha \mathbf{G}_k \mathbf{w}_k, \tag{25}$$

in which $\mathbf{G}_k = [\nabla_\theta f_1(\boldsymbol{\theta}_k), \cdots, \nabla_\theta f_n(\boldsymbol{\theta}_k)]$.

2. **Update $\mathbf{w}$**: With $\boldsymbol{\theta} = \tilde{\boldsymbol{\theta}}_{k+1}$, solve:

$$\mathbf{w}_{k+1} = \operatorname*{argmin}_{\mathbf{w} \in \Delta_{t,\ell_0}^+} \sum_{i=1}^n w_i f_i(\tilde{\boldsymbol{\theta}}_{k+1}). \tag{26}$$

This is a linear program in $\mathbf{w}$, constrained by the sparse unit-capped simplex $\Delta_{t,\ell_0}^+$. To ensure a fair comparison with our method, which uses the most recent weights to update $\boldsymbol{\theta}$ for enhanced robustness, we apply an extrapolation step, updating:

$$\boldsymbol{\theta}_{k+1} = \boldsymbol{\theta}_k - \alpha \mathbf{G}_k \mathbf{w}_{k+1}.$$

This extrapolation aligns BSUM's $\boldsymbol{\theta}$ update with our method's strategy, using the latest weights $\mathbf{w}_{k+1}$ optimized at $\tilde{\boldsymbol{\theta}}_{k+1}$ to reflect current client reliability assessments. By incorporating $\mathbf{w}_{k+1}$, which downweights malicious clients via the sparse capped simplex, the update improves robustness against Byzantine attacks, ensuring that performance differences arise from the algorithm design rather than the precise ordering of updates.

## B.2 FEDLAW VS. BSUM: CAPTURING JOINT OPTIMIZATION DYNAMICS

A key difference distinguishes our method (Equation (6)) from BSUM. In BSUM, $\tilde{\boldsymbol{\theta}}_{k+1}$ is fixed when updating $\mathbf{w}$, simplifying (26) to a linear objective. In contrast, our method minimizes:

$$\sum_{i=1}^n w_i f_i(\boldsymbol{\theta}_k - \alpha \mathbf{G}_k \mathbf{w}). \tag{27}$$

When the weight vector $\mathbf{w}$ inside $f_i$ is fixed to $\mathbf{w}_k$, our method (Equation (6)) reduces to BSUM's linear program (Equation (26)), revealing that BSUM is a suboptimal approximation. This raises a key question: how does our method compare to BSUM in Byzantine-robust federated learning? BSUM updates $\mathbf{w}$ by minimizing $\sum_{i=1}^n w_i f_i(\tilde{\boldsymbol{\theta}}_{k+1})$, relying solely on client losses $f_i(\tilde{\boldsymbol{\theta}}_{k+1})$ to detect malicious clients. However, attacks like the inverse gradient attack, where malicious clients submit flipped gradients but benign losses, are not immediately detectable in losses, delaying BSUM's exclusion of malicious clients and slowing convergence. In contrast, our method optimizes $\sum_{i=1}^n w_i f_i(\boldsymbol{\theta}_k - \alpha \mathbf{G}_k \mathbf{w})$, leveraging both losses and gradients to enhance robustness.

The objective in (27) finds the most internally coherent client subgroup, and the premise is not that the descent direction of a single client matters.

The objective in Eq. (27) can be understood as an inner optimization problem. For a candidate subgroup defined by weights $w$:

1. It first computes a collective descent direction, $d(w) = \mathbf{G}_k w$. This represents the consensus update direction proposed by the subgroup $w$.

2. It then evaluates the quality of this direction by measuring the post-update empirical loss, $L(w) = \sum_i w_i f_i(\boldsymbol{\theta}_k - \alpha d(w))$. This step checks if the collective descent direction $d(w)$ is actually effective for the same subgroup $w$ that proposed it.

The $\arg\min_w$ in Eq. (5) operation therefore searches for the weight vector $w$ that defines a subgroup whose collective descent direction is most effective *for itself*.

- A coherent subgroup (e.g., benign clients, which tend to align due to similar gradient directions) will propose a descent direction $d(w)$ that is effective for all its members, achieving a low objective value.

- Including a malicious outlier poisons $d(w)$, making it a poor descent direction for the benign majority, thus penalizing that choice of $w$.

Our algorithm solves (6) via the projection in Equation (9), where $\mathbf{w}_{k+1} = \operatorname{prox}_{\Delta_{t,\ell_0}^+}(\mathbf{h}_k)$ and $\mathbf{h}_k = \mathbf{w}_k + \beta\alpha\mathbf{G}_k^T\tilde{\mathbf{G}}_{k+1}\mathbf{w}_k - \beta\mathbf{f}(\boldsymbol{\theta}_k - \alpha\mathbf{G}_k\mathbf{w}_k)$. Here, $\tilde{\mathbf{G}}_{k+1} = [\nabla_\theta f_1(\boldsymbol{\theta}_k - \alpha\mathbf{G}_k\mathbf{w}_k), \ldots, \nabla_\theta f_n(\boldsymbol{\theta}_k - \alpha\mathbf{G}_k\mathbf{w}_k)]$ and $\mathbf{f}(\boldsymbol{\theta}_k - \alpha\mathbf{G}_k\mathbf{w}_k) = [f_1(\boldsymbol{\theta}_k - \alpha\mathbf{G}_k\mathbf{w}_k), \ldots, f_n(\boldsymbol{\theta}_k - \alpha\mathbf{G}_k\mathbf{w}_k)]$. The vector $\mathbf{h}_k$ incorporates two factors for detecting malicious clients:

1. **Losses ($\mathbf{f}(\boldsymbol{\theta}_k - \alpha\mathbf{G}_k\mathbf{w}_k)$):** This term, similar to BSUM, assesses client losses. It is effective against attacks like data poisoning, which directly alter losses, but is less reliable when losses remain benign despite malicious gradients.

2. **Gradient Alignment ($\mathbf{G}_k^T\tilde{\mathbf{G}}_{k+1}$):** This inner product quantifies the alignment between gradients at $\boldsymbol{\theta}_k$ and $\boldsymbol{\theta}_k - \alpha\mathbf{G}_k\mathbf{w}_k$ across consecutive rounds. In attacks such as the inverse gradient attack, malicious gradients typically misalign in direction relative to benign ones, enabling early detection and exclusion through the sparsity constraint $\|\mathbf{w}\|_0 \leq s$.

To substantiate this comparison, we provide an empirical evaluation of our method against BSUM (labeled as "FedLAW-BSUM" in the figures), offering evidence to support the theoretical advantages.

**Empirical comparison.** Figures 3 and 4 compare our method with BSUM under four adversarial attack scenarios on MNIST: flipping label, inverse gradient, backdoor, and double attack.

**Figure 3** shows the test accuracy over 200 epochs. Across all attack settings, our method (FedLAW, dashed green) either matches or surpasses the performance of BSUM (solid blue), with the most significant improvements observed under gradient-based attacks. In particular, under the inverse gradient and double attack scenarios, FedLAW converges faster and reaches a higher final accuracy, highlighting its enhanced robustness in detecting and mitigating subtle gradient manipulations.

**Figure 4** further illustrates the client-weight evolution under these adversarial conditions. The top two rows correspond to FedLAW, and the bottom two rows to BSUM. Each subplot displays how the aggregation weight of each client evolves across 100 epochs. While both methods eventually suppress malicious clients (red/orange), FedLAW consistently achieves this suppression earlier, especially under inverse gradient and double attacks, due to its reliance on both loss and gradient information. In contrast, BSUM, which only considers losses, fails to immediately detect malicious clients whose losses remain close to benign ones, delaying their exclusion.

These findings demonstrate that FedLAW more effectively captures the joint optimization dynamics in (22) by incorporating both loss and gradient-based alignment into the weight update process compared to BSUM. FedLAW not only detects malicious behavior earlier but also maintains better model accuracy and stability in the presence of sophisticated adversaries.

## C  THEORETICAL ANALYSIS: A MULTI-LAYERED DEFENSE AGAINST STRATEGIC ATTACKS

We address the critical question of whether a strategic attacker can manipulate our weight-learning process to gain influence. In this section, we present a theoretical analysis of our multi-layered

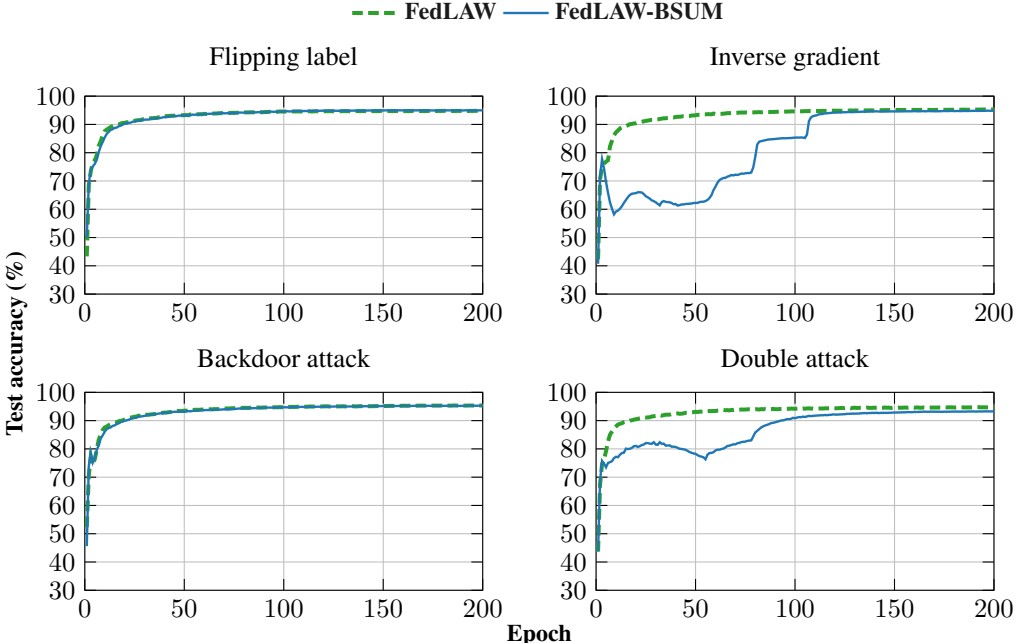

Figure 3: Test accuracy on MNIST for four attack settings ($q$=0.9, 10 clients, $40\%$ malicious), contrasting **FedLAW-BSUM** (solid blue) with **FedLAW** (dashed green).

defense: we study the dynamics of our method, design strategic attacks that attempt to deceive it, and examine how our approach responds to these mimicry attacks.

LAYER 1: FOUNDATIONAL CONSTRAINTS AND ATTACK MODEL

Our defense is built upon a set of foundational constraints that limit an attacker's power a priori.

**Constraints.**

1. **Loss preprocessing.** As a practical first line of defense, the server can filter naive attackers by detecting clients whose reported losses $\{\tilde{f}_{k+1,p}\}$ are statistical outliers (e.g., using a Median Absolute Deviation (MAD) test to flag anomalously low values).

2. **Gradient norm bound.** We assume $\ell_2$ norm of Byzantine is bounded by $\ell_2$ norm of honest mini-batch gradients, which can be handled by gradient clipping.

3. **Weight Capping:** The optimization constraint $\mathbf{w} \in \Delta^+_{t,\ell_0}$ ensures no client weight can exceed the cap $t$. To exactly exclude the $b_f$ clients, $t = \frac{1}{n-b_f}$. This provides a hard limit on the influence of any single client, even in a hypothetical worst-case scenario.

**Remark 3.** *The Layer 1 preprocessing is fully aligned with Assumption 1 and our theoretical analysis. The latter two components are explicitly incorporated in Theorem 2. In addition, the loss-based filtering step is consistent with the assumptions in Theorem 2, where we assume that the loss heterogeneity $\varepsilon_k$ among honest clients is small; this is also supported by the numerical study in Section I.5. Hence, honest clients are unlikely to be excluded by this preprocessing step. Moreover, this filtering helps tighten the bounds in our Byzantine analysis: in (93), the term*

$$\left( \sum_{i \in \Lambda^d} f_i(\boldsymbol{\theta}_k) - \sum_{j \in \Lambda^b} \tilde{f}_j(\boldsymbol{\theta}_k) \right)$$

*directly appears in the upper bound on the bias of our aggregator, and by removing extreme outliers, we prevent this quantity from becoming large.*

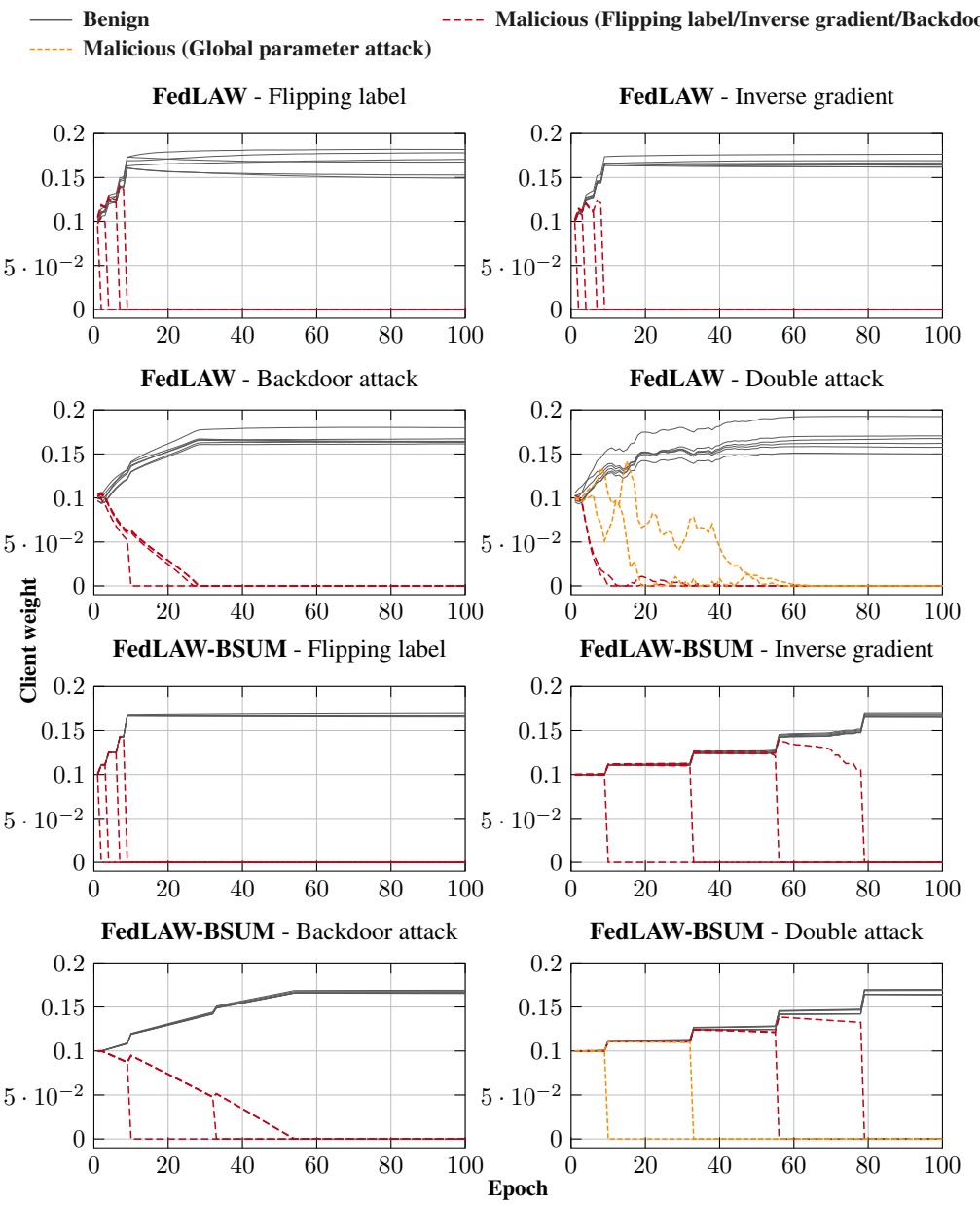

Figure 4: Client-weight dynamics on MNIST under four adversarial settings ($q$=0.9, 10 clients, 40% malicious). **Two top row:** FedLAW. **Two bottom row:** FedLAW-BSUM. Each panel tracks the aggregation weight of every client during the first 100 global epochs for a three-layer MLP (batch size 64, three local epochs). Across all attacks, benign clients (grey) quickly converge to a stable weight, while malicious clients (red/orange) are pushed towards negligible influence. Notably, FedLAW suppresses attackers *faster* than BSUM, especially for the gradient-based attacks (inverse gradient, double attack), illustrating its stronger resilience.

We analyze the critical moment an attack begins. At round $k$, all clients are honest and weights are uniform ($\mathbf{w}_k = \frac{1}{n}\mathbf{1}$). At round $k+1$, clients in a set $\mathcal{H}^{\complement}$ become Byzantine.

LAYER 2: THEORETICAL ANALYSIS OF THE CORE DETECTION MECHANISM

With the attacker's power constrained bounded by $\ell_2$ norm of honest gradients, we now analyze how our algorithm (Eq. (10)) is designed to detect malicious intent. The server computes $\tilde{\boldsymbol{\theta}}_{k+1} = \boldsymbol{\theta}_k - \alpha G_k \mathbf{w}_k$, then collects gradients and losses at $\tilde{\boldsymbol{\theta}}_{k+1}$ from honest clients ($\mathcal{H}$) and Byzantine clients ($\mathcal{H}^{\complement}$). Based on these, it builds the pre-projection vector $\mathbf{h}_k$ with components:

$$h_{k,j} = \frac{1}{n} + \frac{\beta\alpha}{n}\langle \tilde{\mathbf{v}}_{k,j}, \tilde{\mathbf{v}}_{\mathcal{H}} + \mathbf{b}_{\mathcal{H}^{\complement}}\rangle - \beta f_j(\tilde{\boldsymbol{\theta}}_{k+1}). \tag{28}$$

In the next step, the server keeps the $s$ largest entries of $\mathbf{h}_k$ and sets all others to zero. For a Byzantine client $j \in \mathcal{H}^{\complement}$ to remain in the support, its score must dominate that of some honest client $i \in \mathcal{H}$, i.e.,

$$h_{k,j} \geq h_{k,i}. \tag{29}$$

The above inequality can be written by

$$\Delta_{ji,k} = \frac{1}{\beta}(h_{k,j} - h_{k,i}) = \alpha \underbrace{\langle \tilde{\mathbf{v}}_{k,j} - \tilde{\mathbf{v}}_{k,i}, \tilde{\mathbf{v}}_{\mathcal{H}} + \mathbf{b}_{\mathcal{H}^{\complement}}\rangle}_{\text{alignment}} + f_i(\tilde{\boldsymbol{\theta}}_{k+1}) - \hat{f}_j(\tilde{\boldsymbol{\theta}}_{k+1}) \geq 0 \tag{30}$$

We intentionally evaluate FedLAW in the most challenging setting, where in the first attack round it may happen that $\Delta_{ji,k} \geq 0$. Consequently, at round $k+1$ our method can temporarily assign client $j$ a slightly larger weight (still strictly bounded by the cap $t$).

Crucially, FedLAW is adaptive and iterative: any temporary over-weighting of a misaligned client strengthens its contribution to the aggregated gradient in the next round, which in turn accentuates its misalignment signal. Because $w_{k+1,j} > w_{k,j}$ and the weights at round $k+1$ are no longer uniform, the next pre-projection vector $\mathbf{h}_{k+1}$ takes the form

$$h_{k+1,j} = w_{k+1,j} + \alpha\beta\langle \mathbf{b}_{k+1,j}, \sum_{i\in\mathcal{H}} w_{k+1,i}\tilde{\mathbf{v}}_{k+2,i} + \sum_{j\in\mathcal{H}^{\complement}} w_{k+1,j}\mathbf{b}_{k+2,j}\rangle - \beta\hat{f}_j(\tilde{\boldsymbol{\theta}}_{k+2})$$

$$h_{k+1,i} = w_{k+1,i} + \alpha\beta\langle \tilde{\mathbf{v}}_{k+1,i}, \sum_{i\in\mathcal{H}} w_{k+1,i}\tilde{\mathbf{v}}_{k+2,i} + \sum_{j\in\mathcal{H}^{\complement}} w_{k+1,j}\mathbf{b}_{k+2,j}\rangle - \beta f_i(\tilde{\boldsymbol{\theta}}_{k+2}) \tag{31}$$

Then, we have

$$\Delta_{ji,k+1} = w_{k+1,j} - w_{k+1,i} + \alpha\beta \underbrace{\langle \mathbf{b}_{k+1,j} - \tilde{\mathbf{v}}_{k+1,i}, \sum_{i\in\mathcal{H}} w_{k+1,i}\tilde{\mathbf{v}}_{k+2,i} + \sum_{j\in\mathcal{H}^{\complement}} w_{k+1,j}\mathbf{b}_{k+2,j}\rangle}_{\text{misalignment term}}$$

$$+ \beta f_i(\tilde{\boldsymbol{\theta}}_{k+2}) - \beta\hat{f}_j(\tilde{\boldsymbol{\theta}}_{k+2}). \tag{32}$$

Next, we define the following score

$$s_{ji,k+1} = \alpha\langle \mathbf{b}_{k+1,j} - \tilde{\mathbf{v}}_{k+1,i}, \tilde{\mathbf{F}}\rangle + f_i(\tilde{\boldsymbol{\theta}}_{k+2}) - \hat{f}_j(\tilde{\boldsymbol{\theta}}_{k+2}) \tag{33}$$

in which $\tilde{\mathbf{F}} = \sum_{i\in\mathcal{H}} w_{k+1,i}\tilde{\mathbf{v}}_{k+2,i} + \sum_{j\in\mathcal{H}^{\complement}} w_{k+1,j}\mathbf{b}_{k+2,j}$ is our aggregator.

Given (33), the loss difference and the misalignment term are the two signals FedLAW uses to detect malicious clients. In *data poisoning* (e.g., label flipping, backdoor), the attacker changes its data distribution. For a model $\tilde{\boldsymbol{\theta}}$ trained on clean data, this yields (i) a higher poisoned loss $\hat{f}_j(\tilde{\boldsymbol{\theta}}) > f_i(\tilde{\boldsymbol{\theta}})$ and (ii) a gradient $\mathbf{b}_{k+1,j}$ that conflicts with the coherent honest cluster. Thus, *both* terms in $s_{ji,k+1}$ work against the Byzantine client. In a *model attack*, the data remain clean but the submitted gradient $\mathbf{b}_{k+1,j}$ is corrupted (e.g., inverse gradient), so the loss is similar to honest clients, and the misalignment term in $s_{ji,k+1}$ is the main discriminator.

**Mimic attacks:** In the most challenging case, an adversary tries to cancel the negative misalignment term by slightly under-reporting its loss. It must choose $\hat{f}_j(\tilde{\boldsymbol{\theta}}_{k+1}) < f_j(\tilde{\boldsymbol{\theta}}_{k+1})$ while still sending a misaligned gradient. If $\hat{f}_j(\tilde{\boldsymbol{\theta}}_{k+1}) \ll f_j(\tilde{\boldsymbol{\theta}}_{k+1})$, this is easily caught by the loss preprocessing in Layer 1. The harder regime is when the reported loss is only slightly smaller than the honest level, so that $f_i(\tilde{\boldsymbol{\theta}}_{k+1}) - \hat{f}_j(\tilde{\boldsymbol{\theta}}_{k+1})$ partially offsets the alignment penalty in (30). In that first attack round, it may happen that $\Delta_{ji,k} \geq 0$, so at round $k+1$ FedLAW can temporarily assign client $j$ a somewhat larger weight (still bounded above by the cap $t$).

We leverage the proven Byzantine-resilience of our aggregator (Theorem 2). This theorem guarantees that the mean of the aggregated gradient, $\tilde{\mathbf{F}}$, maintains a positive alignment with the true honest mean, i.e.

$$\mathbf{g}_{k+1} = \frac{1}{n - b_f} \sum_{i \in \mathcal{H}} \mathbb{E}\{\mathbf{v}_{k+1,i}\} \tag{34}$$

An effective attack requires the malicious gradient $\mathbf{b}_{k+1,j}$ to be misaligned with this honest direction. The honest mini-batch gradient $\tilde{\mathbf{v}}_{k+1,i}$ is, by definition, coherent with the honest mean $\mathbf{g}_{k+1}$, which is guaranteed by Theorem 2 to be in the same general direction as $\tilde{\mathbf{F}}$. Consequently, for an effective attack

$$\langle \mathbf{b}_{k+1,j} - \tilde{\mathbf{v}}_{k+1,i}, \tilde{\mathbf{F}} \rangle < 0 \tag{35}$$

So, in (33), although the loss difference is positive, the misalignment term above remains negative, which helps to neutralize the effect of the attack. In a strong attack, this term becomes more negative, meaning the loss difference cannot be strongly positive; otherwise, the attack could be detected by loss preprocessing. Consequently, in this case the score $s_{ji,k+1}$ becomes negative and the weight of the Byzantine client starts to decrease. In this mimic attack, due to its more sophisticated nature, FedLAW may require more epochs to reduce the weights of malicious clients to zero compared to data poisoning and inverse-gradient attacks.

An adversary can try to weaken this effect by sending Byzantine gradients that are aligned with the average honest gradient, in which case the misalignment term in (32) becomes close to zero. However, such an attack is inherently weak: since each client's weight is capped by $t = 1/(n - b_f)$ (usually $n \gg b_f$) and the Byzantine gradients closely follow the honest direction, they cannot significantly steer the global model away from the benign trajectory. This behavior is consistent with our empirical results under the LIE attack in Fig. 2. In LIE, Byzantine gradients are generated from the mean and cross-correlation of honest gradients (Sec. I.1), making them much more aligned with honest updates (so that the term (35) becomes close to zero) than in label-flipping or inverse-gradient attacks. As a consequence, even the no-defense baseline performs relatively well in this setting. Moreover, under this type of attack, FedLAW achieves strong performance compared to other robust FL methods.

In summary, when the attacker uses misaligned gradients, the alignment term amplifies evidence against it and suppresses its weight; when the attacker aligns with honest gradients, the attack itself becomes too weak to cause substantial damage. The objective is thus structured so that what is "good" for the optimizer (low marginal score) coincides with what is "good" for the global model.

### LAYER 3: THE DETERMINISTIC ENFORCER – THE SPARSE SIMPLEX

Ultimately, the optimizer's decision is not heuristic. It solves a constrained optimization problem by selecting the $s$ clients with the highest scores. Our analysis proves that under any effective attack, the score of a Byzantine client is mathematically driven to be lower than its honest peers, either immediately (for data poisoning and model attacks) or within a few iterations (for strategic mimicry attacks). The projection onto the sparse simplex is the deterministic mechanism that enforces this ranking, pruning the low-scoring clients and ensuring the system's integrity.

We complement this analysis with an empirical study of a mimic–misaligned model attack in App. I.8 (Fig. 9), where we observe that strategic under-reporting of the loss only delays, but does not prevent, the collapse of malicious weights to (near) zero.

# D    PROJECTION ONTO THE UNIT-CAPPED SIMPLEX

The purpose here is to solve the following optimization problem:

$$\mathbf{x}_{\Delta_t^+} = \mathcal{P}_{\Delta_t^+}(\mathbf{y}) = \underset{\mathbf{x} \in \Delta_t^+}{\operatorname{argmin}} \frac{1}{2} \|\mathbf{x} - \mathbf{y}\|_2^2 \tag{36}$$

where $\Delta_t^+ = \{\mathbf{w} \in \mathbb{R}^n \mid \sum_{i=1}^n w_i = 1, w_i \geq 0, w_i \leq t\}$. In Wang & Lu (2015), using Karush–Kuhn–Tucker (KKT) conditions, a solution for the above projection is determined which is formulated in Algorithm 3.

---

**Algorithm 3** Projection onto the unit capped simplex

---

**Require:** $\mathbf{y} \in \mathbb{R}^n$ is sorted in ascending order: $y_1 \leq y_2 \leq \cdots \leq y_n$
 1: Set $y_0 = -\infty$ and $y_{n+1} = \infty$
 2: Compute partial sums $T_0 = 0$ and $T_k = \frac{1}{t} \sum_{j=1}^k y_j$ for $k = 1, \ldots, n$
 3: **for** $a = 0, 1, \ldots, n$ **do**
 4:     **if** $(\frac{1}{t} == n - a)$ **and** $(y_{a+1} - y_a \geq t)$ **then**
 5:         Set $b = a$
 6:         **break**
 7:     **end if**
 8:     **for** $b = a + 1, \ldots, n$ **do**
 9:         Compute $\gamma = \frac{\frac{1}{t} + b - n + T_a - T_b}{b - a}$
10:         **if** $(\frac{y_a}{t} + \gamma \leq 0)$ **and** $(\frac{y_{a+1}}{t} + \gamma > 0)$ **and** $(\frac{y_b}{t} + \gamma < 1)$ **and** $(\frac{y_{b+1}}{t} \geq 1)$ **then**
11:             **break**
12:         **end if**
13:     **end for**
14: **end for**
15: **Output:** $\mathbf{x} = [0, \ldots, 0, y_{a+1} + t\gamma, \ldots, y_b + t\gamma, t, \ldots, t]$

---

For more details on this algorithm, see Wang & Lu (2015).

# E    PROOF OF THEOREM 1

To prove Theorem 1, we draw inspiration from the analysis used in the proof of Theorem 2 in Kyrillidis et al. (2013). However, it is crucial to highlight a key difference: while Kyrillidis et al. (2013) establishes convergence for the sparse projection onto the unit simplex, our goal is to demonstrate the convergence of the proposed method for the sparse unit-capped simplex, which introduces additional constraints.

To prove Theorem 1, we use the following two steps.

STEP 1: THE $s$-LARGEST ELEMENTS SHOULD BE IN THE SOLUTION

Let $\mathbf{w}$ be an optimal projection of $\mathbf{h}_k$ onto $\Delta_{t,\ell_0}^+$, and assume that there exists an index $i$ among the *s-largest* entries of $\mathbf{h}_k$ such that $w_i = 0$. Suppose also that there exists an index $j \notin \operatorname{supp}(P_{L_s}(\mathbf{h}_k))$ where $w_j > 0$. Define a new vector $\tilde{\mathbf{w}}$ by setting $\tilde{w}_j = 0$ and $\tilde{w}_i = w_j$, while keeping all other entries unchanged. Consequently

$$\|\mathbf{w} - \mathbf{h}_k\|_2^2 = \|\tilde{\mathbf{w}} - \mathbf{h}_k\|_2^2 + 2w_j(h_k^i - h_k^j) \tag{37}$$

Since $w_j(h_k^i - h_k^j) \geq 0$, it follows that $\|\mathbf{w} - \mathbf{h}_k\|_2^2 \geq \|\tilde{\mathbf{w}} - \mathbf{h}_k\|_2^2$, contradicting the assumption that $\mathbf{w}$ is the optimal projection. Thus, the $s$-largest coordinates of $\mathbf{w}$ should be in the solution.

STEP 2: ENSURING THE SIMPLEX CONSTRAINT IS SATISFIED

Once the support set $\mathcal{S}^*$ (of size $s$) in (10) is determined, the optimal solution is obtained by projecting $\mathbf{w}_{\mathcal{S}^*}$ onto the unit-capped simplex. For the projection we utilized the proposed method in Wang &

Lu (2015) and established in Algorithm 3. Since the proposed method used the KKT conditions for this projection, using (10) and (36), we have

$$\|\mathbf{w}_{k+1_{\mathcal{S}^*}} - \mathbf{h}_{\lambda_{\mathcal{S}^*}}\|_2 \le \|\tilde{\mathbf{w}} - \mathbf{h}_{\lambda_{\mathcal{S}^*}}\|_2, \forall \tilde{\mathbf{w}} \in \Delta_t^+ \tag{38}$$

This guarantees that $\mathbf{w}_{k+1_{\mathcal{S}^*}}$ is the optimal projection of $\mathbf{h}_{\lambda_{\mathcal{S}^*}}$ onto $\Delta_t^+$.

At the final step we show that solutions with support $|\mathcal{S}| = s$ are as good as any other solutions with $|\mathcal{S}| < s$. Suppose there exists a solution $\mathbf{w}$ with support $|\mathcal{S}| < s$. Consider extending $|\mathcal{S}|$ to a set $|\mathcal{S}'| = s$ by adding any elements and its protection onto unit-capped simplex results in $\mathbf{w}_2$. Since according to (38), the new solution satisfies $\|\mathbf{w}'_{\mathcal{S}'} - \mathbf{h}_{\lambda_{\mathcal{S}'}}\|_2 \le \|\mathbf{w}_{\mathcal{S}} - \mathbf{h}_{\lambda_{\mathcal{S}}}\|_2$. This results in $\|\mathbf{w}' - \mathbf{h}_\lambda\|_2 \le \|\mathbf{w} - \mathbf{h}_\lambda\|_2$.

## F    PROOF OF THEOREM 2

In this section, we present the proof of Theorem 2.

To simplify the notation, we omit the iteration index $k$ throughout this proof, as the analysis holds for any arbitrary round $k$. We thus denote the true gradient of client $i$ as $\mathbf{v}_i := \nabla_\theta f_i(\boldsymbol{\theta}_k)$ (instead of $\mathbf{v}_{k,i}$) and its mini-batch estimate as $\tilde{\mathbf{v}}_i$ (instead of $\tilde{\mathbf{v}}_{k,i}$).

**Proof roadmap:**    The proof has three main steps. First, we apply Taylor's theorem with exact remainder to rewrite (6) into an analytically tractable form, where gradients are mini-batch based; the stochasticity from mini-batch sampling is then controlled via Hoeffding-type concentration. Second, we introduce a full-batch aggregator $F$ (replacing mini-batch by full gradients in aggregator $\tilde{F}$) and decompose the bias $\|\mathbb{E}\{\tilde{F}\} - \mathbf{g}\|$ into a mini-batch term and a full-batch term, bounding the former by concentration. Third, for the full-batch part, we define an auxiliary selector vector $\mathbf{s}$ representing $\mathbf{g}$ and show that the bias can be bounded in terms of discrepancies $\|\mathbf{v}_i - \mathbf{b}_j\|_2$ between honest and Byzantine gradients. Bounding these discrepancies via our heterogeneity quantities and combining both pieces yields the claimed high-probability bound.

### F.1    OPTIMAL SOLUTION OF (6)

To make the objective in (6) analytically tractable, our first step is to use Taylor's theorem with an exact remainder to reformulate the cost function. For any client $i$, we can express the projected loss as:

$$f_i(\boldsymbol{\theta}_k - \alpha\mathbf{G}_k\mathbf{w}) = f_i(\boldsymbol{\theta}_k) + \langle\nabla f_i(\boldsymbol{\theta}_k), -\alpha\mathbf{G}_k\mathbf{w}\rangle + R_i(\mathbf{w}) \tag{39}$$

where the remainder term $R_i(\mathbf{w})$ is bounded by:

$$|R_i(\mathbf{w})| \le \frac{L_{\max}}{2}\|\alpha\mathbf{G}_k\mathbf{w}\|^2 = \frac{L_{\max}}{2}\alpha^2\|\mathbf{G}_k\mathbf{w}\|^2$$

Using (39), we have

$$\sum_{i=1}^n w_i f_i(\boldsymbol{\theta}_k - \alpha\mathbf{G}_k\mathbf{w}) = \sum_{i=1}^n w_i f_i(\boldsymbol{\theta}_k) - \alpha\langle\sum_{i=1}^n w_i \nabla f_i(\boldsymbol{\theta}_k), \mathbf{G}_k\mathbf{w}\rangle + \sum_{i=1}^n w_i R_i(\mathbf{w})$$

$$= \sum_{i=1}^n w_i f_i(\boldsymbol{\theta}_k) - \alpha\mathbf{w}^T\mathbf{G}_k^T\mathbf{G}_k\mathbf{w} + \sum_{i=1}^n w_i R_i(\mathbf{w}). \tag{40}$$

Using $\mathbf{G}_k = [\mathbf{v}_1, \cdots, \mathbf{v}_n]$, in which $\mathbf{v}_i = \nabla_\theta f_i(\boldsymbol{\theta}_k)$, we have

$$\mathbf{G}_k^T\mathbf{G}_k = \begin{bmatrix} \mathbf{v}_1^T\mathbf{v}_1 & \mathbf{v}_1^T\mathbf{v}_2 & \cdots & \mathbf{v}_1^T\mathbf{v}_n \\ \mathbf{v}_1^T\mathbf{v}_2 & \mathbf{v}_2^T\mathbf{v}_2 & \cdots & \mathbf{v}_2^T\mathbf{v}_n \\ \vdots & \vdots & \cdots & \vdots \\ \mathbf{v}_1^T\mathbf{v}_n & \mathbf{v}_2^T\mathbf{v}_n & \cdots & \mathbf{v}_n^T\mathbf{v}_n \end{bmatrix}. \tag{41}$$

Substituting $\mathbf{v}_i^T\mathbf{v}_j = \frac{\|\mathbf{v}_i\|_2^2 + \|\mathbf{v}_j\|_2^2 - \|\mathbf{v}_i - \mathbf{v}_j\|_2^2}{2}$ in the above matrix yields

$$\tilde{\mathbf{G}}_k = \mathbf{G}_k^T\mathbf{G}_k = \frac{1}{2}\tilde{\mathbf{G}}_k^m - \frac{1}{2}\tilde{\mathbf{G}}_k^d, \tag{42}$$

in which $\tilde{\mathbf{G}}_k^m$ and $\tilde{\mathbf{G}}_k^d$ are given by

$$
\tilde{\mathbf{G}}_k^d = \begin{bmatrix} 0 & \|\mathbf{v}_1 - \mathbf{v}_2\|_2^2 & \cdots & \|\mathbf{v}_1 - \mathbf{v}_n\|_2^2 \\ \|\mathbf{v}_1 - \mathbf{v}_2\|_2^2 & 0 & \cdots & \|\mathbf{v}_2 - \mathbf{v}_n\|_2^2 \\ \vdots & \vdots & \cdots & \vdots \\ \|\mathbf{v}_1 - \mathbf{v}_n\|_2^2 & \|\mathbf{v}_n - \mathbf{v}_2\|_2^2 & \cdots & 0 \end{bmatrix},
$$

$$
\tilde{\mathbf{G}}_k^m = \begin{bmatrix} 2\|\mathbf{v}_1\|_2^2 & \|\mathbf{v}_1\|_2^2 + \|\mathbf{v}_2\|_2^2 & \cdots & \|\mathbf{v}_1\|_2^2 + \|\mathbf{v}_n\|_2^2 \\ \|\mathbf{v}_1\|_2^2 + \|\mathbf{v}_2\|_2^2 & 2\|\mathbf{v}_2\|_2^2 & \cdots & \|\mathbf{v}_n\|_2^2 + \|\mathbf{v}_2\|_2 \\ \vdots & \vdots & \cdots & \vdots \\ \|\mathbf{v}_1\|_2^2 + \|\mathbf{v}_n\|_2^2 & \|\mathbf{v}_2\|_2^2 + \|\mathbf{v}_n\|_2^2 & \cdots & 2\|\mathbf{v}_n\|_2^2 \end{bmatrix}. \tag{43}
$$

By defining $\mathbf{k} = [\|\mathbf{v}_1\|_2^2, \cdots, \|\mathbf{v}_n\|_2^2]^T$ and $\mathbf{1}_n = [1, \cdots, 1]^T$, the matrix $\tilde{\mathbf{G}}_k^m$ can be written by

$$
\tilde{\mathbf{G}}_k^m = \mathbf{1_n} \otimes \mathbf{k}^T + \mathbf{k} \otimes \mathbf{1_n}^T. \tag{44}
$$

Using the above representation and the property $\sum\limits_{i=1}^{n} w_i = 1$, we have

$$
\mathbf{w}^T \tilde{\mathbf{G}}_k^m \mathbf{w} = 2 \sum_{i=1}^{n} w_i \|\mathbf{v}_i\|_2^2. \tag{45}
$$

Replacing (42) in (40) and using the above equation, we obtain

$$
\sum_{i=1}^{n} w_i f_i(\boldsymbol{\theta}_k - \alpha \mathbf{G}_k \mathbf{w}) = \sum_{i=1}^{n} w_i f_i(\boldsymbol{\theta}_k) + \frac{\alpha}{2} \mathbf{w}^T \tilde{\mathbf{G}}_k^d \mathbf{w} - \frac{\alpha}{2} \mathbf{w}^T \tilde{\mathbf{G}}_k^m \mathbf{w} + \sum_{i=1}^{n} w_i R_i(\mathbf{w})
$$

$$
= \sum_{i=1}^{n} w_i f_i(\boldsymbol{\theta}_k) + \frac{\alpha}{2} \sum_{i=1}^{n} \sum_{j=1}^{n} w_i w_j \|\mathbf{v}_i - \mathbf{v}_j\|_2^2
$$

$$
- \alpha \sum_{i=1}^{n} w_i \|\mathbf{v}_i\|_2^2 + \sum_{i=1}^{n} w_i R_i(\mathbf{w})
$$

$$
= \sum_{i=1}^{n} w_i \big( f_i(\boldsymbol{\theta}_k) - \alpha \|\mathbf{v}_i\|_2^2 \big) + \frac{\alpha}{2} \sum_{i=1}^{n} \sum_{j=1}^{n} w_i w_j \|\mathbf{v}_i - \mathbf{v}_j\|_2^2
$$

$$
+ \sum_{i=1}^{n} w_i R_i(\mathbf{w}). \tag{46}
$$

Let $J(\mathbf{w}) = \sum_{i=1}^{n} w_i f_i(\boldsymbol{\theta}_k - \alpha \mathbf{G}_k \mathbf{w})$ be the true objective function and $J_{\text{approx}}(\mathbf{w}) = \sum_{i=1}^{n} w_i(f_i(\boldsymbol{\theta}_k) - \alpha\|\mathbf{v}_i\|_2^2) + \frac{\alpha}{2} \sum_{i=1}^{n} \sum_{j=1}^{n} w_i w_j \|\mathbf{v}_i - \mathbf{v}_j\|_2^2$ be the approximate objective from (40). Consequently,

$$
J(\mathbf{w}) = J_{\text{approx}}(\mathbf{w}) + \sum_{i=1}^{n} w_i R_i(\mathbf{w}) = J_{\text{approx}}(\mathbf{w}) + E_{\text{taylor}}(\mathbf{w}),
$$

where $E_{\text{taylor}}(\mathbf{w}) = \sum_{i=1}^{n} w_i R_i(\mathbf{w})$.

Substituting the cost function in (6) with the above expression, we obtain the following optimization problem

$$
\min_{\mathbf{w} \in \Delta_{t,\ell_0}^+} \sum_{i=1}^{n} w_i(f_i(\boldsymbol{\theta}_k) - \alpha\|\mathbf{v}_i\|_2^2) + \frac{\alpha}{2} \sum_{i=1}^{n} \sum_{j=1}^{n} w_i w_j \|\mathbf{v}_i - \mathbf{v}_j\|_2^2 + E_{\text{taylor}}(\mathbf{w}).
$$

To improve practicality, our analysis also focuses on the case where the aggregator relies on mini-batch gradients rather than full-batch gradients, which is a more realistic scenario. Replacing the full batch gradients and losses with mini-batch in above optimization problem results in

$$
\mathbf{w}^o = \operatorname*{argmin}_{\mathbf{w} \in \Delta_{t,\ell_0}^+} \sum_{i=1}^{n} w_i(f_{i,b}(\boldsymbol{\theta}_k) - \alpha\|\tilde{\mathbf{v}}_i\|_2^2) + \frac{\alpha}{2} \sum_{i=1}^{n} \sum_{j=1}^{n} w_i w_j \|\tilde{\mathbf{v}}_i - \tilde{\mathbf{v}}_j\|_2^2 + E_{\text{taylor}}(\mathbf{w}). \tag{47}
$$

where $f_{i,b}(\boldsymbol{\theta}_k)$ is the mini-batch loss and $\mathbb{E}_{\text{batch}}\{f_{i,b}(\boldsymbol{\theta}_k)\} = f_i(\boldsymbol{\theta}_k)$. To exclude $b_f$ clients, based on Proposition 2, it requires setting $t = \frac{1}{n-b_f}$ and $s = n - b_f$ in the constraint set $\Delta_{t,\ell_0}^+$:

$$
w_i^o = \begin{cases} \frac{1}{n-b_f}, & \text{if } i \in \Lambda_w \\[2mm] 0, & \text{if } i \in \Lambda_w^{\complement}. \end{cases} \tag{48}
$$

where $\Lambda_w = \text{supp}(\mathbf{w}^o)$ and its complement is denoted by $\Lambda_w^{\complement}$. The indices in $\Lambda_w$ corresponding to benign and Byzantine clients are denoted by $\Lambda^c$ and $\Lambda^b$, respectively.

Next, we need to check the first condition in Definition 5, in which it is required to compute $\|\mathbb{E}\{\tilde{F}\} - \mathbf{g}\|_2^2$. It is important to highlight that, unlike Definition 5, we consider a more general setting where population losses and gradients are non-iid. In this case, our function $\tilde{F}$, which is a weighted summation of the gradients with the obtained weight $\mathbf{w}^o$ from the optimization problem (47) when some gradients are replaced by their Byzantine counterparts in (47), is given by

$$
\tilde{F} = \tilde{F}(\tilde{\mathbf{v}}_1, \cdots, \tilde{\mathbf{v}}_{n-b_f}, \mathbf{b}_1, \cdots, \mathbf{b}_{b_f}) = \frac{1}{n-b_f}\Big( \sum_{i \in \Lambda^c} \tilde{\mathbf{v}}_i + \sum_{j \in \Lambda^b} \mathbf{b}_j \Big), \tag{49}
$$

where $\Lambda_w = \Lambda^c \bigcup \Lambda^b$. It is important to note that in the above equation, $\tilde{\mathbf{v}}_i$ is the mini-batch gradient of size $B$ for an honest client $i$ for $i = 1, \ldots, n - b_f$. To maintain consistency with the notation in Definition 5, we denote the honest and Byzantine gradients as $\tilde{\mathbf{v}}_i$ and $\mathbf{b}_j$, respectively. Also, the full batch aggregator, $F$, is a theoretical construct, which is given by

$$
F = F(\mathbf{v}_1, \cdots, \mathbf{v}_{n-b_f}, \mathbf{b}_1, \cdots, \mathbf{b}_{b_f}) = \frac{1}{n-b_f}\Big( \sum_{i \in \Lambda^c} \mathbf{v}_i + \sum_{j \in \Lambda^b} \mathbf{b}_j \Big), \tag{50}
$$

### F.2 HIGH-PROBABILITY BOUND VIA HOEFFDING'S INEQUALITY

This section provides a formal proof for the high-probability bound on the intra-client sampling deviation.

**Lemma 6** (High-Probability Sampling Deviation Bound). *Let $\tilde{\mathbf{v}}_i$ be the mini-batch gradient of size $B$ for an honest client $i$, and let $\mathbf{v}_i = \nabla_\theta f_i(\boldsymbol{\theta}_k)$ be the corresponding population gradient. We assume that the deviation of any single-sample gradient from its population mean is bounded:*

$$
\|\nabla_\theta f_i(\boldsymbol{\theta}_k; z) - \mathbf{v}_i\| \leq R_k.
$$

*Then, for any failure probability $\delta \in (0, 1)$, the following bound on the sampling deviation holds with probability at least $1 - \delta$:*

$$
\|\tilde{\mathbf{v}}_i - \mathbf{v}_i\| \leq \varepsilon_S, \quad \text{where} \quad \varepsilon_S = \sqrt{\frac{2R_k^2 \log(2d/\delta)}{B}}.
$$

*Proof.* The proof relies on a vector version of Hoeffding's inequality. We define the zero-mean random variable $\mathbf{X}_j = \nabla_\theta f_i(\boldsymbol{\theta}_k; z_j) - \mathbf{v}_i$. It is straightforward to show that $\mathbb{E}\{\mathbf{X}_j\} = \mathbf{0}$. The sampling deviation is the average of $B$ such independent random variables:

$$
\tilde{\mathbf{v}}_i - \mathbf{v}_i = \frac{1}{B}\sum_{j=1}^B \mathbf{X}_j.
$$

Based on the bounded deviation assumption, we know $\|\nabla f_i(\boldsymbol{\theta}_k; z) - \mathbf{v}_i\| \leq R_k$. Applying the Hoeffding's inequality for the average of $B$ independent, zero-mean random vectors $\mathbf{X}_j \in \mathbb{R}^d$, where $\|\mathbf{X}_j\|_2 \leq R_k$ for all $j$, states that for any $t > 0$:

$$
\mathbb{P}\left( \left\| \frac{1}{B}\sum_{j=1}^B \mathbf{X}_j \right\| \geq t \right) \leq 2d \cdot \exp\left( -\frac{Bt^2}{2R_k^2} \right).
$$

In our case, the average is the sampling deviation, $\|\tilde{\mathbf{v}}_i - \mathbf{v}_i\|$. Substituting this into the general form gives:

$$\mathbb{P}\left(\|\tilde{\mathbf{v}}_i - \mathbf{v}_i\| \geq t\right) \leq 2d \cdot \exp\left(-\frac{Bt^2}{2R_k^2}\right).$$

We set the failure probability to be at most $\delta$ and solve for the bound $t = \varepsilon_S$:

$$\delta \geq 2d \cdot \exp\left(-\frac{B\varepsilon_S^2}{2R_k^2}\right).$$

Solving this inequality for $\varepsilon_S$:

$$\frac{\delta}{2d} \geq \exp\left(-\frac{B\varepsilon_S^2}{2R_k^2}\right)$$

$$\log\left(\frac{2d}{\delta}\right) \leq \frac{B\varepsilon_S^2}{2R_k^2}$$

$$\frac{2R_k^2 \log(2d/\delta)}{B} \leq \varepsilon_S^2.$$

This shows that if we choose $\varepsilon_S = \sqrt{\frac{2R_k^2 \log(2d/\delta)}{B}}$, the probability of the deviation exceeding this value is at most $\delta$. Therefore, the bound holds with probability at least $1 - \delta$. □

### F.3 DECOMPOSING THE ERROR WITH THE TRIANGLE INEQUALITY

We can relate the practical error to the full batch error by adding and subtracting the full batch aggregator, $F$ (defined in (50)):

$$\|\mathbb{E}\{\tilde{F}\} - \mathbf{g}\| = \|\mathbb{E}\{\tilde{F} - F + F\} - \mathbf{g}\|$$
$$= \|(\mathbb{E}\{\tilde{F}\} - \mathbb{E}\{F\}) + (\mathbb{E}\{F\} - \mathbf{g})\|$$
$$\leq \underbrace{\|\mathbb{E}\{\tilde{F} - F\}\|}_{\text{Perturbation Error}} + \underbrace{\|\mathbb{E}\{F\} - \mathbf{g}\|}_{\text{Ideal Heterogeneity Error}}.$$

We now bound each of these two terms separately.

**Bounding the Perturbation Error:** The first term, $\|\mathbb{E}\{\tilde{F} - F\}\|$, represents how much the mini-batch noise perturbs the output of the aggregator. Consequently,

$$\tilde{F} - F = \frac{1}{n - b_f} \sum_{i \in \Lambda^c} (\tilde{\mathbf{v}}_i - \mathbf{v}_i).$$

With high probability $1 - \delta$, using Lemma 6 we have $\|\tilde{\mathbf{v}}_i - \mathbf{v}_i\| \leq \varepsilon_S$. This leads to a bound on the perturbation:

$$\|\tilde{F} - F\| \leq \frac{|\Lambda^c|}{n - b_f} \varepsilon_S \leq \varepsilon_S,$$

Taking the expectation:

$$\|\mathbb{E}\{\tilde{F} - F\}\| \leq \mathbb{E}\{\|\tilde{F} - F\|\} \leq \varepsilon_S. \tag{51}$$

**Bounding the Ideal Error:** To compute the error $\|\mathbb{E}\{F\} - \mathbf{g}\|$, we first define the vector $\mathbf{s}$ as follows

$$\mathbf{s} = \frac{1}{n - b_f} \sum_{i \in \Lambda^c} \mathbf{v}_i + \frac{1}{n - b_f} \sum_{j=1}^{|\Lambda^b|} \mathbf{e}_j \tag{52}$$

in which $\mathbf{e}_j$ is the $j$-elements in the set $\mathcal{E}$, which is constructed as follows:

1. If $|\Lambda^c| \geq |\Lambda^b|$, $\mathcal{E}$ is constructed by a random selection of $|\Lambda^b|$ elements from the set $\bar{\mathcal{V}} = \{\mathbf{v}_j, \forall j \in \Lambda^c\}$.

2. If $|\Lambda^c| < |\Lambda^b|$, $\mathcal{E}$ is constructed as

$$\mathcal{E} = \left\{ \underbrace{\bar{\mathcal{V}}, \ldots, \bar{\mathcal{V}}}_{z \text{ times}}, \mathcal{P} \right\} \tag{53}$$

in which $z = \lfloor \frac{|\Lambda^b|}{|\Lambda^c|} \rfloor$ and $\mathcal{P}$ is constructed as a random selection of $|\Lambda^b| - z|\Lambda^c|$ entries from $\bar{\mathcal{V}}$.

Taking the expectation from both sides of (52) results in

$$\mathbb{E}\{\mathbf{s}\} = \frac{1}{n - b_f} \sum_{i \in \Lambda^c} \mathbb{E}\{\mathbf{v}_i\} + \frac{1}{n - b_f} \sum_{j=1}^{|\Lambda^b|} \mathbb{E}\{\mathbf{e}_j\} + \frac{1}{n - b_f} \sum_{i \in \Lambda^d} \mathbb{E}\{\mathbf{v}_i\} - \frac{1}{n - b_f} \sum_{i \in \Lambda^d} \mathbb{E}\{\mathbf{v}_i\} \tag{54}$$

in which $\Lambda^d = \{1, 2, \ldots, n - b_f\} \setminus \Lambda^c$. Based on the definition of the set $\mathcal{E}$, using $\mathbb{E}\{\mathbf{v}_i\} = \mathbf{g}_i$, $|\Lambda^c| + |\Lambda^b| = n - b_f$, and $\mathbf{g} = \frac{1}{n-b} \sum_{j=1}^{n-b_f} \mathbf{g}_j$ we have

$$\mathbb{E}\{\mathbf{s}\} = \frac{1}{n - b_f} \sum_{i=1}^{n-b_f} \mathbf{g}_i + \frac{1}{n - b_f} \sum_{j=1}^{|\Lambda^b|} \mathbb{E}\{\mathbf{e}_j\} - \frac{1}{n - b_f} \sum_{i \in \Lambda^d} \mathbb{E}\{\mathbf{v}_i\}$$

$$= \mathbf{g} + \frac{1}{n - b_f} \sum_{j=1}^{|\Lambda^b|} \mathbf{g}_{\text{map}(j)} - \frac{1}{n - b_f} \sum_{i \in \Lambda^d} \mathbf{g}_i. \tag{55}$$

in which each vector $\mathbf{e}_j$ is mapped to one of the vectors $\mathbf{v}_i$ for all $1 \le i \le n - b_f$, i.e. $\mathbf{e}_j = \mathbf{v}_{\text{map(j)}}$.

Now, we compute $\|\mathbb{E}\{F\} - \mathbf{g}\|_2^2$. Since $\mathbb{E}\{\mathbf{s}\} + \frac{1}{n-b_f} \sum_{i \in \Lambda^d} \mathbf{g}_i - \frac{1}{n-b_f} \sum_{j=1}^{|\Lambda^b|} \mathbf{g}_{\text{map(j)}} = \mathbf{g}$,

$$\|\mathbb{E}\{F\} - \mathbf{g}\|_2 = \|\mathbb{E}\{F\} - \mathbb{E}\{\mathbf{s}\} - \frac{1}{n - b_f} \sum_{i \in \Lambda^d} \mathbf{g}_i + \frac{1}{n - b_f} \sum_{j=1}^{|\Lambda^b|} \mathbf{g}_{\text{map(j)}}\|_2 \le$$

$$\|\mathbb{E}\{F - \mathbf{s}\}\|_2 + \frac{1}{n - b_f} \sum_{i \in \Lambda^d} \|\mathbf{g}_i - \mathbf{g}\|_2 + \frac{1}{n - b_f} \sum_{j=1}^{|\Lambda^b|} \|\mathbf{g}_{\text{map(j)}} - \mathbf{g}\|_2 \tag{56}$$

Since $\frac{1}{n-b_f} \sum_{i=1}^{n-b_f} \|\mathbf{g}_i - \mathbf{g}\|_2^2 \le H_k^2$, using Cauchy-Schwarz inequality $\frac{1}{n-b_f} \sum_{i \in \Lambda^d} \|\mathbf{g}_i - \mathbf{g}\|_2 \le \frac{1}{n-b_f} \sqrt{\sum_{i \in \Lambda^d} \|\mathbf{g}_i - \mathbf{g}\|_2^2} \sqrt{|\Lambda_b|} \le \frac{b_f}{\sqrt{n-b_f}} H_k$, we have

$$\|\mathbb{E}\{F\} - \mathbf{g}\|_2 \le \|\mathbb{E}\{F - \mathbf{s}\}\|_2 + \frac{2b_f}{\sqrt{n - b_f}} H_k \tag{57}$$

Substituting $F$ and $\mathbf{s}$ in the above inequality results in

$$\|\mathbb{E}\{F - s\}\|_2^2 = \|\mathbb{E}\left\{ \frac{1}{n - b_f} \left( \sum_{i \in \Lambda^c} \mathbf{v}_i + \sum_{j \in \Lambda^b} \mathbf{b}_j - \sum_{i \in \Lambda^c} \mathbf{v}_i - \sum_{j=1}^{|\Lambda^b|} \mathbf{e}_j \right) \right\}\|_2^2$$

$$= \|\mathbb{E}\left\{ \frac{1}{n - b_f} \left( \sum_{j \in \Lambda^b} \mathbf{b}_j - \sum_{j=1}^{|\Lambda^b|} \mathbf{e}_j \right) \right\}\|_2^2. \tag{58}$$

Using Jensen inequality, we have

$$\|\mathbb{E}\{F - s\}\|_2^2 \le \mathbb{E}\left\{ \frac{1}{(n - b_f)^2} \left\| \left( \sum_{j \in \Lambda^b} \mathbf{b}_j - \sum_{j=1}^{|\Lambda^b|} \mathbf{e}_j \right) \right\|_2^2 \right\}$$

$$= \frac{1}{(n - b_f)^2} \mathbb{E}\left\{ \left\| \left( \sum_{j=1}^{|\Lambda^b|} \mathbf{b}_{\Lambda^b(j)} - \mathbf{e}_j \right) \right\|_2^2 \right\}$$

$$\le \frac{|\Lambda^b|}{(n - b_f)^2} \mathbb{E}\left\{ \sum_{j \in \Lambda^b} \left\| \mathbf{b}_{\Lambda^b(j)} - \mathbf{e}_j \right\|_2^2 \right\} \tag{59}$$

Based on the definition of the set $\mathcal{E}$, $\mathbf{e}_j$ is one of the vectors in the set $\bar{\mathcal{V}}$ which includes all vectors $\mathbf{v}_j$ for all $j \in \Lambda^c$. Consequently,

$$\|\mathbb{E}\{F - s\}\|_2^2 \leq \frac{|\Lambda^b|}{(n - b_f)^2} \mathbb{E}\Big\{ \sum_{j \in \Lambda^b} \Big\| \mathbf{b}_{\Lambda^b(j)} - \mathbf{e}_j \Big\|_2^2 \Big\} \leq \frac{|\Lambda^b|}{(n - b_f)^2} \mathbb{E}\Big\{ \sum_{i \in \Lambda^c} \sum_{j \in \Lambda^b} \|\mathbf{v}_i - \mathbf{b}_j\|_2^2 \Big\} =$$
$$\frac{|\Lambda^b|}{(n - b_f)^2} \sum_{i \in \Lambda^c} \sum_{j \in \Lambda^b} \mathbb{E}\Big\{ \|\mathbf{v}_i - \mathbf{b}_j\|_2^2 \Big\}. \tag{60}$$

Thus, to bound the squared error of the aggregation rule, $\|\mathbb{E}\{F - s\}\|_2^2$, we need to compute an upper bound for $\mathbb{E}\Big\{ \|\mathbf{v}_i - \mathbf{b}_j\|_2^2 \Big\}$. The remainder of the proof addresses this step.

**Proof of Theorem 2** : It is crucial to note that, if some mini-batch gradients $\tilde{\mathbf{v}}_i$ are replaced by their Byzantine counterparts $\mathbf{b}_i$, the optimization problem (47) involves a mixture of both Byzantine and mini-batch gradients. Furthermore, under the assumption stated in the above lemma, the aggregator has access to all honest mini-batch losses $f_{i,b}(\boldsymbol{\theta}_k)$. Without loss of generality, we assume that the last $b_f$ gradient vectors have been replaced with their Byzantine versions. As a result, the gradient matrix can be expressed as:

$$\mathbf{G}_k = [\tilde{\mathbf{v}}_1, \cdots, \tilde{\mathbf{v}}_{n-b_f}, \mathbf{b}_1, \cdots, \mathbf{b}_{b_f}]. \tag{61}$$

In the above equation, we denote the honest mini-batch and Byzantine gradients by $\tilde{\mathbf{v}}_i$ and $\mathbf{b}_j$, respectively, for $i = 1, \ldots, n - b_f$. From (49), and noting that $\mathbf{w}^o$ is the minimizer of (47), we obtain

$$\sum_{i=1}^{n} w_i^o f_{i,b}(\boldsymbol{\theta}_k) - \alpha \sum_{i \in \Lambda^c} w_i^o \|\tilde{\mathbf{v}}_i\|_2^2 - \alpha \sum_{i \in \Lambda^b} w_i^o \|\mathbf{b}_i\|_2^2$$
$$+ \frac{\alpha}{2} \Big( \sum_{i \in \Lambda^c} \sum_{j \in \Lambda^b} w_i^o w_j^o \|\tilde{\mathbf{v}}_i - \mathbf{b}_j\|_2^2 + \sum_{i \in \Lambda^c} \sum_{j \in \Lambda^c} w_i^o w_j^o \|\tilde{\mathbf{v}}_i - \tilde{\mathbf{v}}_j\|_2^2$$
$$+ \sum_{i \in \Lambda^b} \sum_{j \in \Lambda^b} w_i^o w_j^o \|\mathbf{b}_i - \mathbf{b}_j\|_2^2 \Big) + E_{\text{taylor}}(\mathbf{w}^o)$$
$$\leq \sum_{i=1}^{n} w_i^t f_{i,b}(\boldsymbol{\theta}_k) - \alpha \sum_{i=1}^{n-b_f} w_i^t \|\tilde{\mathbf{v}}_i\|_2^2$$
$$+ \frac{\alpha}{2} \sum_{i=1}^{n-b_f} \sum_{j=1}^{n-b_f} w_i^t w_j^t \|\tilde{\mathbf{v}}_i - \tilde{\mathbf{v}}_j\|_2^2 + E_{\text{taylor}}(\mathbf{w}^t) \tag{62}$$

in which $\mathbf{w}^t$ is a feasible set of weights given by

$$w_i^t = \begin{cases} \frac{1}{n - b_f}, & \text{if } 1 \leq i \leq n - b_f \\ 0, & \text{if } n - b_f + 1 \leq i \leq n. \end{cases} \tag{63}$$

Based on the inequality (62), we have

$$\frac{\alpha}{2} \sum_{i \in \Lambda^c} \sum_{j \in \Lambda^b} w_i^o w_j^o \|\tilde{\mathbf{v}}_i - \mathbf{b}_j\|_2^2$$
$$\leq \sum_{i=1}^{n} w_i^t f_{i,b}(\boldsymbol{\theta}_k) - \sum_{i=1}^{n} w_i^o f_{i,b}(\boldsymbol{\theta}_k) + \alpha \sum_{i \in \Lambda^c} w_i^o \|\tilde{\mathbf{v}}_i\|_2^2 + \alpha \sum_{i \in \Lambda^b} w_i^o \|\mathbf{b}_i\|_2^2 - \alpha \sum_{i=1}^{n-b_f} w_i^t \|\tilde{\mathbf{v}}_i\|_2^2$$
$$+ \frac{\alpha}{2} \sum_{i=1}^{n-b_f} \sum_{j=1}^{n-b_f} w_i^t w_j^t \|\tilde{\mathbf{v}}_i - \tilde{\mathbf{v}}_j\|_2^2 + E_{\text{taylor}}(\mathbf{w}^t) - E_{\text{taylor}}(\mathbf{w}^o). \tag{64}$$

Replacing $w_i^o$ and $w_j^t$ based on (48) and (63) in the above inequality results in

$$\frac{\alpha}{2(n-b_f)^2} \sum_{i\in\Lambda^c} \sum_{j\in\Lambda^b} \|\tilde{\mathbf{v}}_i - \mathbf{b}_j\|_2^2$$

$$\leq \frac{1}{n-b_f}\Big(\sum_{i\in\Lambda^d} f_{i,b}(\boldsymbol{\theta}_k) - \sum_{j\in\Lambda^b} f_{j,b}(\boldsymbol{\theta}_k)\Big) + \frac{\alpha}{n-b_f}\sum_{i\in\Lambda^b}\|\mathbf{b}_i\|_2^2 - \frac{\alpha}{n-b_f}\sum_{i\in\Lambda^d}\|\tilde{\mathbf{v}}_i\|_2^2$$

$$+ \frac{\alpha}{2(n-b_f)^2} \sum_{i=1}^{n-b_f}\sum_{j=1}^{n-b_f} \|\tilde{\mathbf{v}}_i - \tilde{\mathbf{v}}_j\|_2^2 + E_{\text{taylor}}(\mathbf{w}^t) - E_{\text{taylor}}(\mathbf{w}^o). \tag{65}$$

in which $\Lambda^d = \{1,2,\ldots,n-b_f\}\setminus\Lambda^c$, implying $|\Lambda^d| = n - b_f - |\Lambda^c| = |\Lambda^b|$. Using the assumption $\|\mathbf{b}_j\| \leq \max_{l\in\mathcal{H}}\|\tilde{\mathbf{v}}_l\|$, we have

$$\frac{\alpha}{2(n-b_f)^2} \sum_{i\in\Lambda^c} \sum_{j\in\Lambda^b} \|\tilde{\mathbf{v}}_i - \mathbf{b}_j\|_2^2$$

$$\leq \frac{1}{n-b_f}\Big(\sum_{i\in\Lambda^d} f_{i,b}(\boldsymbol{\theta}_k) - \sum_{j\in\Lambda^b} f_{j,b}(\boldsymbol{\theta}_k)\Big) + \frac{\alpha b_f}{n-b_f}\|\tilde{\mathbf{v}}_{\text{map}(k)}\|_2^2 - \frac{\alpha}{n-b_f}\sum_{i\in\Lambda^d}\|\tilde{\mathbf{v}}_i\|_2^2$$

$$+ \frac{\alpha}{2(n-b_f)^2} \sum_{i=1}^{n-b_f}\sum_{j=1}^{n-b_f} \|\tilde{\mathbf{v}}_i - \tilde{\mathbf{v}}_j\|_2^2 + E_{\text{taylor}}(\mathbf{w}^t) - E_{\text{taylor}}(\mathbf{w}^o). \tag{66}$$

where map(k) that was selected by the optimizer.

FROM MINI-BATCH GRADIENT TO FULL-BATCH GRADIENTS

In (60), we deal with the full batch gradients $\mathbf{v}_i$ while in the above equation we have mini-batch gradient $\tilde{\mathbf{v}}_i$, so we need to formulate the left-hand side of the inequality (66) based on full batch gradients. Since the $\mathbf{v}_i$ is unbiased estimator of mini-batch gradient $\tilde{\mathbf{v}}_i$ over batches, i.e. $\mathbb{E}_{\text{batch}}\{\tilde{\mathbf{v}}_i\} = \mathbf{v}_i$ (see Subsection 2.2). So, we have

$$\mathbb{E}_{\text{batch}}\{\|\tilde{\mathbf{v}}_i - b_j\|_2^2\} = \mathbb{E}_{\text{batch}}\{\|(\tilde{\mathbf{v}}_i - \mathbf{v}_i) + (\mathbf{v}_i - b_j)\|_2^2\}$$

$$= \mathbb{E}_{\text{batch}}\{\|\tilde{\mathbf{v}}_i - \mathbf{v}_i\|_2^2 + 2\langle\tilde{\mathbf{v}}_i - \mathbf{v}_i, \mathbf{v}_i - b_j\rangle + \|\mathbf{v}_i - b_j\|_2^2\}$$

$$= \mathbb{E}_{\text{batch}}\{\|\tilde{\mathbf{v}}_i - \mathbf{v}_i\|_2^2\} + 2\langle\mathbb{E}_{\text{batch}}\{\tilde{\mathbf{v}}_i - \mathbf{v}_i\}, \mathbf{v}_i - b_j\rangle + \|\mathbf{v}_i - b_j\|_2^2$$

$$= \mathbb{E}_{\text{batch}}\{\|\tilde{\mathbf{v}}_i - \mathbf{v}_i\|_2^2\} + 2\langle\mathbf{v}_i - \mathbf{v}_i, \mathbf{v}_i - b_j\rangle + \|\mathbf{v}_i - b_j\|_2^2$$

$$= \mathbb{E}_{\text{batch}}\{\|\tilde{\mathbf{v}}_i - \mathbf{v}_i\|_2^2\} + 0 + \|\mathbf{v}_i - b_j\|_2^2$$

$$= \|\mathbf{v}_i - b_j\|_2^2 + \mathbb{E}_{\text{batch}}\{\|\tilde{\mathbf{v}}_i - \mathbf{v}_i\|_2^2\}.$$

Based on the above equation and Lemma 6, we have

$$\|\mathbf{v}_i - b_j\|_2^2 \leq \mathbb{E}_{\text{batch}}\{\|\tilde{\mathbf{v}}_i - b_j\|_2^2\} \leq \|\mathbf{v}_i - b_j\|_2^2 + \varepsilon_S^2 \tag{67}$$

Similarly, we can write

$$\mathbb{E}_{\text{batch}}\{\|\tilde{\mathbf{v}}_i - \tilde{\mathbf{v}}_j\|_2^2\} = \mathbb{E}_{\text{batch}}\{\|(\mathbf{v}_i - \mathbf{v}_j) + (\tilde{\mathbf{v}}_i - \mathbf{v}_i) - (\tilde{\mathbf{v}}_j - \mathbf{v}_j)\|_2^2\}$$

$$= \|\mathbf{v}_i - \mathbf{v}_j\|_2^2 + \mathbb{E}_{\text{batch}}\{\|\tilde{\mathbf{v}}_i - \mathbf{v}_i\|_2^2\} + \mathbb{E}_{\text{batch}}\{\|\tilde{\mathbf{v}}_j - \mathbf{v}_j\|_2^2\}$$

$$- 2\mathbb{E}_{\text{batch}}\{\langle\tilde{\mathbf{v}}_i - \mathbf{v}_i, \tilde{\mathbf{v}}_j - \mathbf{v}_j\rangle\}. \tag{68}$$

Since mini-batch samples for different clients are independent:

$$\mathbb{E}_{\text{batch}}\{\langle\tilde{\mathbf{v}}_i - \mathbf{v}_i, \tilde{\mathbf{v}}_j - \mathbf{v}_j\rangle\} = \langle\mathbb{E}_{\text{batch}}\{\tilde{\mathbf{v}}_i - \mathbf{v}_i\}, \mathbb{E}_{\text{batch}}\{\tilde{\mathbf{v}}_j - \mathbf{v}_j\}\rangle = 0.$$

Using the results of Lemma 6 and above equation in (68) results in:

$$\mathbb{E}_{\text{batch}}\{\|\tilde{\mathbf{v}}_i - \tilde{\mathbf{v}}_j\|_2^2\} \leq \|\mathbf{v}_i - \mathbf{v}_j\|_2^2 + 2\varepsilon_S^2. \tag{69}$$

Similarly, we have

$$\mathbb{E}_{\text{batch}}\{\|\tilde{\mathbf{v}}_i\|_2^2\} = \mathbb{E}_{\text{batch}}\{\|\mathbf{v}_i + (\tilde{\mathbf{v}}_i - \mathbf{v}_i)\|_2^2\}$$

$$= \|\mathbf{v}_i\|_2^2 + 2\langle\mathbf{v}_i, \mathbb{E}_{\text{batch}}\{\tilde{\mathbf{v}}_i - \mathbf{v}_i\}\rangle + \mathbb{E}_{\text{batch}}\{\|\tilde{\mathbf{v}}_i - \mathbf{v}_i\|_2^2\}$$

Based on the above equation and using Lemma 6, we have

$$\|\mathbf{v}_i\|_2^2 \le \mathbb{E}_{\text{batch}}\{\|\tilde{\mathbf{v}}_i\|_2^2\} \le \|\mathbf{v}_i\|_2^2 + \varepsilon_S^2 \tag{70}$$

Taking the expectation over batches in (66), $\mathbb{E}_{\text{batch}}\{f_{i,b}(\boldsymbol{\theta}_k)\} = f_i(\boldsymbol{\theta}_k)$ and replacing the expectation with the results (67), (69) and (70) yields

$$\frac{\alpha}{2(n-b_f)^2} \sum_{i\in\Lambda^c}\sum_{j\in\Lambda^b} \|\mathbf{v}_i - \mathbf{b}_j\|_2^2$$

$$\le \frac{1}{n-b_f}\Big(\sum_{i\in\Lambda^d} f_i(\boldsymbol{\theta}_k) - \sum_{j\in\Lambda^b} f_j(\boldsymbol{\theta}_k)\Big) + \frac{\alpha b_f}{n-b_f}\mathbb{E}_{\text{batch}}\{\|\tilde{\mathbf{v}}_{\text{map}(k)}\|_2^2\} - \frac{\alpha}{n-b_f}\sum_{i\in\Lambda^d}\|\mathbf{v}_i\|_2^2$$

$$+ \frac{\alpha}{2(n-b_f)^2}\sum_{i=1}^{n-b_f}\sum_{j=1}^{n-b_f} \|\mathbf{v}_i - \mathbf{v}_j\|_2^2 + \alpha\varepsilon_S^2 + \mathbb{E}_{\text{batch}}\{E_{\text{taylor}}(\mathbf{w}^t) - E_{\text{taylor}}(\mathbf{w}^o)\}. \tag{71}$$

BOUNDING THE LOSS HETEROGENEITY TERM

The first term on the right-hand side of the above inequality captures the effect of non-IID losses. We take its expectation:

$$\mathbb{E}\{\text{Loss Term}\} = \frac{1}{n-b_f}\left(\sum_{i\in\Lambda^d} m_i - \sum_{j\in\Lambda^b} m_j\right).$$

Let $m_k = m_{avg} + \delta_k$, where $\delta_k = m_k - m_{avg}$.

$$\sum_{i\in\Lambda^d} m_i - \sum_{j\in\Lambda^b} m_j = \sum_{i\in\Lambda^d}(m_{avg} + \delta_i) - \sum_{j\in\Lambda^b}(m_{avg} + \delta_j)$$

$$= (|\Lambda^d| - |\Lambda^b|)m_{avg} + \left(\sum_{i\in\Lambda^d}\delta_i - \sum_{j\in\Lambda^b}\delta_j\right).$$

Since $|\Lambda^d| = |\Lambda^b|$, the $m_{avg}$ terms cancel perfectly. We are left with bounding the sum of the deviations. Taking the absolute value:

$$\left|\sum_{i\in\Lambda^d}\delta_i - \sum_{j\in\Lambda^b}\delta_j\right| \le \sum_{i\in\Lambda^d}|\delta_i| + \sum_{j\in\Lambda^b}|\delta_j| = \sum_{k\in\Lambda^d\cup\Lambda^b}|m_k - m_{avg}|.$$

Since the set of selected and discarded clients is a subset of all honest clients, this sum is bounded by the total sum of deviations over all honest clients:

$$\sum_{k\in\Lambda^d\cup\Lambda^b}|m_k - m_{avg}| \le \sum_{k\in\mathcal{H}}|m_k - m_{avg}|.$$

Using our new assumption, $\sum_{k\in\mathcal{H}}|m_k - m_{avg}| \le (n-b_f)\varepsilon_k$. Therefore:

$$\left|\mathbb{E}\{\text{Loss Term}\}\right| = \frac{1}{n-b_f}\left|\sum_{i\in\Lambda^d}\delta_i - \sum_{j\in\Lambda^b}\delta_j\right| \le \frac{(n-b_f)\varepsilon_k}{n-b_f} = \varepsilon_k. \tag{72}$$

BOUNDING THE PAIRWISE DISTANCE TERM

First, we find an upper bound for the average expected squared distance between any two honest clients.

$$\mathbb{E}\{\|\mathbf{v}_i - \mathbf{v}_j\|_2^2\} = \mathbb{E}\{\|(\mathbf{g}_i - \mathbf{g}_j) + ((\mathbf{v}_i - \mathbf{g}_i) - (\mathbf{v}_j - \mathbf{g}_j))\|_2^2\}$$

$$= \|\mathbf{g}_i - \mathbf{g}_j\|_2^2 + \mathbb{E}\{\|(\mathbf{v}_i - \mathbf{g}_i) - (\mathbf{v}_j - \mathbf{g}_j)\|_2^2\}$$

$$= \|\mathbf{g}_i - \mathbf{g}_j\|_2^2 + \text{Var}(\mathbf{v}_i - \mathbf{v}_j)$$

$$\le \|\mathbf{g}_i - \mathbf{g}_j\|_2^2 + 4d\sigma_k^2.$$

To bound the average of $\|\mathbf{g}_i - \mathbf{g}_j\|_2^2$, we add and subtract $\mathbf{g}$ and note that $\sum_k (\mathbf{g}_k - \mathbf{g}) = 0$:

$$\frac{1}{(n-b_f)^2} \sum_{i,j \in \mathcal{H}} \|\mathbf{g}_i - \mathbf{g}_j\|_2^2 = \frac{1}{(n-b_f)^2} \sum_{i,j} \|(\mathbf{g}_i - \mathbf{g}) - (\mathbf{g}_j - \mathbf{g})\|_2^2$$

$$= \frac{2(n-b_f)}{(n-b_f)^2} \sum_{k=1}^{n-b_f} \|\mathbf{g}_k - \mathbf{g}\|_2^2 \leq \frac{2}{n-b_f} \cdot (n-b_f) H_k^2 = 2H_k^2.$$

Therefore, the average expected squared distance is bounded as:

$$\frac{1}{(n-b_f)^2} \sum_{i,j \in \mathcal{H}} \mathbb{E}\{\|\mathbf{v}_i - \mathbf{v}_j\|_2^2\} \leq 2H_k^2 + \frac{(4d\sigma_k^2)(n-b_f-1)}{n-b_f}. \tag{73}$$

BOUNDING THE NORM-DIFFERENCE TERM

This section provides a formal proof for the bound on the term $\mathcal{D} = \sum_{i \in \Lambda^b} \mathbb{E}\{\|\mathbf{b}_i\|_2^2\} - \sum_{i \in \Lambda^d} \mathbb{E}\{\|\mathbf{v}_i\|_2^2\}$, which is a key component of the main resilience proof for the practical mini-batch aggregator.

**Lemma 7** (Bound on the Mini-Batch Norm-Difference Term). *Under assumptions (1) and assume that $\|\mathbf{b}_j\| \leq \max_{l \in \mathcal{H}} \|\tilde{\mathbf{v}}_l\|$.*

*Then, with probability at least $1 - \delta$ (over the mini-batch sampling events), the norm-difference term $\mathcal{D}$ is bounded by:*

$$\mathcal{D} \leq b_f(2K_k^2 + d\sigma_k^2 + \varepsilon_S^2). \tag{74}$$

*Proof.* To prove the lemma, we begin with the definition of $\mathcal{D}$. The attacker's gradient norm is bounded by $\max_{l \in \mathcal{H}} \|\tilde{\mathbf{v}}_l\|$. Therefore, for each Byzantine client $j \in \Lambda^b$, its expected squared norm is bounded by the expected squared norm of some honest client 'map(k)' that was selected by the optimizer. This allows us to write:

$$\mathcal{D} = \sum_{j \in \Lambda^b} \mathbb{E}\{\|\mathbf{b}_j\|_2^2\} - \sum_{i \in \Lambda^d} \mathbb{E}\{\|\mathbf{v}_i\|_2^2\}$$

$$\leq \left( b_f \mathbb{E}\{\|\tilde{\mathbf{v}}_{\mathrm{map}(k)}\|_2^2\} - \sum_{i \in \Lambda^d} \mathbb{E}\{\|\mathbf{v}_i\|_2^2\} \right), \tag{75}$$

According to (71), we need to incorporate the expectation over batch for the expectation term $\mathbb{E}\{\|\tilde{\mathbf{v}}_{\mathrm{map}(k)}\|_2^2\}$ in the right hand side of the above inequality, so the expectation in this term will be jointly over batch and data distribution. The mini-batch gradient $\tilde{\mathbf{v}}_k$ is an unbiased estimate of the population gradient $\mathbf{v}_k$, meaning $\mathbb{E}_{\mathrm{batch}}\{\tilde{\mathbf{v}}_k\} = \mathbf{v}_k$. The total expectation is taken over both the mini-batch sampling and the population distribution. We have:

$$\mathbb{E}\{\|\tilde{\mathbf{v}}_k\|_2^2\} = \mathbb{E}\{\|\mathbf{v}_k + (\tilde{\mathbf{v}}_k - \mathbf{v}_k)\|_2^2\}$$

$$= \mathbb{E}\{\|\mathbf{v}_k\|_2^2 + 2\langle \mathbf{v}_k, \tilde{\mathbf{v}}_k - \mathbf{v}_k \rangle + \|\tilde{\mathbf{v}}_k - \mathbf{v}_k\|_2^2\}.$$

The cross-term vanishes under the total expectation because the inner expectation over the batch is zero: $\mathbb{E}\{\langle \mathbf{v}_k, \tilde{\mathbf{v}}_k - \mathbf{v}_k \rangle\} = \mathbb{E}\{\langle \mathbf{v}_k, \mathbb{E}_{\mathrm{batch}}\{\tilde{\mathbf{v}}_k\} - \mathbf{v}_k \rangle\} = \mathbb{E}\{\langle \mathbf{v}_k, \mathbf{v}_k - \mathbf{v}_k \rangle\} = 0$. Consequently,

$$\mathbb{E}\{\|\tilde{\mathbf{v}}_k\|_2^2\} = \mathbb{E}\{\|\mathbf{v}_k\|_2^2\} + \mathbb{E}\{\|\tilde{\mathbf{v}}_k - \mathbf{v}_k\|_2^2\}. \tag{76}$$

We now substitute (76) into (75), resulting in

$$\mathcal{D} \leq \left[ b_f \left( \mathbb{E}\{\|\mathbf{v}_{\mathrm{map}(k)}\|_2^2\} + \mathbb{E}\{\|\mathrm{dev}_{\mathrm{map}}\|_2^2\} \right) - \sum_{i \in \Lambda^d} \mathbb{E}\{\|\mathbf{v}_i\|_2^2\} \right]$$

where dev denotes the sampling deviation vector $(\tilde{\mathbf{v}} - \mathbf{v})$. We first bound the population level. Using $\mathbb{E}\{\|\mathbf{v}_k\|_2^2\} = \|\mathbf{g}_k\|_2^2 + \mathrm{Var}(\mathbf{v}_k) \leq \|\mathbf{g}_k\|_2^2 + d\sigma_k^2$:

$$\sum_{i \in \Lambda^d} \left( \mathbb{E}\{\|\mathbf{v}_{\mathrm{map}(k)}\|_2^2\} - \mathbb{E}\{\|\mathbf{v}_k\|_2^2\} \right) \leq \sum_{i \in \Lambda^d} \left( (\|\mathbf{g}_{\mathrm{map}(k)}\|_2^2 + d\sigma_k^2) - \|\mathbf{g}_k\|_2^2 \right)$$

$$= \sum_{i \in \Lambda^d} \left( \|\mathbf{g}_{\mathrm{map}(k)}\|_2^2 - \|\mathbf{g}_k\|_2^2 \right) + b_f d\sigma_k^2.$$

We now apply the Bounded Norm Deviation assumption $(K^2)$ to the difference of squared norms:

$$\left| \|\mathbf{g}_{\mathrm{map(k)}}\|_2^2 - \|\mathbf{g}_i\|_2^2 \right| = \left| (\|\mathbf{g}_{\mathrm{map(k)}}\|_2^2 - \|\mathbf{g}\|_2^2) - (\|\mathbf{g}_i\|_2^2 - \|\mathbf{g}\|_2^2) \right|$$

$$\leq \left| \|\mathbf{g}_{\mathrm{map(k)}}\|_2^2 - \|\mathbf{g}\|_2^2 \right| + \left| \|\mathbf{g}_i\|_2^2 - \|\mathbf{g}\|_2^2 \right| \leq 2K_k^2.$$

This gives the final, $\|\mathbf{g}\|_2$-independent bound for the norm-difference term:

$$\sum_{k=1}^{b_f} \left( \mathbb{E}\{\|\mathbf{v}_{\mathrm{map}(k)}\|_2^2\} - \mathbb{E}\{\|\mathbf{v}_k\|_2^2\} \right) \leq \sum_{k=1}^{b_f} 2K_k^2 + b_f d\sigma_k^2 = b_f(2K_k^2 + d\sigma_k^2). \tag{77}$$

Next, we bound the sampling deviation. The high-probability bound states that $\|\mathrm{dev}\|^2 \leq \varepsilon_S^2$ for any client. To get an upper bound on the difference, we have:

$$\mathbb{E}\{\|\mathrm{dev}_{\mathrm{map}}\|_2^2\} \leq \varepsilon_S^2.$$

Summing over the $b_f$ clients, this entire term is bounded by $b_f \varepsilon_S^2$.

Finally, combining the bounds for the two separated terms, we arrive at the final result. With probability at least $1 - \delta$:

$$\mathcal{D} \leq \left( b_f(2K_k^2 + d\sigma_k^2) \right) + \left( b_f \varepsilon_S^2 \right) = b_f(2K_k^2 + d\sigma_k^2 + \varepsilon_S^2).$$

This completes the proof of Lemma 7. $\qquad\square$

BOUNDING THE TAYLOR ERROR TERM

The new term introduced by the exact analysis is $E_{\mathrm{new}} = (E_{\mathrm{taylor}}(\mathbf{w}^t) - E_{\mathrm{taylor}}(\mathbf{w}^o))$. We now bound its magnitude.

$$|E_{\mathrm{new}}| = \left| \left( \sum w_i^t R_i(\mathbf{w}^t) - \sum w_i^o R_i(\mathbf{w}^o) \right) \right|$$

$$\leq \left( \sum w_i^t |R_i(\mathbf{w}^t)| + \sum w_i^o |R_i(\mathbf{w}^o)| \right)$$

$$\leq \frac{L_{\max}}{2} \alpha^2 \left( \|\mathbf{G}_k \mathbf{w}^t\|^2 + \|\mathbf{G}_k \mathbf{w}^o\|^2 \right)$$

$$= \frac{L_{\max}\alpha^2}{2} \left( \|\mathbf{G}_k \mathbf{w}^t\|^2 + \|\mathbf{G}_k \mathbf{w}^o\|^2 \right)$$

To get a concrete bound, using the assumption $\|\mathbf{b}_i\|_2 \leq \max_{l \in \mathcal{H}} \|\tilde{\mathbf{v}}_l\|_2$, and since $\mathbf{w}$ is in the simplex, $\|\frac{1}{n-b_f} \left( \sum_{i \in \Lambda^c} \tilde{\mathbf{v}}_i + \sum_{j \in \Lambda^b} \mathbf{b}_j \right) \|^2 \leq \frac{1}{n-b_f} \left( \sum_{i \in \Lambda^c} \|\tilde{\mathbf{v}}_i\|_2^2 + \sum_{j \in \Lambda^b} \|\mathbf{b}_j\|_2^2 \right) \leq \frac{1}{n-b_f} \left( \sum_{i \in \Lambda^c} \|\tilde{\mathbf{v}}_i\|_2^2 + b_f \|\tilde{\mathbf{v}}_{\mathrm{map(k)}}\|_2^2 \right)$. Similarly, $\|\mathbf{G}_k \mathbf{w}^o\|^2 \leq \frac{1}{n-b_f} \left( \sum_{i=1}^{n-b_f} \|\tilde{\mathbf{v}}_i\|_2^2 \right)$. Consequently,

$$\mathbb{E}\{|E_{\mathrm{new}}|\} \leq \frac{L_{\max}\alpha^2}{2(n-b_f)} \left( \sum_{i=1}^{n-b_f} \mathbb{E}\{\|\tilde{\mathbf{v}}_i\|_2^2\} + \sum_{i \in \Lambda^c} \mathbb{E}\{\|\tilde{\mathbf{v}}_i\|_2^2\} + b_f \mathbb{E}\{\|\tilde{\mathbf{v}}_{\mathrm{map(k)}}\|_2^2\} \right)$$

$$\leq \frac{L_{\max}\alpha^2}{2(n-b_f)} \left( \sum_{i=1}^{n-b_f} \mathbb{E}\{\|\mathbf{v}_i\|_2^2\} + \sum_{i \in \Lambda^c} \mathbb{E}\{\|\mathbf{v}_i\|_2^2\} + b_f \mathbb{E}\{\|\mathbf{v}_{\mathrm{map(k)}}\|_2^2\} \right) + L_{\max}\alpha^2 \varepsilon_S^2 \quad \text{by using (76)}$$

$$\tag{78}$$

Rearranging and using the assumption that $\mathrm{Var}(\mathbf{v}_i) \leq d\sigma_k^2$, we get:

$$\mathbb{E}\{\|\mathbf{v}_i\|^2\} = \|\mathbf{g}_i\|^2 + \mathrm{Var}(\mathbf{v}_i) \leq \|\mathbf{g}_i\|^2 + d\sigma_k^2$$

We now sum the above result over all $n - b_f$ honest clients:

$$\sum_{i=1}^{n-b_f} \mathbb{E}\{\|\mathbf{v}_i\|^2\} \leq \sum_{i=1}^{n-b_f} (\|\mathbf{g}_i\|^2 + d\sigma_k^2) = \left( \sum_{i=1}^{n-b_f} \|\mathbf{g}_i\|^2 \right) + (n - b_f)d\sigma_k^2$$

The final piece is to bound the sum of squared population gradients using the heterogeneity constant $H$ and the mean gradient $\mathbf{g}$. We use the "add and subtract $\mathbf{g}$" trick:

$$\sum_{i=1}^{n-b_f} \|\mathbf{g}_i\|^2 = \sum_{i=1}^{n-b_f} \|\mathbf{g}_i - \mathbf{g} + \mathbf{g}\|^2$$

$$= \sum_{i=1}^{n-b_f} \left( \|\mathbf{g}_i - \mathbf{g}\|^2 + 2\langle \mathbf{g}_i - \mathbf{g}, \mathbf{g}\rangle + \|\mathbf{g}\|^2 \right)$$

$$= \sum_{i=1}^{n-b_f} \|\mathbf{g}_i - \mathbf{g}\|^2 + 2\left\langle \sum_{i=1}^{n-b_f} (\mathbf{g}_i - \mathbf{g}), \mathbf{g} \right\rangle + \sum_{i=1}^{n-b_f} \|\mathbf{g}\|^2$$

The middle term is zero because $\sum_{i=1}^{n-b_f}(\mathbf{g}_i - \mathbf{g}) = (\sum \mathbf{g}_i) - (n-b_f)\mathbf{g} = (n-b_f)\mathbf{g} - (n-b_f)\mathbf{g} = 0$. This leaves us with:

$$\sum_{i=1}^{n-b_f} \|\mathbf{g}_i\|^2 = \sum_{i=1}^{n-b_f} \|\mathbf{g}_i - \mathbf{g}\|^2 + (n-b_f)\|\mathbf{g}\|^2$$

Using the heterogeneity assumption from your proof, $\sum_{i=1}^{n-b_f} \|\mathbf{g}_i - \mathbf{g}\|^2 \le (n-b_f)H_k^2$, we get our final bound on this sum:

$$\sum_{i=1}^{n-b_f} \|\mathbf{g}_i\|^2 \le (n-b_f)H_k^2 + (n-b_f)\|\mathbf{g}\|^2 \tag{79}$$

To compute the upper bound for $\sum_{i \in \Lambda^c} \mathbb{E}\{\|\mathbf{v}_i\|_2^2\} + b_f \mathbb{E}\{\|\mathbf{v}_{\text{map}(k)}\|_2^2\}$, we have

$$\sum_{i \in \Lambda^c} \mathbb{E}\{\|\mathbf{v}_i\|_2^2\} + b_f \mathbb{E}\{\|\mathbf{v}_{\text{map}(k)}\|_2^2\}$$

$$\le (n-b_f)d\sigma_k^2 + \sum_{i \in \Lambda^c} \left| \|\mathbf{g}_i\|_2^2 - \|\mathbf{g}\|_2^2 \right| + b_f \left| \|\mathbf{g}_{\text{map}(k)}\|_2^2 - \|\mathbf{g}\|_2^2 \right| + (n-b_f)\|\mathbf{g}\|_2^2$$

$$\le (n-b_f)d\sigma_k^2 + (n-b_f)K_k^2 + (n-b_f)\|\mathbf{g}\|_2^2. \tag{80}$$

Replacing (79) and (80) into (78) gives us:

$$\mathbb{E}\{|E_{\text{new}}|\} \le L_{\max}\alpha^2 \left( \|\mathbf{g}\|^2 + \frac{H_k^2}{2} + \frac{K_k^2}{2} + d\sigma_k^2 + \varepsilon_S^2 \right) \tag{81}$$

ASSEMBLING THE RESULT FOR IDEAL ERROR

We substitute the bounds from (72), (73), (74), and (81) into the expectation of the main inequality (71).

$$\frac{\alpha}{2(n-b_f)^2} \sum_{i \in \Lambda^c, j \in \Lambda^b} \mathbb{E}\{\|\mathbf{v}_i - \mathbf{b}_j\|_2^2\}$$

$$\le \varepsilon_k + \frac{\alpha}{n-b_f} \left( b_f(2K_k^2 + d\sigma_k^2 + \varepsilon_S^2) \right) + \frac{(2\alpha d\sigma_k^2)(n-b_f-1)}{n-b_f} 2d\sigma_k^2$$

$$+ \alpha H_k^2 + L_{\max}\alpha^2 \left( \|\mathbf{g}\|^2 + \frac{H_k^2}{2} + \frac{K_k^2}{2} + d\sigma_k^2 + \varepsilon_S^2 \right) + \alpha\varepsilon_S^2$$

$$= \varepsilon_k + \frac{2K_k^2\alpha b_f}{n-b_f} + \frac{\varepsilon_S^2 \alpha b_f}{n-b_f} + \alpha H_k^2 + \frac{(\alpha d\sigma_k^2)(2n-b_f-2)}{n-b_f}$$

$$+ L_{\max}\alpha^2 \left( \|\mathbf{g}\|^2 + \frac{H_k^2}{2} + \frac{K_k^2}{2} + d\sigma_k^2 + \varepsilon_S^2 \right) + \alpha\varepsilon_S^2.$$

Dividing the entire inequality by $\alpha/2$, we obtain the final bound on the expected distance:

$$\frac{1}{(n-b_f)^2} \sum_{i \in \Lambda^c} \sum_{j \in \Lambda^b} \mathbb{E}\left\{ \|\mathbf{v}_i - \mathbf{b}_j\|^2 \right\}$$

$$\leq \frac{2\varepsilon_k}{\alpha} + \left( \frac{4K_k^2 b_f}{n-b_f} + \frac{2\varepsilon_S^2 b_f}{n-b_f} + 2H_k^2 + 2\varepsilon_S^2 \right) + \frac{(2d\sigma^2)(2n - b_f - 2)}{n - b_f}$$

$$+ 2L_{\max}\alpha \left( \|\mathbf{g}\|^2 + \frac{H_k^2}{2} + \frac{K_k^2}{2} + d\sigma_k^2 + \varepsilon_S^2 \right). \tag{82}$$

To eliminate the $\|\mathbf{g}\|^2$ dependency and using $\alpha \leq \frac{1}{L_{\max}}$, we absorb the $2L_{\max}\alpha\|\mathbf{g}\|^2$ part of the Taylor error into the same budget:

$$2L_{\max}\alpha\|\mathbf{g}\|^2 \leq C_{\text{het}} \implies \alpha \leq \min\{\frac{1}{L_{\max}}, \frac{C_{\text{het}}}{2L_{\max}\|\mathbf{g}\|^2}\}$$

where where the constant "absorption budget" $C_{\text{het}}$ is defined as:

$$C_{\text{het}} := \frac{4K_k^2 b_f}{n - b_f} + 2H_k^2 + \frac{2d\sigma_k^2(2n - b_f - 2)}{n - b_f} + \frac{2\varepsilon_S^2 n}{n - b_f}$$

Using inequality (82) with $2L_{\max}\alpha\|\mathbf{g}\|^2 \leq C_{\text{het}}$ in (60) and noting that $|\Lambda^d| \leq b_f$ results in

$$\|\mathbb{E}\{F - s\}\|_2 \leq \sqrt{b_f \left( \frac{2\varepsilon_k}{\alpha} + 2C_{\text{het}} + 2L_{\max}\alpha(\frac{H_k^2}{2} + \frac{K_k^2}{2} + d\sigma_k^2 + \varepsilon_S^2) \right)} \tag{83}$$

Incorporating the above inequality in (57), we obtain

$$\|\mathbb{E}\{F\} - \mathbf{g}\|_2 \leq \sqrt{2b_f \left( \frac{\varepsilon_k}{\alpha} + C_{\text{het}} + L_{\max}\alpha(\frac{H_k^2}{2} + \frac{K_k^2}{2} + d\sigma_k^2 + \varepsilon_S^2) \right)} + \frac{2b_f}{\sqrt{n - b_f}} H_k \tag{84}$$

Combining the above results with (51) gives us

$$\|\mathbb{E}\{\tilde{F}\} - \mathbf{g}\|_2 \leq \underbrace{\sqrt{2b_f \left( \frac{\varepsilon_k}{\alpha} + C_{\text{het}} + L_{\max}\alpha(\frac{H_k^2}{2} + \frac{K_k^2}{2} + d\sigma_k^2 + \varepsilon_S^2) \right)} + \frac{2b_f}{\sqrt{n - b_f}} H_k + \varepsilon_S}_{\eta}$$

$$\tag{85}$$

By assumption, $\eta < \|\mathbf{g}\|_2$, i.e. $\mathbb{E}\{\tilde{F}\}$ belongs to a ball centered at $\mathbf{g}$ with radius $\eta$. This implies

$$\langle \mathbb{E}\{\tilde{F}\}, \mathbf{g} \rangle \geq (\|\mathbf{g}\|_2 - \eta)\|\mathbf{g}\|_2 = (1 - \sin\alpha)\|\mathbf{g}\|_2^2 \tag{86}$$

To finalize the proof, we need to verify the second condition of Definition 5 using our method. To do this, we have

$$\|\tilde{F}\|_2 = \|\frac{1}{n - b_f}\left( \sum_{i \in \Lambda^c} \tilde{\mathbf{v}}_i + \sum_{j \in \Lambda^b} \mathbf{b}_j \right)\|_2 = \frac{1}{n - b_f}\|\sum_{i \in \Lambda^c} \tilde{\mathbf{v}}_i + \sum_{j \in \Lambda^b} \mathbf{b}_j\|_2 \tag{87}$$

Using the triangle inequality in the above equation and $\|\mathbf{b}_i\|_2^2 \leq \max_{l \in \mathcal{H}} \|\tilde{\mathbf{v}}_l\|_2^2$ results in

$$\|\tilde{F}\|_2 \leq \frac{1}{n - b_f}\left( \sum_{i \in \Lambda^c} \|\tilde{\mathbf{v}}_i\|_2 + \sum_{j \in \Lambda^b} \|\mathbf{b}_j\|_2 \right) \leq \frac{1}{n - b_f}\left( \sum_{i \in \Lambda^c} \|\tilde{\mathbf{v}}_i\|_2 + b_f \max_{l \in \mathcal{H}} \|\tilde{\mathbf{v}}_l\|_2 \right) \tag{88}$$

Using $\max_{l \in \mathcal{H}} \|\tilde{\mathbf{v}}_l\| \leq \sum_{l \in \mathcal{H}} \|\tilde{\mathbf{v}}_l\|_2$, we have

$$\|\tilde{F}\|_2 \leq \frac{1}{n - b_f}\left( \sum_{i \in \Lambda^c} \|\tilde{\mathbf{v}}_i\|_2 + b_f \max_{l \in \mathcal{H}} \|\tilde{\mathbf{v}}_l\| \right)$$

$$\leq \frac{1}{n - b_f}\left( \sum_{l \in \mathcal{H}} \|\tilde{\mathbf{v}}_l\|_2 + b_f \sum_{l \in \mathcal{H}} \|\tilde{\mathbf{v}}_l\|_2 \right) = \frac{1 + b_f}{n - b_f} \sum_{l \in \mathcal{H}} \|\tilde{\mathbf{v}}_l\|_2.$$

Finally, using the inequality $(a+b)^r \leq 2^{r-1}(a^r+b^r)$ and the multinomial theorem, we have

$$\|\tilde{F}\|_2^r \leq \frac{(1+b_f)^r}{(n-b_f)^r} \Big( \sum_{i=1}^{n-b_f} \|\mathbf{v}_i\|_2 + (n-b_f)\varepsilon_S \Big)^r \leq$$

$$\frac{2^{r-1}(1+b_f)^r}{(n-b_f)^r} \Big( (n-b_f)^r \varepsilon_S^r + \sum_{\substack{r_1+\cdots+r_n=r \\ r_1,\ldots,r_n \geq 0}} \frac{n!}{r_1!r_2!\ldots r_n!} \|\mathbf{v}_1\|_2^{r_1} \|\mathbf{v}_2\|_2^{r_2} \ldots \|\mathbf{v}_n\|_2^{r_n} \Big) \tag{89}$$

Since $G_1, \ldots, G_n$ are independent, we obtain

$$\mathbb{E}\{\|\tilde{F}\|_2^r\} \leq \frac{2^{r-1}(1+b_f)^r}{(n-b_f)^r} \sum_{\substack{r_1+\cdots+r_n=r \\ r_1,\ldots,r_n \geq 0}} \frac{n!}{r_1!r_2!\ldots r_n!} \prod_{i=1}^{n} \mathbb{E}\{\|G_i\|_2^{r_i}\} + 2^{r-1}(1+b_f)^r \varepsilon_S^r \tag{90}$$

This concludes the proof of Theorem 2.

## G   BYZANTINE RESILIENCE AGAINST ADVERSARIAL LOSS AND GRADIENT

**Theorem 8.** *If all the assumptions stated in Assumption 1 hold, and suppose $b_f$ mini-batch gradient updates $\tilde{\mathbf{v}}_i$ and the corresponding loss values $f_i$ are replaced with their Byzantine counterparts $\mathbf{b}_i$ and $\tilde{f}_i$, respectively (e.g. data poisoning attack) with $2b_f + 2 \leq n$. Then, for a step-size $0 < \alpha \leq \alpha_{\max}$ in which $\alpha_{\max} = \min\left\{ \frac{1}{L_{\max}}, \frac{C_{het}}{2L_{\max}\|\mathbf{g}\|^2} \right\}$, we have*

$$\|\mathbb{E}\{\tilde{F}\} - \mathbf{g}\|_2 \leq \underbrace{\sqrt{2b_f\Big(\frac{1}{\alpha}(\varepsilon_k + \frac{b_f}{n-b_f}m_{avg}) + C_{het} + L_{\max}\alpha(\frac{H_k^2}{2} + \frac{K_k^2}{2} + d\sigma_k^2 + \varepsilon_S^2)\Big)} + \frac{2b_f}{\sqrt{n-b_f}}H_k + \varepsilon_S}_{\tilde{\eta}}.$$

$$\tag{91}$$

*and if $\tilde{\eta} < \|\mathbf{g}\|_2$, we have*

$$\langle \mathbb{E}\{\tilde{F}\}, \mathbf{g} \rangle \geq (\|\mathbf{g}\|_2 - \tilde{\eta})\|\mathbf{g}\|_2 = (1 - \sin\alpha)\|\mathbf{g}\|_2^2. \tag{92}$$

*Proof.* The proof of the above lemma closely follows the approach used in Theorem 2. However, unlike Theorem 2, which considers only the presence of Byzantine gradient updates, this theorem accounts for both Byzantine gradient updates and Byzantine loss values, requiring additional care in the analysis.

The main difference between the results of both loss and gradient attacks, compared to the model attack, is that some losses are replaced with their Byzantine versions. So, based on (66), we need to replace the losses of adversarial clients with the Byzantine version, resulting in

$$\frac{\alpha}{2(n-b_f)^2} \sum_{i \in \Lambda^c} \sum_{j \in \Lambda^b} \|\mathbf{v}_i - \mathbf{b}_j\|_2^2 \leq \frac{1}{n-b_f} \Big( \sum_{i \in \Lambda^d} f_i(\boldsymbol{\theta}_k) - \sum_{j \in \Lambda^b} \tilde{f}_j(\boldsymbol{\theta}_k) \Big) +$$

$$\underbrace{\frac{\alpha}{n-b_f} \sum_{i \in \Lambda^b} \|\mathbf{b}_i\|_2^2 - \frac{\alpha}{n-b_f} \sum_{i \in \Lambda^d} \|\mathbf{v}_i\|_2^2 + \frac{\alpha}{2(n-b_f)^2} \sum_{i=1}^{n-b_f} \sum_{j=1}^{n-b_f} \|\mathbf{v}_i - \mathbf{v}_j\|_2^2}_{term1}}{+ E_{\text{taylor}}(\mathbf{w}^t) - E_{\text{taylor}}(\mathbf{w}^o)} \tag{93}$$

Using the assumptions that $\|\mathbf{b}_i\|_2^2 \leq \max_{l \in \mathcal{H}} \|\tilde{\mathbf{v}}_l\|_2^2$, the byzantine losses $\tilde{f}_j$ are nonnegative, and using $m_i = \mathbb{E}\{f_i(\boldsymbol{\theta})\}$, we have

$$\frac{1}{(n-b_f)^2} \sum_{i \in \Lambda^c} \sum_{j \in \Lambda^b} \mathbb{E}\{\|\mathbf{v}_i - \mathbf{b}_j\|_2^2\} \leq \underbrace{\frac{2}{\alpha(n-b_f)} \sum_{i \in \Lambda^d} m_i}_{term2} + \frac{2}{\alpha}\mathbb{E}\{term1\}. \tag{94}$$

It is worth noting that $term1$ is exactly the same as the term following the loss difference in the right-hand side of the inequality (66). Using (82), we can write

$$\frac{2}{\alpha}\mathbb{E}\{term1\} = \left(\frac{4K_k^2 b_f}{n-b_f} + \frac{2\varepsilon_S^2 b_f}{n-b_f} + 2H_k^2\right) + \frac{(2d\sigma_k^2)(2n-b_f-2)}{n-b_f}$$
$$+ 2L_{\max}\alpha\left(\|\mathbf{g}\|^2 + \frac{H_k^2}{2} + \frac{K_k^2}{2} + d\sigma_k^2 + \varepsilon_S^2\right). \tag{95}$$

Now we need to bound the term $term2$. Based on Assumption 1, we know $\frac{1}{n-b_f}\sum_{i=1}^{n-b_f}|m_i - m_{avg}| \leq \varepsilon_k$ and using $|\Lambda^d| \leq b_f$, we have

$$\frac{2}{\alpha(n-b_f)}\sum_{i\in\Lambda^d} m_i - m_{avg} + m_{avg} \leq \frac{2}{\alpha(n-b_f)}\left(\sum_{i\in\Lambda^d}|m_i - m_{avg}| + b_f m_{avg}\right)$$
$$\leq \frac{2}{\alpha}\left(\varepsilon_k + \frac{b_f}{n-b_f}m_{avg}\right). \tag{96}$$

Using (95) and (96) in (94) yields to

$$\frac{1}{(n-b_f)^2}\sum_{i\in\Lambda^c}\sum_{j\in\Lambda^b}\mathbb{E}\{\|\mathbf{v}_i - \mathbf{b}_j\|_2^2\}$$
$$\leq \frac{2}{\alpha}\left(\varepsilon_k + \frac{b_f}{n-b_f}m_{avg}\right) + \left(\frac{4K_k^2 b_f}{n-b_f} + \frac{2\varepsilon_S^2 b_f}{n-b_f} + 2H_k^2\right) + \frac{(2d\sigma_k^2)(2n-b_f-2)}{n-b_f}$$
$$+ 2L_{\max}\alpha\left(\|\mathbf{g}\|^2 + \frac{H_k^2}{2} + \frac{K_k^2}{2} + d\sigma_k^2 + \varepsilon_S^2\right). \tag{97}$$

The rest of the proof is the same as the one used in the proof of Theorem 2. Consequently,

$$\|\mathbb{E}\{\tilde{F}\} - \mathbf{g}\|_2 \leq \sqrt{2b_f\left(\frac{1}{\alpha}(\varepsilon_k + \frac{b_f}{n-b_f}m_{avg}) + C_{\text{het}} + L_{\max}\alpha(\frac{H_k^2}{2} + \frac{K_k^2}{2} + d\sigma_k^2 + \varepsilon_S^2)\right)}$$
$$+ \frac{2b_f}{\sqrt{n-b_f}}H_k + \varepsilon_S. \tag{98}$$

By assumption, $\tilde{\eta} < \|\mathbf{g}\|_2$, i.e. $\mathbb{E}\{\tilde{F}\}$ belongs to a ball centered at $\mathbf{g}$ with radius $\tilde{\eta}$. This implies

$$\langle\mathbb{E}\{\tilde{F}\}, \mathbf{g}\rangle \geq (\|\mathbf{g}\|_2 - \tilde{\eta})\|\mathbf{g}\|_2 = (1 - \sin\alpha)\|\mathbf{g}\|_2^2. \tag{99}$$

The remainder of the proof proceeds by following the same reasoning as the argument presented after (86).

This concludes the proof of Theorem 8. $\qquad\square$

## H  PROOF OF THEOREM 3 AND DETERMINING LIPSCHITZ CONSTANT $L_w$

### H.1  PROOF OF THEOREM 3 FOR L-SMOOTH (ITEM 1)

In this section, we provide the proof of Item 1 of Theorem 3.

**Theorem 9.** *Consider the cost function* (1) *under Assumption 1. The sequence* $\{\boldsymbol{\theta}_k, \mathbf{w}_k\}_{k=1}^{\infty}$ *generated by Algorithm 2 satisfies the corresponding result of Item 1 of Theorem 3.*

*Proof.* To streamline notation and maintain consistency with Assumption 1, we denote the true population gradient by $\mathbf{v}_{k,i} = \nabla_\theta f_i(\boldsymbol{\theta}_k)$, and the mini-batch gradient by $\tilde{\mathbf{v}}_{k,i}$. It is important to note that, in the optimality condition of our method—since it is solved on the server side—we work with mini-batch gradients, whereas in the descent lemma we analyze the true gradients.

Since each $f_i$ is continuously differentiable and its gradient $\nabla_\theta f_i$ is $L_i$-Lipschitz continuous, Lemma 4 directly implies that

$$f_i(\tilde{\boldsymbol{\theta}}_{k+1}) \leq f_i(\boldsymbol{\theta}_k) + \langle\mathbf{v}_{k,i}, \tilde{\boldsymbol{\theta}}_{k+1} - \boldsymbol{\theta}_k\rangle + \frac{L_i}{2}\|\tilde{\boldsymbol{\theta}}_{k+1} - \boldsymbol{\theta}_k\|_2^2. \tag{100}$$

Multiplying both sides of the above inequality by $w_{k,i}$ and summing over $i = 1$ to $n$, we have:

$$\sum_{i=1}^{n} w_{k,i} f_i(\tilde{\boldsymbol{\theta}}_{k+1}) \leq \sum_{i=1}^{n} w_{k,i} f_i(\boldsymbol{\theta}_k) + \langle \sum_{i=1}^{n} w_{k,i} \mathbf{v}_{k,i}, \tilde{\boldsymbol{\theta}}_{k+1} - \boldsymbol{\theta}_k \rangle + \frac{\sum_{i=1}^{n} w_{k,i} L_i}{2} \|\tilde{\boldsymbol{\theta}}_{k+1} - \boldsymbol{\theta}_k\|_2^2.$$
(101)

It is important to note that in the above inequality, all gradients $\{\mathbf{v}_{k,i}\}_{i=1}^{n}$ are honest ones; otherwise, the descent lemma breaks. We use the notation $F_k^c(\mathbf{w}_k) = \sum_{i=1}^{n} w_{k,i} \mathbf{v}_{k,i}$ for the all honest gradients.

In the first step of our proposed method to find the $\boldsymbol{\theta}$, we utilize (5). Since, $\tilde{\boldsymbol{\theta}}_{k+1} = \boldsymbol{\theta}_k - \alpha \mathbf{G}_k \mathbf{w}_k = \boldsymbol{\theta}_k - \alpha F_k(\mathbf{w}_k)$ is the minimizer of (5) when $\mathbf{w} = \mathbf{w}^k$, we can write

$$\langle \sum_{i \in \mathcal{H}} w_{k,i} \tilde{\mathbf{v}}_{k,i} + \sum_{i \in \mathcal{H}^\complement} w_{k,i} \mathbf{b}_{k,i}, \tilde{\boldsymbol{\theta}}_{k+1} - \boldsymbol{\theta}_k \rangle + \frac{1}{2\alpha} \|\tilde{\boldsymbol{\theta}}_{k+1} - \boldsymbol{\theta}_k\|_2^2 \leq 0.$$
(102)

Here, the term $\sum_{i \in \mathcal{H}} w_{k,i} \tilde{\mathbf{v}}_{k,i} + \sum_{i \in \mathcal{H}^\complement} w_{k,i} \mathbf{b}_{k,i}$ accounts for both the contributions from the set of honest clients $\mathcal{H}$ and the complement set $\mathcal{H}^\complement$. Crucially, because the optimization problem (5) is solved at the server side, the gradients of Byzantine clients may be arbitrarily substituted by $\mathbf{b}_{k,i}$. This explains why the inequality above involves $\mathbf{b}_{k,i}$ rather than the true gradients.

Adding the above inequality to (101) gives us

$$\sum_{i=1}^{n} w_{k,i} f_i(\tilde{\boldsymbol{\theta}}_{k+1}) + \langle F_k(\mathbf{w}_k), -\alpha F_k(\mathbf{w}_k) \rangle + \frac{1}{2\alpha} \|\tilde{\boldsymbol{\theta}}_{k+1} - \boldsymbol{\theta}_k\|_2^2 \leq \sum_{i=1}^{n} w_{k,i} f_i(\boldsymbol{\theta}_k) +$$

$$\langle F_k^c(\mathbf{w}_k), -\alpha F_k(\mathbf{w}_k) \rangle + \frac{\sum_{i=1}^{n} w_{k,i} L_i}{2} \|\tilde{\boldsymbol{\theta}}_{k+1} - \boldsymbol{\theta}_k\|_2^2.$$
(103)

Simplifying this inequality and using $\sum_{i=1}^{n} w_{k,i} L_i \leq \sum_{i=1}^{n} w_{k,i} L_{\max} \leq L_{\max}$, we have

$$\sum_{i=1}^{n} w_{k,i} f_i(\tilde{\boldsymbol{\theta}}_{k+1}) \leq \sum_{i=1}^{n} w_{k,i} f_i(\boldsymbol{\theta}_k) + \alpha \langle F_k(\mathbf{w}_k) - F_k^c(\mathbf{w}_k), F_k(\mathbf{w}_k) \rangle$$

$$+ \left( \frac{L_{\max}}{2} - \frac{1}{2\alpha} \right) \alpha^2 \|F_k(\mathbf{w}_k)\|_2^2.$$
(104)

On the other hand using Lemma 4 for $\mathbf{w}^T \mathbf{f}(\boldsymbol{\theta}_k - \alpha \mathbf{G}_k \mathbf{w})$ results in

$$\sum_{i=1}^{n} w_{k+1,i} f_i(\boldsymbol{\theta}_k - \alpha \mathbf{G}_k \mathbf{w}_{k+1}) \leq \sum_{i=1}^{n} w_{k,i} f_i(\boldsymbol{\theta}_k - \alpha \mathbf{G}_k \mathbf{w}_k) +$$

$$\langle \nabla_w \mathbf{w}_k^T \mathbf{f}(\boldsymbol{\theta}_k - \alpha \mathbf{G}_k \mathbf{w}_k), \mathbf{w}_{k+1} - \mathbf{w}_k \rangle + \frac{L_w}{2} \|\mathbf{w}_{k+1} - \mathbf{w}_k\|_2^2.$$
(105)

In terms of $\boldsymbol{\theta}_{k+1} = \boldsymbol{\theta}_k - \alpha \mathbf{G}_k \mathbf{w}_{k+1}$ and $\tilde{\boldsymbol{\theta}}_{k+1} = \boldsymbol{\theta}_k - \alpha \mathbf{G}_k \mathbf{w}_k$, this inequality reads

$$\sum_{i=1}^{n} w_{k+1,i} f_i(\boldsymbol{\theta}_{k+1}) \leq \sum_{i=1}^{n} w_{k,i} f_i(\tilde{\boldsymbol{\theta}}_{k+1}) + \langle \nabla_w \mathbf{w}_k^T \mathbf{f}(\tilde{\boldsymbol{\theta}}_{k+1}), \mathbf{w}_{k+1} - \mathbf{w}_k \rangle + \frac{L_w}{2} \|\mathbf{w}_{k+1} - \mathbf{w}_k\|_2^2$$
(106)

In the above inequality, since we are writing a descent lemma, all gradients are true.

It is straightforward to show that the optimization problem (9) is equivalent to the following formulation:

$$\mathbf{w}_{k+1} = \operatorname*{argmin}_{\mathbf{w}} \quad \langle \nabla_w \mathbf{w}_k^T \mathbf{f}(\boldsymbol{\theta}_k - \alpha \mathbf{G}_k \mathbf{w}_k), \mathbf{w} - \mathbf{w}_k \rangle + \frac{1}{2\beta_{k+1}} \|\mathbf{w} - \mathbf{w}_k\|_2^2 + \delta_{\Delta_{t,\ell_0}^+}(\mathbf{w}). \quad (107)$$

The term $\nabla_w \mathbf{w}_k^T \mathbf{f}(\boldsymbol{\theta}_k - \alpha \mathbf{G}_k \mathbf{w}_k) = \mathbf{f}(\tilde{\boldsymbol{\theta}}_{k+1}) - \alpha \mathbf{G}_k^T \tilde{\mathbf{G}}_{k+1} \mathbf{w}_k$ in the above inequality. However, the similar term in (106) does not include Byzantine data, so we just replace $\mathcal{H} = \{1, \cdots, n\}$ in (106), using notation $\tilde{\mathbf{G}}_{k+1}^c$ instead of $\tilde{\mathbf{G}}_{k+1}$.

From (107), since $\mathbf{w}_{k+1}$ minimizes the objective function, its corresponding objective value is less than or equal to that of any other feasible choice, including $\mathbf{w} = \mathbf{w}_k$. This yields the inequality:

$$\langle \nabla_w \sum_{i=1}^n w_{k,i} f_i(\tilde{\boldsymbol{\theta}}_{k+1}), \mathbf{w}_{k+1} - \mathbf{w}_k \rangle + \frac{1}{2\beta_{k+1}} \|\mathbf{w}_{k+1} - \mathbf{w}_k\|_2^2 + \delta_{\Delta_{t,\ell_0}^+}(\mathbf{w}_{k+1}) \leq \delta_{\Delta_{t,\ell_0}^+}(\mathbf{w}_k) \tag{108}$$

Adding the above inequality to (106) yields:

$$\sum_{i=1}^n w_{k+1,i} f_i(\boldsymbol{\theta}_{k+1}) + \delta_{\Delta_{t,\ell_0}^+}(\mathbf{w}_{k+1}) \leq \sum_{i=1}^n w_{k,i} f_i(\tilde{\boldsymbol{\theta}}_{k+1}) +$$
$$\langle \alpha \mathbf{G}_k^T(\tilde{\mathbf{G}}_{k+1} - \tilde{\mathbf{G}}_{k+1}^c)\mathbf{w}_k, \mathbf{w}_{k+1} - \mathbf{w}_k \rangle + (\frac{L_w}{2} - \frac{1}{2\beta_{k+1}})\|\mathbf{w}_{k+1} - \mathbf{w}_k\|_2^2 + \delta_{\Delta_{t,\ell_0}^+}(\mathbf{w}_k) \tag{109}$$

It is straightforward to show that the above inequality can be written

$$\sum_{i=1}^n w_{k+1,i} f_i(\boldsymbol{\theta}_{k+1}) + \delta_{\Delta_{t,\ell_0}^+}(\mathbf{w}_{k+1}) \leq \sum_{i=1}^n w_{k,i} f_i(\tilde{\boldsymbol{\theta}}_{k+1}) +$$
$$\alpha \langle \mathbf{F}_k(\mathbf{w}_{k+1}) - \mathbf{F}_k(\mathbf{w}_k), \mathbf{F}_{k+1}(\mathbf{w}_k) - \mathbf{F}_{k+1}^c(\mathbf{w}_k) \rangle$$
$$+ (\frac{L_w}{2} - \frac{1}{2\beta_{k+1}})\|\mathbf{w}_{k+1} - \mathbf{w}_k\|_2^2 + \delta_{\Delta_{t,\ell_0}^+}(\mathbf{w}_k) \tag{110}$$

Adding inequality (104) to the above inequality yields:

$$\sum_{i=1}^n w_{k+1,i} f_i(\boldsymbol{\theta}_{k+1}) + \sum_{i=1}^n w_{k,i} f_i(\tilde{\boldsymbol{\theta}}_{k+1}) + \delta_{\Delta_{t,\ell_0}^+}(\mathbf{w}_{k+1}) \leq \sum_{i=1}^n w_{k,i} f_i(\boldsymbol{\theta}_k) + \sum_{i=1}^n w_{k,i} f_i(\tilde{\boldsymbol{\theta}}_{k+1}) +$$
$$\alpha \langle \mathbf{F}_k(\mathbf{w}_{k+1}) - \mathbf{F}_k(\mathbf{w}_k), \mathbf{F}_{k+1}(\mathbf{w}_k) - \mathbf{F}_{k+1}^c(\mathbf{w}_k) \rangle + \alpha \langle \mathbf{F}_k(\mathbf{w}_k) - \mathbf{F}_k^c(\mathbf{w}_k), \mathbf{F}_k(\mathbf{w}_k) \rangle +$$
$$(\frac{L_w}{2} - \frac{1}{2\beta_{k+1}})\|\mathbf{w}_{k+1} - \mathbf{w}_k\|_2^2 + (\frac{L_{\max}}{2} - \frac{1}{2\alpha})\|\boldsymbol{\theta}_{k+1} - \boldsymbol{\theta}_k\|_2^2 + \delta_{\Delta_{t,\ell_0}^+}(\mathbf{w}_k) \tag{111}$$

Simplifying the above inequality results in:

$$Q_{k+1} \leq Q_k + \underbrace{\alpha \langle \mathbf{F}_k(\mathbf{w}_{k+1}) - \mathbf{F}_k(\mathbf{w}_k), \mathbf{F}_{k+1}(\mathbf{w}_k) - \mathbf{F}_{k+1}^c(\mathbf{w}_k) \rangle}_{\mathcal{E}_{k+1}}$$
$$+ \underbrace{\alpha \langle \mathbf{F}_k(\mathbf{w}_k) - \mathbf{F}_k^c(\mathbf{w}_k), \mathbf{F}_k(\mathbf{w}_k) \rangle}_{\mathcal{I}_k}$$
$$+ \left(\frac{L_{\max}}{2} - \frac{1}{2\alpha}\right)\alpha^2 \underbrace{\|\mathbf{F}_k(\mathbf{w}_{k+1})\|_2^2}_{\mathcal{F}_k} + \left(\frac{L_w}{2} - \frac{1}{2\beta_{k+1}}\right)\|\mathbf{w}_{k+1} - \mathbf{w}_k\|_2^2, \tag{112}$$

where $Q_{k+1} = \sum_{i=1}^n w_{k+1,i} f_i(\boldsymbol{\theta}_{k+1}) + \delta_{\Delta_{t,\ell_0}^+}(\mathbf{w}_{k+1})$. To guarantee the exclusion of exactly $b_f$ clients, we set $t = 1/s$ and $s = n - b_f$ (see proposition 2). With this choice, the nonzero weights become $1/(n - b_f)$, ensuring that our algorithm removes precisely $b_f$ clients.

To continue the proof, we take the expectation of the above inequality over all sources of randomness. We first focus on deriving an upper bound for the expectation of the cross-time-step error term:

$$\mathcal{E}_{k+1} = \alpha \langle F_k(\mathbf{w}_{k+1}) - F_k(\mathbf{w}_k), F_{k+1}(\mathbf{w}_k) - F_{k+1}^c(\mathbf{w}_k) \rangle,$$

which is formalized in the following lemma.

**Lemma 10.** *Let Item B2 of Assumptions 1 hold. The expected error is bounded by:*

$$\mathbb{E}[\mathcal{E}_{k+1}] \leq 2\alpha\zeta_{k+1}\left(\|\mathbf{g}_k\| + \sqrt{K_k^2 + d\sigma_k^2 + \varepsilon_S^2}\right). \tag{113}$$

*Proof.* The proof proceeds by first separating the randomness from different time steps.

Let $\mathbb{E}_k[\cdot]$ denote the expectation conditioned on all information up to step $k$. We rewrite the total expectation using the law of total expectation, $\mathbb{E}[\cdot] = \mathbb{E}_k[\mathbb{E}_{k+1}[\cdot]]$. Let $E_{Byz}^{k+1} = F_{k+1}(\mathbf{w}_k) - F_{k+1}^c(\mathbf{w}_k)$.

$$\mathbb{E}[\mathcal{E}_{k+1}] = \alpha \cdot \mathbb{E}_k \left[ \langle F_k(\mathbf{w}_{k+1}) - F_k(\mathbf{w}_k), \mathbb{E}_{k+1}[E_{Byz}^{k+1}] \rangle \right].$$

The inner expectation is the bias of the aggregator at step $k + 1$ using the weights from step $k$. By our Bias Assumption, its norm is bounded by $\zeta_{k+1}$. Applying the Cauchy-Schwarz inequality:

$$\mathbb{E}[\mathcal{E}_{k+1}] \leq \alpha \zeta_{k+1} \cdot \mathbb{E}[\|\Delta F_k\|], \tag{114}$$

where $\Delta F_k = F_k(\mathbf{w}_{k+1}) - F_k(\mathbf{w}_k)$. The problem is now reduced to finding an upper bound for $\mathbb{E}[\|\Delta F_k\|]$.

The change in the aggregator is $\Delta F_k = \mathbf{G}_k(\mathbf{w}_{k+1} - \mathbf{w}_k)$. Let $m_k$ be the number of clients whose weights are swapped between iterations. Under the simplified weight structure, $\Delta F_k = \frac{1}{n-b_f}(\sum_{i \in \mathcal{A}_k^c} \tilde{\mathbf{v}}_{k,i} + \sum_{i \in \mathcal{A}_k^b} \mathbf{b}_{k,i} - \sum_{j \in \mathcal{R}_k^c} \tilde{\mathbf{v}}_{k,j} - \sum_{j \in \mathcal{R}_k^b} \mathbf{b}_{k,j})$ in which $\mathbf{b}_{k,j}$ is Byzantine gradient of client j at round $k$. The change in weights from $\mathbf{w}_k$ to $\mathbf{w}_{k+1}$ occurs because some clients are removed from the active set and replaced by others. We formally define these sets of swapped clients:

- Let $\mathcal{A}_k = \text{supp}(\mathbf{w}_{k+1}) \setminus \text{supp}(\mathbf{w}_k)$ be the set of "added" clients.

- Let $\mathcal{R}_k = \text{supp}(\mathbf{w}_k) \setminus \text{supp}(\mathbf{w}_{k+1})$ be the set of "removed" clients.

Since the size of the active set is constant at $s = n - b_f$, we have $|\mathcal{A}_k| = |\mathcal{R}_k| = m_k$. We can further partition these sets into honest ($\mathcal{H}$) and Byzantine ($\mathcal{H}^{\complement}$) clients:

- Added honest/Byzantine: $\mathcal{A}_k^c = \mathcal{A}_k \cap \mathcal{H}$, $\mathcal{A}_k^b = \mathcal{A}_k \cap \mathcal{H}^{\complement}$.

- Removed honest/Byzantine: $\mathcal{R}_k^c = \mathcal{R}_k \cap \mathcal{H}$, $\mathcal{R}_k^b = \mathcal{R}_k \cap \mathcal{H}^{\complement}$.

Under the simplified weight structure where non-zero weights are $1/(n - b_f)$, the change in the aggregator $\Delta F_k$ can be written explicitly as:

$$\Delta F_k = \frac{1}{n - b_f} \left( \left( \sum_{i \in \mathcal{A}_k^c} \tilde{\mathbf{v}}_{k,i} + \sum_{i \in \mathcal{A}_k^b} \mathbf{b}_{k,i} \right) - \left( \sum_{j \in \mathcal{R}_k^c} \tilde{\mathbf{v}}_{k,j} + \sum_{j \in \mathcal{R}_k^b} \mathbf{b}_{k,j} \right) \right).$$

To bound $\mathbb{E}[\|\Delta F_k\|]$, we first bound the expected squared norm, $\mathbb{E}[\|\Delta F_k\|^2]$, using the assumption $\|\mathbf{b}_{k,j}\| \leq \max_{i \in \mathcal{H}} \|\tilde{\mathbf{v}}_{k,i}\|_2$ and then use Jensen's inequality.

$$\mathbb{E}[\|\Delta F_k\|^2] \leq \frac{2}{(n - b_f)^2} \left( \mathbb{E} \left\{ \left\| \sum_{i \in \mathcal{A}_k^c} \tilde{\mathbf{v}}_{k,i} + \sum_{i \in \mathcal{A}_k^b} \mathbf{b}_{k,i} \right\|^2 \right\} \right.$$

$$+ \mathbb{E} \left\{ \left\| \sum_{i \in \mathcal{R}_k^c} \tilde{\mathbf{v}}_{k,i} + \sum_{i \in \mathcal{R}_k^b} \mathbf{b}_{k,i} \right\|^2 \right\} \right)$$

$$\leq \frac{2}{(n - b_f)^2} \left( m_k \left( \sum_{i \in \mathcal{A}_k^c} \mathbb{E}\{\|\tilde{\mathbf{v}}_{k,i}\|^2\} + |\mathcal{A}_k^b| \mathbb{E}\{\|\tilde{\mathbf{v}}_{k,\text{map(i)}}\|_2^2\} \right) \right.$$

$$+ m_k \left( \sum_{j \in \mathcal{R}_k} \mathbb{E}\{\|\tilde{\mathbf{v}}_{k,i}\|^2\} + |\mathcal{R}_k^b| \mathbb{E}\{\|\tilde{\mathbf{v}}_{k,\text{map(i)}}\|_2^2\} \right) \right).$$

where $\text{map(i)} = \text{argmax}_{i \in \mathcal{H}} \|\tilde{\mathbf{v}}_{k,i}\|_2$. The next step is to bound the expected squared norm of an arbitrary client's gradient, $\mathbb{E}[\|\tilde{\mathbf{v}}_{k,i}\|^2]$. For an honest client $i$, using (76), and $\mathbb{E}\{\|\mathbf{v}_{k,i}\|^2\} \leq \|\mathbf{g}_k\|^2 + K_k^2 + d\sigma_k^2$

$$\mathbb{E}[\|\tilde{\mathbf{v}}_{k,i}\|^2] \leq \|\mathbf{g}_k\|^2 + K_k^2 + d\sigma_k^2 + \varepsilon_S^2.$$

Substituting this in:

$$\mathbb{E}[\|\Delta F_k\|^2] \leq \frac{4m_k^2(\|\mathbf{g}_k\|^2 + K_k^2 + d\sigma_k^2 + \varepsilon_S^2)}{(n - b_f)^2}.$$

Using Jensen's inequality, $\mathbb{E}[X] \leq \sqrt{\mathbb{E}[X^2]}$, we get the bound on the expected norm:

$$\mathbb{E}[\|\Delta F_k\|] \leq \sqrt{\frac{4m_k^2(\|\mathbf{g}_k\|^2 + K_k^2 + d\sigma_k^2 + \varepsilon_S^2)}{(n - b_f)^2}}. \tag{115}$$

The bound in (115) depends on $\|\mathbf{g}_k\|$. To split it, we use the inequality $\sqrt{x + y} \leq \sqrt{x} + \sqrt{y}$ for $x, y \geq 0$.

$$\sqrt{\|\mathbf{g}_k\|^2 + K_k^2 + d\sigma_k^2 + \varepsilon_S^2} \leq \sqrt{\|\mathbf{g}_k\|^2} + \sqrt{K_k^2 + d\sigma_k^2 + \varepsilon_S^2} = \|\mathbf{g}_k\| + \sqrt{K_k^2 + d\sigma_k^2 + \varepsilon_S^2}.$$

Let $C_K = \sqrt{K_k^2 + d\sigma_k^2 + \varepsilon_S^2}$. Substituting this into the bound for $\mathbb{E}[\|\Delta F_k\|]$:

$$\mathbb{E}[\|\Delta F_k\|] \leq \frac{2m_k}{n - b_f}(\|\mathbf{g}_k\| + C_K).$$

using the above inequality from (114) and $m_k \leq n - b_f$, we have:

$$\begin{aligned}
\mathbb{E}[\mathcal{E}_{k+1}] &\leq \alpha\zeta_{k+1} \cdot \mathbb{E}[\|\Delta F_k\|] \\
&\leq \alpha\zeta_{k+1} \cdot \frac{2m_k}{n - b_f}(\|\mathbf{g}_k\| + C_K) \\
&\leq 2\alpha\zeta_{k+1}\|\mathbf{g}_k\| + 2\alpha\zeta_{k+1}C_K.
\end{aligned}$$

This completes the proof of Lemma 10. $\qquad\square$

Next, we focus on deriving the upper bound of the expectation of $\mathcal{I}_k$ in (112).

**Lemma 11** (Bound on the Single-Step Bias Term). *Let Assumptions 1 hold. Then, the expectation of the single-step bias term is bounded by:*

$$\mathbb{E}_k\{\alpha\langle F_k - F_k^c, F_k\rangle\} \leq \alpha\left(\zeta_k\|\mathbf{g}_k\| + \zeta_k^2 + \sigma_{F,k}^2 + \sigma_k\sqrt{d}\sqrt{\zeta_k^2 + \sigma_{F,k}^2}\right). \tag{116}$$

*Proof.* The derivation proceeds by decomposing the inner product with respect to the true gradient $\mathbf{g}_k$. The constant $\alpha$ can be handled at the end. We focus on finding an upper bound for $\mathcal{I}_k = \mathbb{E}_k[\langle F_k - F_k^c, F_k\rangle]$.

First, we have

$$\langle F_k - F_k^c, F_k\rangle = \langle (F_k - \mathbf{g}_k) - (F_k^c - \mathbf{g}_k), \mathbf{g}_k + (F_k - \mathbf{g}_k)\rangle.$$

We can expand this inner product, which results in four terms:

$$\langle F_k - F_k^c, F_k\rangle = \underbrace{\langle F_k - \mathbf{g}_k, \mathbf{g}_k\rangle}_{\text{Term 1}} + \underbrace{\|F_k - \mathbf{g}_k\|^2}_{\text{Term 2}}$$
$$- \underbrace{\langle F_k^c - \mathbf{g}_k, \mathbf{g}_k\rangle}_{\text{Term 3}} - \underbrace{\langle F_k^c - \mathbf{g}_k, F_k - \mathbf{g}_k\rangle}_{\text{Term 4}}.$$

We now take the conditional expectation $\mathbb{E}_k[\cdot]$ of each of the four terms.

- **Term 1:** Since $\mathbf{g}_k$ is deterministic at step $k$, we have:

$$\mathbb{E}_k[\langle F_k - \mathbf{g}_k, \mathbf{g}_k \rangle] = \langle \mathbb{E}_k[F_k] - \mathbf{g}_k, \mathbf{g}_k \rangle.$$

  By the Cauchy-Schwarz inequality and the bounded bias assumption:

$$\langle \mathbb{E}_k[F_k] - \mathbf{g}_k, \mathbf{g}_k \rangle \leq \|\mathbb{E}_k[F_k] - \mathbf{g}_k\| \cdot \|\mathbf{g}_k\| \leq \zeta_k \|\mathbf{g}_k\|.$$

- **Term 2:** This is the Mean Squared Error (MSE) of our aggregator, which decomposes into squared bias and variance:

$$\mathbb{E}_k[\|F_k - \mathbf{g}_k\|^2] = \|\mathbb{E}_k[F_k] - \mathbf{g}_k\|^2 + \mathrm{Var}_k(F_k) \leq \zeta_k^2 + \sigma_{F,k}^2.$$

- **Term 3:** Since the clean aggregator $F_k^c$ is unbiased ($\mathbb{E}_k[F_k^c] = \mathbf{g}_k$), this term's expectation is zero:

$$\mathbb{E}_k[\langle F_k^c - \mathbf{g}_k, \mathbf{g}_k \rangle] = \langle \mathbb{E}_k[F_k^c] - \mathbf{g}_k, \mathbf{g}_k \rangle = \langle \mathbf{g}_k - \mathbf{g}_k, \mathbf{g}_k \rangle = 0.$$

- **Term 4:** For the final cross-term, we use the Cauchy-Schwarz inequality for random vectors, $|\mathbb{E}[\langle X, Y \rangle]| \leq \sqrt{\mathbb{E}[\|X\|^2]\mathbb{E}[\|Y\|^2]}$.

$$\begin{aligned}
-\mathbb{E}_k[\langle F_k^c - \mathbf{g}_k, F_k - \mathbf{g}_k \rangle] &\leq |\mathbb{E}_k[\langle F_k^c - \mathbf{g}_k, F_k - \mathbf{g}_k \rangle]| \\
&\leq \sqrt{\mathbb{E}_k[\|F_k^c - \mathbf{g}_k\|^2]} \cdot \sqrt{\mathbb{E}_k[\|F_k - \mathbf{g}_k\|^2]} \\
&\leq \sqrt{d\sigma_k^2} \cdot \sqrt{\zeta_k^2 + \sigma_{F,k}^2} = \sigma_k \sqrt{d(\zeta_k^2 + \sigma_{F,k}^2)}.
\end{aligned}$$

  Summing these four bounds gives the result for $\mathcal{I}_k$. Multiplying by $\alpha$ completes the proof of Lemma 11.

$\square$

Next, we focus on deriving the lower bound of the expectation of $\mathcal{F}_k$ in (112).

**Lemma 12** (Bound on the Expected Squared Aggregator Norm). *Let Assumptions 1 hold. Then, we have:*

$$\mathbb{E}_k[\|F_k\|_2^2] \geq \|\mathbf{g}_k\|_2^2 - 2\zeta_k \|\mathbf{g}_k\|_2. \tag{117}$$

*Proof.* We know

$$\mathbb{E}_k[\|F_k\|_2^2] = \mathrm{Var}_k(F_k) + \|\mathbb{E}_k[F_k]\|_2^2. \tag{118}$$

Since $\mathrm{Var}_k(F_k) \geq 0$, we can therefore drop this term, resulting in

$$\mathbb{E}_k[\|F_k\|_2^2] \geq \|\mathbb{E}_k[F_k]\|_2^2. \tag{119}$$

Using the inequality $\|a + b\|^2 \geq (\|a\| - \|b\|)^2 = \|a\|^2 - 2\|a\|\|b\| + \|b\|^2$, we can write

$$\|\mathbb{E}_k[F_k]\|_2^2 = \|\mathbf{g}_k + (\mathbb{E}_k[F_k] - \mathbf{g}_k)\|_2^2 \geq \|\mathbf{g}_k\|_2^2 - 2\|\mathbf{g}_k\|_2\|\mathbb{E}_k[F_k] - \mathbf{g}_k\|_2 + \|\mathbb{E}_k[F_k] - \mathbf{g}_k\|_2^2.$$

We now use our aggregator's bias assumption, $\|\mathbb{E}_k[F_k] - \mathbf{g}_k\| \leq \zeta_k$ . We can substitute this into the inequality:

$$\|\mathbb{E}_k[F_k]\|_2^2 \geq \|\mathbf{g}_k\|_2^2 - 2\zeta_k \|\mathbf{g}_k\|_2 + \|\mathbb{E}_k[F_k] - \mathbf{g}_k\|_2^2.$$

Since the final term is non-negative, we can drop it from the right-hand side. This gives us:

$$\|\mathbb{E}_k[F_k]\|_2^2 \geq \|\mathbf{g}_k\|_2^2 - 2\zeta_k \|\mathbf{g}_k\|_2. \tag{120}$$

Using (120) in (119) yields to

$$\mathbb{E}_k[\|F_k\|_2^2] \geq \|\mathbb{E}_k[F_k]\|_2^2 \geq \|\mathbf{g}_k\|_2^2 - 2\zeta_k \|\mathbf{g}_k\|_2.$$

This completes the proof of Lemma 12. $\square$

Now we return to (112). We take the expectation $\mathbb{E}[\cdot]$ and substitute the bounds for $\mathcal{E}_{k+1}$, $\mathcal{I}_k$ and $\mathcal{F}_k$ from (113), (116), and (117), respectively.

$$\mathbb{E}\{Q_{k+1}\} \leq \mathbb{E}\{Q_k\} + 2\alpha\zeta_{k+1}\left(\|\mathbf{g}_k\| + \sqrt{K_k^2 + d\sigma_k^2 + \varepsilon_S^2}\right) \qquad \text{(from } \mathcal{E}_k\text{)}$$

$$+ \left(\alpha(\zeta_k\|\mathbf{g}_k\| + \zeta_k^2 + \sigma_{F,k}^2 + \sigma_k\sqrt{d}\sqrt{\zeta_k^2 + \sigma_{F,k}^2})\right) \qquad \text{(from } \mathcal{I}_k\text{)}$$

$$- C_\alpha\alpha^2\left(\|\mathbf{g}_k\|_2^2 - 2\zeta_k\|\mathbf{g}_k\|_2\right) \qquad \text{(from} \mathcal{F}_k\text{)}$$

$$- C_{\beta_{k+1}}\mathbb{E}_k\{\|\mathbf{w}_{k+1} - \mathbf{w}_k\|_2^2\}. \qquad \text{(Weight Descent)}$$

Let $C_\alpha = (\frac{1}{2\alpha} - \frac{L_{\max}}{2}) > 0$ and $C_{\beta_{k+1}} = (\frac{1}{2\beta_{k+1}} - \frac{L_w}{2}) > 0$, resulting in $\alpha < \frac{1}{L_{\max}}$ and $\beta_{k+1} < \frac{1}{L_w}$. We group terms by their dependence on $\|\mathbf{g}_k\|$. Let $B_k\alpha = \alpha(2\zeta_{k+1} + \zeta_k + 2C_\alpha\alpha\zeta_k)$ be the coefficient of the rebound term. Let $\mathcal{C}_{err,k+1} = 2\zeta_{k+1}\sqrt{K_k^2 + d\sigma_k^2 + \varepsilon_S^2} + \left(\zeta_k^2 + \sigma_{F,k}^2 + \sigma_k\sqrt{d}\sqrt{\zeta_k^2 + \sigma_{F,k}^2}\right)$ which collects all constant error terms.

$$\mathbb{E}\{Q_{k+1}\} \leq \mathbb{E}\{Q_k\} - C_\alpha\alpha^2\|\mathbf{g}_k\|_2^2 + B_k\alpha\|\mathbf{g}_k\| + \mathcal{C}_{err,k+1}\alpha - C_{\beta_{k+1}}\mathbb{E}[\|\mathbf{w}_{k+1} - \mathbf{w}_k\|_2^2].$$

The term $B_k\|\mathbf{g}_k\|$ is positive and could counteract the main descent. We use Young's inequality, $ab \leq \frac{\gamma}{2}a^2 + \frac{1}{2\gamma}b^2$, on $B_k\|\mathbf{g}_k\|$:

$$B_k\alpha\|\mathbf{g}_k\| \leq \frac{\gamma}{2}\|\mathbf{g}_k\|^2 + \frac{B_k^2\alpha^2}{2\gamma}.$$

Here, $\gamma$ is a free parameter. We make a standard strategic choice to ensure descent: we set the "rebound" from Young's inequality to be half of the main descent, i.e., $\frac{\gamma}{2} = \frac{1}{2}C_\alpha\alpha^2$, so $\gamma = C_\alpha\alpha^2$. The combined coefficient of $\|\mathbf{g}_k\|^2$ becomes:

$$-C_\alpha\alpha^2 + \frac{\gamma}{2} = -C_\alpha\alpha^2 + \frac{C_\alpha\alpha^2}{2} = -\frac{C_\alpha\alpha^2}{2} = -\left(\frac{\alpha}{4} - \frac{L_{\max}\alpha^2}{4}\right).$$

This is strictly negative for $\alpha < 1/L_{\max}$, guaranteeing descent. The other constant term from Young's inequality is $\frac{B_k^2}{2\gamma}$. Consequently,

$$\mathbb{E}\{Q_{k+1}\} \leq \mathbb{E}\{Q_k\} - C_1\alpha\|\mathbf{g}_k\|^2 + C_{2,k}\alpha - C_{\beta_{k+1}}\mathbb{E}[\|\mathbf{w}_{k+1} - \mathbf{w}_k\|_2^2], \qquad (121)$$

where $C_1 = \frac{1}{4}(1 - L_{\max}\alpha) > 0$ for $0 < \alpha < 1/L_{\max}$, and $C_{2,k}$ is the constant that groups all bias and variance terms: $C_{2,k} = \mathcal{C}_{err,k+1} + \frac{B_k^2}{4C_1}$.

Taking the total expectation from the above inequality, since $\mathbf{g}_k$ is deterministic, and sum from $k = 0$ to $T - 1$:

$$C_1\alpha\sum_{k=0}^{T-1}\|\mathbf{g}_k\|^2 + \sum_{k=0}^{T-1}C_{\beta_{k+1}}\mathbb{E}[\|\mathbf{w}_{k+1} - \mathbf{w}_k\|_2^2] \leq (\mathbb{E}[Q_0] - \mathbb{E}[Q_T]) + \sum_{k=1}^{T}C_{2,k}\alpha. \qquad (122)$$

Now, we keep the first term of the left hand side of the inequality (122), resulting in

$$C_1\alpha\sum_{k=0}^{T-1}\|\mathbf{g}_k\|^2 \leq (\mathbb{E}[Q_0] - \mathbb{E}[Q_T]) + \sum_{k=1}^{T}C_{2,k}\alpha.$$

Let $Q^\star = \inf_{\boldsymbol{\theta},\mathbf{w}} Q(\boldsymbol{\theta}, \mathbf{w})$ be the minimum value of our objective, which we assume is bounded. By definition, for any $k$, $Q_T \geq Q^\star$, and therefore:

$$\mathbb{E}[Q_0] - \mathbb{E}[Q_T] \leq Q_0 - Q^\star. \qquad (123)$$

Substituting this back into our main sum:

$$C_1\alpha\sum_{k=0}^{T-1}\|\mathbf{g}_k\|^2 \leq (Q_0 - Q^\star) + \sum_{k=1}^{T}C_{2,k}\alpha.$$

Dividing by $TC_1\alpha$:

$$\frac{1}{T}\sum_{k=0}^{T-1}\|\mathbf{g}_k\|^2 \leq \frac{Q_0 - Q^\star}{TC_1\alpha} + \frac{\sum_{k=1}^{T}C_{2,k}}{TC_1}.$$

Taking the limit as $T \to \infty$, the first term on the right-hand side vanishes, leaving the final bound on the average of the squared gradients:

$$\lim_{T\to\infty}\sup\frac{1}{T}\sum_{k=0}^{T-1}\|\mathbf{g}_k\|^2 \leq \lim_{T\to\infty}\frac{\sum_{k=1}^{T-1}C_{2,k}}{TC_1}. \tag{124}$$

Now, we provide a detailed analysis of the order of this final error rate.

Since $C_1 = \mathcal{O}(1)$, the final error rate is determined by $C_{2,k}$. Based on the definition of $C_{2,k}$, we have

$$C_{2,k} = \underbrace{2\zeta_{k+1}\sqrt{K_k^2 + d\sigma_k^2 + \varepsilon_S^2} + \left(\zeta_k^2 + \sigma_{F,k}^2 + \sigma_k\sqrt{d}\sqrt{\zeta_k^2 + \sigma_{F,k}^2}\right)}_{\mathcal{C}_{err,k+1}} + \underbrace{\frac{(2\zeta_{k+1} + \zeta_k + 2C_\alpha\alpha)^2}{4C_1}}_{\frac{B_k^2}{4C_1}} \tag{125}$$

As $k \to \infty$, we have $\zeta_k \to \zeta_\infty$, $\sigma_{F,k}^2 \to \sigma_{F,\infty}^2$, and $\sigma_k^2 \to \sigma_\infty^2$. This implies the limits of the coefficients are:

$$\lim_{k\to\infty}\mathcal{C}_{err,k+1} = 2\zeta_\infty\sqrt{K_k^2 + d\sigma_\infty^2 + \varepsilon_S^2} + \left(\zeta_\infty^2 + \sigma_{F,\infty}^2 + \sigma_\infty\sqrt{d}\sqrt{\zeta_\infty^2 + \sigma_{F,\infty}^2}\right)$$
$$= \mathcal{O}(\zeta_\infty^2 + \sigma_{F,\infty}^2)$$
$$\lim_{k\to\infty}\frac{B_k^2}{4C_1} = \frac{(2\zeta_\infty + \zeta_\infty + 4C_1\zeta_\infty)^2}{4C_1} = \mathcal{O}(\zeta_\infty^2) \tag{126}$$

For the upper bound in (124), we use a fundamental result from analysis: if a sequence $x_k$ converges to a limit $L$, then its Cesàro mean (average) $\frac{1}{T}\sum x_k$ also converges to $L$. Since we established that $C_{2,k}$ converges to $C_{2,\infty}$. Using (126), we have

$$\lim_{T\to\infty}\sup\frac{1}{T}\sum_{k=0}^{T-1}\|\mathbf{g}_k\|^2 \leq \mathcal{O}(\zeta_\infty^2 + \sigma_{F,\infty}^2). \tag{127}$$

Now, we need to show that for the hybrid step-size schedule in Assumption 1, the aggregation weight $\mathbf{w}_{k+1}$ converges to $\mathbf{w}_k$ as $k \to \infty$.

**Lemma 13.** *Consider the inequality* (122) *with the hybrid step-size schedule in Assumption 1. The expected weight updates will vanish:*

$$\lim_{k\to\infty}\mathbb{E}[\|\mathbf{w}_{k+1} - \mathbf{w}_k\|^2] = 0.$$

*Proof by Contradiction.* Let $x_k := \mathbb{E}[\|\mathbf{w}_{k+1} - \mathbf{w}_k\|^2]$. We begin from (122) with neglecting non-negative term $C_1\alpha\sum_{k=0}^{T-1}\|\mathbf{g}_k\|^2$ from left hand side of inequality and using (123):

$$\sum_{k=0}^{T-1}\left(\frac{1}{2\beta_{k+1}} - \frac{L_w}{2}\right)x_k \leq Q_0 - Q^\star + \sum_{k=0}^{T-1}C_{2,k}\alpha. \tag{128}$$

Assume for contradiction that $\{x_k\}$ does not converge to 0. This implies the existence of a constant $\epsilon > 0$ and an infinite set of indices $K$ such that $x_k \geq \epsilon$ for all $k \in K$ with $K_T = K \cap \{0, \ldots, T-1\}$. Following the standard contradiction argument, this leads to the inequality:

$$\epsilon\sum_{k\in K_T}\left(\frac{1}{2\beta_{k+1}} - \frac{L_w}{2}\right) \leq Q_0 - Q^\star + \alpha\sum_{k=0}^{T-1}C_{2,k}.$$

Taking the limit of the above inequality when $T \to \infty$ give us:

$$\epsilon \lim_{T \to \infty} \frac{1}{T} \sum_{k \in K_T} \left( \frac{1}{2\beta_{k+1}} - \frac{L_w}{2} \right) \leq \lim_{T \to \infty} \frac{Q_0 - Q^\star}{T} + \alpha \lim_{T \to \infty} \frac{1}{T} \sum_{k=0}^{T-1} C_{2,k}.$$

The first term on the right-hand side of the above inequality converges to zero as $T \to \infty$ because $Q_0 - Q^\star$ is bounded. The second term is a Cesàro mean of the sequence $\{C_{2,k}\}$, and as shown converges to $C_{2,\infty}$. The key step is to analyze the asymptotic growth of both sides. So, we have

$$\epsilon \cdot \lim_{T \to \infty} \frac{1}{T} \sum_{k \in K_T} \left( \frac{1}{2\beta_{k+1}} - \frac{L_w}{2} \right) \leq \alpha C_{2,\infty}.$$

Since $\beta_{k+1} \to 0$, the terms $(1/(2\beta_{k+1}) - L_w/2) \to \infty$. The Cesàro mean (average) of a sequence that diverges to infinity also diverges to infinity. Thus, the left-hand side is infinite, leading to the contradiction $\infty \leq \alpha C_{2,\infty}$. The assumption must be false, and therefore $\lim_{k \to \infty} x_k = 0$. This concludes the proof of Lemma 13. □

Now, we need to show that the aggregation weight $\mathbf{w}_k$ converges to the critical point of (6) as $k \to \infty$. As we know $\lim_{k \to \infty} \mathbf{w}_k = \mathbf{w}^\star$, which is a fixed point. Using (8), we have

$$\mathbf{w}^\star = \operatorname*{argmin}_{\mathbf{w}} \Phi_\infty(\mathbf{w}^\star) + \langle \nabla \Phi_\infty(\mathbf{w}^\star), \mathbf{w} - \mathbf{w}^\star \rangle + \frac{1}{2\beta_\infty} \|\mathbf{w} - \mathbf{w}^\star\|_2^2 + \delta_{\Delta_{t,\ell_0}^+}(\mathbf{w}) \tag{129}$$

Using the optimality condition for the above equation gives us

$$0 \in \nabla \Phi_\infty(\mathbf{w}^\star) + \frac{1}{\beta_\infty}(\mathbf{w}^\star - \mathbf{w}^\star) + \partial \delta_{\Delta_{t,\ell_0}^+}(\mathbf{w}^\star) \tag{130}$$

where $\partial$ denotes subgradient. As $k \to \infty$, $\beta_\infty \to 0$, so $\frac{1}{\beta_\infty}(\mathbf{w}^\star - \mathbf{w}^\star) = 0$. Using this point results in

$$0 \in \nabla \Phi_k(\mathbf{w}^\star) + \partial \delta_{\Delta_{t,\ell_0}^+}(\mathbf{w}^\star), \tag{131}$$

which is exactly the optimality condition for (6). Since this represents the optimal solution of (6) as $k \to \infty$, it also satisfies the Byzantine resilience bound (14) in the limit. Hence, we conclude that $\zeta_\infty \leq \eta$.

This concludes the proof of Theorem 9. □

## H.2 PROOF OF THEOREM 3 FOR L-SMOOTH AND STRONGLY CONVEX (ITEM 2)

We extend the previous analysis to the case where each loss function $f_i(\boldsymbol{\theta})$ is $\mu$-strongly convex.

**Theorem 14.** *Consider the cost function* (1) *under the assumptions of Theorem 3 and Item 2. The sequence* $\{\boldsymbol{\theta}_k, \mathbf{w}_k\}_{k=1}^\infty$ *generated by Algorithm 2 satisfies the corresponding result of Item 2 of Theorem 3.*

*Proof.* To prove the above theorem, since $f_i(\boldsymbol{\theta})$ is a $\mu$ strongly convex function, we need to show that the $Q(\boldsymbol{\theta}, \mathbf{w})$ is also $\mu$- strongly convex with respect to $\boldsymbol{\theta}$, which is formulated in the following lemma.

**Lemma 15.** *Let the function be* $Q(\boldsymbol{\theta}, \mathbf{w}) = \sum_{i=1}^n w_i f_i(\boldsymbol{\theta}) + \delta_{\Delta_{t,\ell_0}^+}(\mathbf{w})$. *If each* $f_i(\boldsymbol{\theta})$ *is* $\mu$-*strongly convex with respect to* $\boldsymbol{\theta}$ *and* $\mathbf{w}$ *is on the unit sparse capped simplex, then* $Q(\boldsymbol{\theta}, \mathbf{w})$ *is also* $\mu$-*strongly convex with respect to* $\boldsymbol{\theta}$.

*Proof.* The proof analyzes convexity with respect to $\boldsymbol{\theta}$ for a fixed, valid $\mathbf{w}$. The term $\delta_{\Delta_{t,\ell_0}^+}(\mathbf{w})$ is constant with respect to $\boldsymbol{\theta}$, and adding a constant does not affect the convexity or the strong convexity parameter of a function. Therefore, $Q(\boldsymbol{\theta}, \mathbf{w})$ is $\mu$-strongly convex with respect to $\boldsymbol{\theta}$ if the weighted sum $H(\boldsymbol{\theta}, \mathbf{w}) = \sum_{i=1}^n w_i f_i(\boldsymbol{\theta})$ is $\mu$-strongly convex.

We use the definition that a function $h(\boldsymbol{\theta})$ is $\mu$-strongly convex if $h(\boldsymbol{\theta}) - \frac{\mu}{2}\|\boldsymbol{\theta}\|^2$ is convex. We analyze this for $H(\boldsymbol{\theta}, \mathbf{w})$:

$$H(\boldsymbol{\theta}, \mathbf{w}) - \frac{\mu}{2}\|\boldsymbol{\theta}\|^2 = \left(\sum_{i=1}^{n} w_i f_i(\boldsymbol{\theta})\right) - \left(\sum_{i=1}^{n} w_i\right)\frac{\mu}{2}\|\boldsymbol{\theta}\|^2$$

$$= \sum_{i=1}^{n} w_i \left(f_i(\boldsymbol{\theta}) - \frac{\mu}{2}\|\boldsymbol{\theta}\|^2\right).$$

By assumption, each function $f_i(\boldsymbol{\theta})$ is $\mu$-strongly convex, so each term $(f_i(\boldsymbol{\theta}) - \frac{\mu}{2}\|\boldsymbol{\theta}\|^2)$ is convex. Since the weights $w_i \geq 0$, the expression above is a non-negative weighted sum of convex functions, which is itself a convex function.

Thus, $H(\boldsymbol{\theta}, \mathbf{w})$ is $\mu$-strongly convex. As established, this implies that $Q(\boldsymbol{\theta}, \mathbf{w})$ is also $\mu$-strongly convex with respect to $\boldsymbol{\theta}$. $\qquad\square$

To continue the proof, we define:

$$F(\boldsymbol{\theta}_k, \mathbf{w}_k) = \sum_{i=1}^{n} w_{k,i} f_i(\boldsymbol{\theta}_k) + \delta_{\Delta_{t,\ell_0}^+}(\mathbf{w}_k), \quad \nabla_\theta F(\boldsymbol{\theta}_k, \mathbf{w}_k) = \sum_{i=1}^{n} w_{k,i}\nabla_\theta f_i(\boldsymbol{\theta}_k) \qquad (132)$$

in which $f_i(\boldsymbol{\theta}_k)$ and $\nabla_\theta f_i(\boldsymbol{\theta}_k)$ for all $1 \leq i \leq n$ denote the honest losses and gradients. Additionally, according to Lemma 15, $F(\boldsymbol{\theta}, \mathbf{w})$ is $\mu$-strongly convex with respect to $\boldsymbol{\theta}$.

Next, we need to determine the variance of the global true gradients, which is formalized in the following lemma. Furthermore, as mentioned, we exclude $b_f$ clients, so in our algorithm we have $t = 1/s$ and $s = n - b_f$.

**Lemma 16.** *Under Assumption 1, the variance of the global true gradient is bounded:*
$$\sigma_{g,k}^2 := var(\nabla_\theta F(\boldsymbol{\theta}_k, \mathbf{w}_k)) = \mathbb{E}\left[\|\nabla_\theta F(\boldsymbol{\theta}_k, \mathbf{w}_k) - \mathbf{g}_k\|_2^2\right] \leq d\sigma_k^2.$$

*Proof.* We start from the definition of $\sigma_{g,k}^2$ and substitute the definition of $F(\boldsymbol{\theta}_k, \mathbf{w}_k)$:

$$\sigma_g^2 = \mathbb{E}\left[\|\nabla\left(\sum_{i=1}^{n} w_{k,i} f_i(\boldsymbol{\theta}_k)\right) - \mathbb{E}\left[\nabla\left(\sum_{i=1}^{n} w_{k,i} f_i(\boldsymbol{\theta}_k)\right)\right]\|_2^2\right]$$

$$= \mathbb{E}\left[\|\sum_{i=1}^{n} w_{k,i}\nabla f_i(\boldsymbol{\theta}_k) - \sum_{i=1}^{n} w_{k,i}\mathbb{E}[\nabla f_i(\boldsymbol{\theta}_k)]\|_2^2\right] \qquad \text{by linearity of } \nabla, \mathbb{E}$$

$$= \mathbb{E}\left[\|\sum_{i=1}^{n} w_{k,i}\left(\nabla f_i(\boldsymbol{\theta}_k) - \mathbb{E}[\nabla f_i(\boldsymbol{\theta}_k)]\right)\|_2^2\right].$$

We use the inequality $\|\sum_{i=1}^{N} \mathbf{x}_i\|_2^2 \leq N\sum_{i=1}^{N}\|\mathbf{x}_i\|_2^2$ and the property capped in $\Delta_{t,\ell_0}^+$, i.e. $w_{k,i}^2 \leq t^2$. Here, $N \leq s$.

$$\sigma_{g,k}^2 \leq st^2\mathbb{E}\left[\sum_{i\in\text{supp}(\mathbf{w}_k)} \|\nabla f_i(\boldsymbol{\theta}_k) - \mathbb{E}[\nabla f_i(\boldsymbol{\theta}_k)]\|_2^2\right]$$

$$= st^2 \sum_{i\in\text{supp}(\mathbf{w}_k)} \mathbb{E}\left[\|\nabla f_i(\boldsymbol{\theta}_k) - \mathbb{E}[\nabla f_i(\boldsymbol{\theta}_k)]\|_2^2\right] \qquad \text{by linearity of } \mathbb{E}$$

$$= st^2 \sum_{i\in\text{supp}(\mathbf{w}_k)} var(\nabla f_i(\boldsymbol{\theta}_k)).$$

Now, we apply the given bound $var(\nabla f_i(\boldsymbol{\theta}_k)) \leq d\sigma_k^2$:

$$\sigma_{g,k}^2 \leq st^2 \sum_{i\in\text{supp}(\mathbf{w}_k)} (d\sigma_k^2)$$

$$\leq (st)^2 d\sigma_k^2.$$

Replacing $s = n - b_f$ and $t = 1/s$ gives us the proof of Lemma 16. $\qquad\square$

Since $F(\boldsymbol{\theta}, \mathbf{w})$ in (132) is $\mu$-strongly function with respect to $\boldsymbol{\theta}$, applying this to generated sequence $\{\boldsymbol{\theta}_k, \mathbf{w}_k\}$ by Algorithm 2 yields to

$$2\mu(F(\boldsymbol{\theta}_k, \mathbf{w}_k) - F(\boldsymbol{\theta}^\star, \mathbf{w}_k)) \leq \|\nabla_\theta F(\boldsymbol{\theta}_k, \mathbf{w}_k)\|_2^2.$$

Since the adversary just replaces the honest gradients with Byzantine gradients, so $F(\boldsymbol{\theta}_k, \mathbf{w}_k) = Q(\boldsymbol{\theta}_k, \mathbf{w}_k)$. Using this point and taking the expectation from both side of the above inequality yields to

$$2\mu \cdot \mathbb{E}\{Q(\boldsymbol{\theta}_k, \mathbf{w}_k) - Q(\boldsymbol{\theta}^\star, \mathbf{w}_k)\} \leq \mathbb{E}\left[\|\nabla_\theta F(\boldsymbol{\theta}_k, \mathbf{w}_k)\|_2^2\right]. \tag{133}$$

The term $\mathbb{E}[\|\nabla_\theta F(\boldsymbol{\theta}_k, \mathbf{w}_k)\|_2^2]$ can be written

$$\begin{aligned}
\mathbb{E}\left[\|\nabla_\theta F(\boldsymbol{\theta}_k, \mathbf{w}_k)\|_2^2\right] &= \|\mathbb{E}[\nabla_\theta F(\boldsymbol{\theta}_k, \mathbf{w}_k)]\|_2^2 + \mathbb{E}\left[\|\nabla_\theta F(\boldsymbol{\theta}_k, \mathbf{w}_k) - \mathbb{E}[\nabla_\theta F(\boldsymbol{\theta}_k, \mathbf{w}_k)]\|_2^2\right] \\
&= \|\mathbf{g}_k\|_2^2 + var(\nabla_\theta F(\boldsymbol{\theta}_k, \mathbf{w}_k)) \\
&= \|\mathbf{g}_k\|_2^2 + \sigma_{g,k}^2. \tag{134}
\end{aligned}$$

Substituting the decomposition from (134) back into our main inequality (133):

$$2\mu\mathbb{E}\{Q(\boldsymbol{\theta}_k, \mathbf{w}_k) - Q(\boldsymbol{\theta}^\star, \mathbf{w}_k)\} \leq \|\mathbf{g}_k\|_2^2 + \sigma_{g,k}^2.$$

Using Lemma 16 to bound $\sigma_g^2 \leq d\sigma_k^2$, we get a lower bound on $\|\mathbf{g}_k\|_2$:

$$2\mu\mathbb{E}\{Q(\boldsymbol{\theta}_k, \mathbf{w}_k) - Q(\boldsymbol{\theta}^\star, \mathbf{w}_k)\} - d\sigma_k^2 \leq \|\mathbf{g}_k\|_2^2. \tag{135}$$

According to (121), by neglecting the negative term $-\sum_{k=0}^{T-1} C_{\beta_{k+1}} \mathbb{E}\{\|\mathbf{w}_{k+1} - \mathbf{w}_k\|_2^2\}$ and replacing $\|\mathbf{g}_k\|_2^2$ by the lower bound (135), we have

$$E_{k+1} \leq (1-a)E_k + ad\sigma_k^2 + a\mathbb{E}\{Q^\star(\mathbf{w}_k) - Q^\star\} + C_{2,k}\alpha. \tag{136}$$

where $E_{k+1} = \mathbb{E}\{Q(\boldsymbol{\theta}_{k+1}, \mathbf{w}_{k+1}) - Q^\star\}$ and $E_k = \mathbb{E}\{Q(\boldsymbol{\theta}_k, \mathbf{w}_k) - Q^\star\}$ where $Q^\star = Q(\boldsymbol{\theta}^\star, \mathbf{w}^\star)$ and $\mathbf{w}^\star$ be a limit point of the sequence, i.e. $\lim_{k\to\infty} \mathbf{w}_k = \mathbf{w}^*$

$$\boldsymbol{\theta}^\star = \underset{\boldsymbol{\theta}}{\arg\min}\, F(\boldsymbol{\theta}, \mathbf{w}^*) \qquad\qquad \text{by using (132)}$$

This allows us to analyze the asymptotic behavior of the recurrence in (136). The inequality is a standard form of a Robbins-Siegmund-type lemma, $E_{k+1} \leq (1-a)E_k + b_k$, where:

- $a = 2\mu C_1 \alpha$.

- $b_k = a\left(d\sigma_k^2 + \mathbb{E}\{Q^\star(\mathbf{w}_k) - Q^\star\}\right) + C_{2,k}\alpha$.

The convergence of such a sequence is determined by the asymptotic behavior of the error term $b_k$ relative to the descent term $a$. Specifically, we analyze the limit of their ratio:

$$\lim_{k\to\infty} \frac{b_k}{a} = \lim_{k\to\infty} \left(d\sigma_k^2 + \mathbb{E}\{Q^\star(\mathbf{w}_k) - Q^\star\} + \frac{C_{2,k}}{2\mu C_1}\right).$$

From Lemma 13, we have established $\mathbf{w}_{k+1} \to \mathbf{w}_k$ with limit point $\mathbf{w}^\star$ and using the continuity of $Q$, we have $\lim_{k\to\infty} \mathbb{E}\{Q^\star(\mathbf{w}_k) - Q^\star\} = 0$. Consequently,

$$\limsup_{k\to\infty} \frac{b_k}{a} = d\sigma_\infty^2 + \frac{C_{2,\infty}}{2\mu C_1}.$$

In this case, the optimization error converges to a neighborhood of the optimum, with the size of the error ball given by this limit:

$$\limsup_{k\to\infty} E_k \leq d\sigma_\infty^2 + \frac{C_{2,\infty}}{2\mu C_1} = \mathcal{O}\left(\zeta_\infty^2 + \sigma_{F,\infty}^2 + \sigma_\infty^2\right).$$

This completes the proof of Lemma 14. $\qquad\square$

## H.3 DETERMINING LIPSCHITZ CONSTANT $L_w$

In the following lemma, we derive a bound for the Lipschitz constant $L_w$, assuming $\nabla_\theta f_i$ is Lipschitz continuous and $\|\nabla_\theta f_i(\boldsymbol{\theta})\|_2 \leq C$ for all $\boldsymbol{\theta}$.

**Lemma 17.** *Let $f_i : \mathbb{R}^d \to \mathbb{R}$ be continuously differentiable with $\nabla_\theta f_i$ being $L_i$-Lipschitz continuous and bounded, $\|\nabla_\theta f_i(\boldsymbol{\theta})\| \leq C$ for all $\boldsymbol{\theta} \in \mathbb{R}^d$ and $i = 1, \ldots, n$. Define:*

$$h(\mathbf{w}) = \sum_{i=1}^n w_i f_i \left( \boldsymbol{\theta} - \alpha \sum_{j=1}^n w_j \nabla_\theta f_j(\boldsymbol{\theta}) \right), \tag{137}$$

*for $\mathbf{w} \in \Delta_{t,\ell_0}^+ = \{\mathbf{w} \in \mathbb{R}^n : \sum_i w_i = 1, w_i \geq 0, w_i \leq t, \|\mathbf{w}\|_0 \leq s\}$, $\alpha > 0$, and fixed $\boldsymbol{\theta}$. Then, $\nabla_w h(\mathbf{w})$ is $L_w$-Lipschitz continuous with:*

$$L_w \leq \alpha C^2(n^{3/2} + n + \alpha n L_{\max} + \alpha n^2 \frac{L_{\max}\varrho}{2}), \tag{138}$$

*where $L_{\max} = \max\limits_{i=1,\ldots,n} L_i$, and $\varrho = \sqrt{2(kt^2 + r^2)}$ as defined in Equation (21) of Lemma 5.*

*Proof.* Define $\mathbf{z}(\mathbf{w}) = \boldsymbol{\theta} - \alpha\mathbf{G}\mathbf{w}$, where $\mathbf{G} = [\nabla_\theta f_1(\boldsymbol{\theta}), \ldots, \nabla_\theta f_n(\boldsymbol{\theta})] \in \mathbb{R}^{d \times n}$. Then:

$$h(\mathbf{w}) = \sum_{i=1}^n w_i f_i(\mathbf{z}(\mathbf{w})).$$

The gradient is:
$$\nabla_w h(\mathbf{w}) = \mathbf{f}(\mathbf{z}) - \alpha\mathbf{G}^T \tilde{\mathbf{G}}(\mathbf{w})\mathbf{w},$$

where $\mathbf{f}(\mathbf{z}) = [f_1(\mathbf{z}), \ldots, f_n(\mathbf{z})]^T$, and $\tilde{\mathbf{G}}(\mathbf{w}) = [\nabla_\theta f_1(\mathbf{z}), \ldots, \nabla_\theta f_n(\mathbf{z})] \in \mathbb{R}^{d \times n}$. We need to show that:
$$\|\nabla_w h(\mathbf{w}_2) - \nabla_w h(\mathbf{w}_1)\|_2 \leq L_w \|\mathbf{w}_2 - \mathbf{w}_1\|_2.$$

We compute
$$\nabla_w h(\mathbf{w}_2) - \nabla_w h(\mathbf{w}_1) = \mathbf{f}(\mathbf{z}_2) - \mathbf{f}(\mathbf{z}_1) - \alpha\mathbf{G}^T(\tilde{\mathbf{G}}(\mathbf{w}_2)\mathbf{w}_2 - \tilde{\mathbf{G}}(\mathbf{w}_1)\mathbf{w}_1),$$

where $\mathbf{z}_j = \boldsymbol{\theta} - \alpha\mathbf{G}\mathbf{w}_j$, $j = 1, 2$. Thus:

$$\|\nabla_w h(\mathbf{w}_2) - \nabla_w h(\mathbf{w}_1)\|_2 \leq \|\mathbf{f}(\mathbf{z}_2) - \mathbf{f}(\mathbf{z}_1)\|_2 + \alpha\|\mathbf{G}^T(\tilde{\mathbf{G}}(\mathbf{w}_2)\mathbf{w}_2 - \tilde{\mathbf{G}}(\mathbf{w}_1)\mathbf{w}_1)\|_2. \tag{139}$$

**First Term** ($\|\mathbf{f}(\mathbf{z}_2) - \mathbf{f}(\mathbf{z}_1)\|_2$): Since $f_i$ is $L_i$-smooth, the descent lemma gives:

$$f_i(\mathbf{z}_2) - f_i(\mathbf{z}_1) \leq \nabla_\theta f_i(\mathbf{z}_1)^T (\mathbf{z}_2 - \mathbf{z}_1) + \frac{L_i}{2}\|\mathbf{z}_2 - \mathbf{z}_1\|^2. \tag{140}$$

Since $\mathbf{z}_2 - \mathbf{z}_1 = \alpha\mathbf{G}(\mathbf{w}_1 - \mathbf{w}_2)$, and $\|\mathbf{G}\|_F \leq \sqrt{n}C$ (as $\|\nabla_\theta f_i(\boldsymbol{\theta})\| \leq C$):
$$\|\mathbf{z}_2 - \mathbf{z}_1\| \leq \alpha\sqrt{n}C\|\mathbf{w}_2 - \mathbf{w}_1\|_2.$$

Similarly, we can write

$$f_i(\mathbf{z}_1) - f_i(\mathbf{z}_2) \leq \nabla_\theta f_i(\mathbf{z}_2)^T (\mathbf{z}_1 - \mathbf{z}_2) + \frac{L_i}{2}\|\mathbf{z}_2 - \mathbf{z}_1\|_2^2. \tag{141}$$

Since $\|\nabla_\theta f_i(\mathbf{z}_1)\|_2 \leq C$, using (140), (141), and the result (21) in Lemma 5 $\|\mathbf{w}_2 - \mathbf{w}_1\|_2 \leq \varrho$ where $\varrho = \sqrt{2(kt^2 + r^2)} \leq \sqrt{2}$:

$$|f_i(\mathbf{z}_2) - f_i(\mathbf{z}_1)| \leq \alpha(\sqrt{n} + \frac{L_i\varrho}{2}n\alpha)C^2\|\mathbf{w}_2 - \mathbf{w}_1\|_2.$$

Thus:

$$\|\mathbf{f}(\mathbf{z}_2) - \mathbf{f}(\mathbf{z}_1)\|_2 \leq \sum_{i=1}^n |f_i(\mathbf{z}_2) - f_i(\mathbf{z}_1)| \leq \alpha \sum_{i=1}^n (\sqrt{n} + \frac{L_i\varrho}{2}n\alpha)C^2\|\mathbf{w}_2 - \mathbf{w}_1\|_2 \leq$$

$$\alpha(n^{3/2} + n^2\alpha\frac{L_{\max}\varrho}{2})C^2\|\mathbf{w}_2 - \mathbf{w}_1\|_2. \tag{142}$$

**Second Term** ($\|\mathbf{G}^T(\tilde{\mathbf{G}}(\mathbf{w}_2)\mathbf{w}_2 - \tilde{\mathbf{G}}(\mathbf{w}_1)\mathbf{w}_1)\|_2$):

$$\tilde{\mathbf{G}}(\mathbf{w}_2)\mathbf{w}_2 - \tilde{\mathbf{G}}(\mathbf{w}_1)\mathbf{w}_1 = \sum_{i=1}^{n} w_{i,2}\left(\nabla_\theta f_i(\mathbf{z}_2) - \nabla_\theta f_i(\mathbf{z}_1)\right) + \sum_{i=1}^{n}(w_{i,2} - w_{i,1})\nabla_\theta f_i(\mathbf{z}_1).$$

Using the Lipschitz property of $\nabla_\theta f_i$

$$\|\nabla_\theta f_i(\mathbf{z}_2) - \nabla_\theta f_i(\mathbf{z}_1)\|_2 \leq L_i\|\mathbf{z}_2 - \mathbf{z}_1\|_2 \leq L_i\alpha\sqrt{n}C\|\mathbf{w}_2 - \mathbf{w}_1\|_2.$$

Utilizing the property $\sum_{i=1}^{n} w_{i,2}L_i \leq L_{\max}\sum_{i=1}^{n} w_{i,2} = L_{\max}$

$$\left\|\sum_{i=1}^{n} w_{i,2}\left(\nabla_\theta f_i(\mathbf{z}_2) - \nabla_\theta f_i(\mathbf{z}_1)\right)\right\|_2 \leq \sum_{i=1}^{n} w_{i,2}L_i\alpha\sqrt{n}C\|\mathbf{w}_2 - \mathbf{w}_1\|_2 \leq \alpha L_{\max}\sqrt{n}C\|\mathbf{w}_2 - \mathbf{w}_1\|_2. \tag{143}$$

Also, we have

$$\left\|\sum_{i=1}^{n}(w_{i,2} - w_{i,1})\nabla_\theta f_i(\mathbf{z}_1)\right\|_2 \leq \sum_{i=1}^{n}|w_{i,2} - w_{i,1}|C \leq C\sqrt{n}\|\mathbf{w}_2 - \mathbf{w}_1\|_2. \tag{144}$$

Combining (144) and (143) gives us

$$\|\mathbf{G}^T(\tilde{\mathbf{G}}(\mathbf{w}_2)\mathbf{w}_2 - \tilde{\mathbf{G}}(\mathbf{w}_1)\mathbf{w}_1)\|_2 \leq \|\mathbf{G}\|_F(\alpha L_{\max}\sqrt{n}C + C\sqrt{n})\|\mathbf{w}_2 - \mathbf{w}_1\|_2 \leq$$
$$(\alpha L_{\max}nC^2 + C^2 n)\|\mathbf{w}_2 - \mathbf{w}_1\|_2.$$

So, we have

$$\alpha\|\mathbf{G}^T(\tilde{\mathbf{G}}(\mathbf{w}_2)\mathbf{w}_2 - \tilde{\mathbf{G}}(\mathbf{w}_1)\mathbf{w}_1)\|_2 \leq \alpha(\alpha L_{\max}nC^2 + C^2 n)\|\mathbf{w}_2 - \mathbf{w}_1\|_2. \tag{145}$$

Replacing (142) and (145) in (139):

$$L_w \leq \alpha C^2\left(n^{3/2} + n + \alpha n L_{\max} + \alpha n^2 \frac{L_{\max}\varrho}{2}\right). \tag{146}$$

Thus, $\nabla_w h(\mathbf{w})$ is $L_w$-Lipschitz continuous. $\qquad\square$

The Lipschitz constant $L_w$ can be determined in two ways. A more precise approach is to compute $L_w$ directly by analyzing the gradient of $h(\mathbf{w})$ defined in (137), though this may be computationally complex. Alternatively, when direct computation is challenging, $L_w$ can be bounded as in (146) under the assumptions of Lemma 17.

## I   ADDITIONAL EXPERIMENTAL SETUPS AND RESULTS

### I.1   EXPERIMENTAL FRAMEWORK OVERVIEW

**Implementation Details**: We implemented our federated learning framework using Python 3.12.7 and PyTorch as the primary deep learning library. All experiments were conducted on GPU-accelerated hardware to ensure efficient training and evaluation. Specifically, we utilized CUDA for GPU support and ran our models on a variety of high-performance GPUs, including NVIDIA A100, NVIDIA A40, and Tesla V100-SXM2-32GB. These resources allowed for large-scale parallelization and significantly reduced computation time during training. For further implementation details, we have shared the complete source code.

**Data Heterogeneity and Malicious clients**: In this study, we utilize two datasets, MNIST and CIFAR10, both of which contain ten labels. As described in Subsection 5.1, the number of groups corresponds to the number of labels, which is ten. A training example with label $l$ is assigned to group $l$ with probability $q$, and to other groups with probability $\frac{1-q}{9}$. Each group consists of a subset of clients, and since this study involves 200 clients divided into ten groups, each group contains 20 clients.

For selecting malicious clients, we adopt a group-oriented approach. Specifically, we randomly select $n_{gm} = \lceil \frac{\text{number of malicious}}{\text{number of clients per group}} \rceil$ groups. Malicious clients are first chosen from within a single group.

If there are remaining malicious clients to be assigned, we select them from other groups, repeating this process until all malicious clients have been selected.

For example, for a fraction of malicious clients $0.3$ in this study, the number of malicious clients is 60. Since the number of clients per group is 20, the malicious clients are selected from three random groups.

Now, a question may arise: why is this methodology employed instead of randomly selecting malicious clients? In fact, this methodology is a specific case of random selection and represents one of the most difficult and challenging cases. Assume the fraction of malicious clients is $0.3$ and the selected random groups are $i$, $j$, and $k$ ($i \neq j \neq k$). It is straightforward to show that the attack corrupts all datasets with labels $i$, $j$, and $k$ with probability $q + 2\frac{1-q}{9} = \frac{2}{9} + \frac{7q}{9}$, which can be a high probability, especially for non-IID data with a high degree of non-IIDness, as considered in the numerical study ($q = \{0.6, 0.9\}$).

In this case, if we cannot detect the malicious clients, they can significantly reduce the test accuracy. In contrast, if the malicious clients are selected randomly, they may be distributed among all groups (for example, uniform selection). In this scenario, the attack may have a minor effect because the benign data dominates the malicious data.

Furthermore, we examine the effect of group-oriented malicious selection compared to random selection on FedAvg. We observed that the test accuracy of FedAvg drops significantly for group-oriented selection compared to random selection. In summary, this explanation demonstrates that group-oriented selection is one of the most difficult and challenging scenarios for detecting malicious clients. This allows us to compare the proposed method to state-of-the-art Byzantine-robust FL approaches in a highly challenging setting.

**Attacks**:

- **Flipping label**: Malicious clients train the model using a poisoned dataset, where the label of each class $l$ is changed to $L - l - 1$, where $L$ represents the total number of labels (in our study $L$ is 10).

- **Backdoor attack**: Malicious clients train the model using a poisoned dataset, where a black square of size $8 \times 8$ pixels is added to the center of the image, and its label is randomly changed to a label between 0 and $L - 1$.

- **Inverse gradient**: Malicious clients compute gradients based on their local datasets to minimize the loss function and then flip the sign of their gradients.

- **Global parameter attack**: At each round, the server sends the global parameters to the clients. Malicious clients add Gaussian noise, $\mathcal{N}(\nu_1 \mu, \nu_2 \sigma^2)$, into the global parameters $\boldsymbol{\theta}$, where $\mu$ represents the mean of the global parameters, $\sigma$ denotes their standard deviation, $\nu_1$ and $\nu_2 > 0$ are arbitrary real-valued constants. In this study, we set $\nu_1 = -5$ and $\nu_2 = 1.5$.

- **Double attack**: In this scenario, during a communication round, an attack targets a fraction of clients. In a subsequent communication round, a different attack affects a separate set of clients that were not impacted by the initial attack. Specifically, this study assumes that the first attack, an inverse gradient, takes place during the second communication round, targeting 50% of the malicious clients. Later, in the fifth communication round, a global random parameter attack is executed, affecting the remaining 50% of malicious clients who were not involved in the first attack. For instance, if the proportion of malicious clients is 40%, then 20% of the clients are impacted by the inverse gradient attack, while the random parameter attack targets the other 20%.

- **LIE (Little Is Enough) attack**: Malicious clients first compute the coordinate-wise mean $\mu_k$ and standard deviation $\sigma_k$ of the client updates in round $k$. They then submit forged updates of the form

$$\mathbf{b}_{k,j} = \mu_k + z\,\sigma_k, \quad \forall j \in \mathcal{H}^{\complement},$$

where $z > 0$ is a small constant ensuring that the forged updates remain within a plausible range. In our experiments, we use the original stealth bound for $z$ as proposed in Baruch et al. (2019).

## I.2 HYPERPARAMETER SETTINGS

We detail the hyperparameter configurations used for all state-of-the-art methods and our proposed algorithm. For Krum, the retained clients were set to $(1 - \text{fraction of malicious}) \times \text{total clients} - 2$, and for Trimmed Mean, the trimming fraction matched the fraction of malicious clients. All methods with tunable hyperparameters (e.g., Bulyan, FedLAW, CClip, Huber) were tuned via grid search using a dedicated validation split (80 % for training, 10 % for validation, and 10 % for testing). The chosen values corresponded to the settings that achieved the highest validation accuracy after 200 communication rounds on MNIST and 400 rounds on CIFAR-10. All methods shared a common training configuration with the learning rate $\alpha = 0.01$, a batch size of 64 for the MNIST dataset and 16 for CIFAR-10, and 3 local epochs. The total number of communication rounds used to update the global model parameters was 200 for MNIST and 400 for CIFAR-10. An overview of the selected hyperparameters for each method is summarized in Table 1.

Table 1: Selected hyper-parameters for all defences.

| Method | Hyper-parameter(s) | Chosen value |
|---|---|---|
| Krum | Retained clients | $(1 - \text{fraction of malicious}) \times n - 2$ |
| Trimmed Mean | Trimming fraction | fraction of malicious |
| Bulyan | Candidate pool size
Inner aggregation size | grid: 20/40/50
$(1 - \text{fraction of malicious}) \times n - 2$ |
| FedLAW | $\beta$
Sparsity budget $s$
Client weight upper bound $t$ | grid: $1 \times 10^{-2} - 1 \times 10^{-4}$
$(1 - \text{fraction of malicious}) \times n$
$1/(s - 10)$ |
| CClip | Clipping radius $\tau$
Fixed-point iterations | grid: 0.1, 10
1 |
| CClip + Bucketing | Clipping radius $\tau$
Fixed-point iterations
Bucketing factor | grid: 0.1, 10
1
2 |
| RFA (Geometric Median) | Smoothing $\nu$
Iterations $R$ | $10^{-6}$
3 |
| RFA + Bucketing | Same as RFA
Bucketing factor | $\nu = 10^{-6}$, $R = 3$
2 |
| Huber | Threshold $\tau$ | grid: 0.12, 0.2 |
| Coordinate-wise Median | – | – |

**FedLAW: sensitivity to $\beta$.** We evaluated the impact of the FedLAW hyperparameter $\beta$ under the *flipping label* attack with a strongly non-IID split ($q = 0.9$) and 40 % malicious clients. The test accuracy (mean $\pm$ standard deviation across runs) is reported in Table 2; accuracy curves with error bars appear in Figs. 5 and 6.

For MNIST, accuracy remains consistently high across a wide range of $\beta$, staying above 86 % for all $\beta \geq 6.3 \times 10^{-4}$. Performance peaks around 87.7 % when $\beta = 10^{-2}$, but values between $6.3 \times 10^{-4}$ and $10^{-2}$ yield nearly indistinguishable results.

For CIFAR-10, accuracy generally stays in the 55–60% band, with some variance at a few values of $\beta$. The best mean accuracy of 59.7 % is achieved at $\beta = 10^{-2}$, and most settings in the $10^{-3} - 10^{-2}$ range perform comparably.

Overall, FedLAW is robust to the choice of $\beta$; near-optimal performance is obtained without extensive tuning, especially for $\beta$ in the $10^{-3} - 10^{-2}$ range.

**Bulyan: sensitivity to *inner aggregation size* and *candidate pool size*.** In our main experiments, we fix the inner aggregation size to $(1 - \text{fraction of malicious}) \times \text{total clients} - 2$ and tune only the candidate pool size. However, in Fig. 7, we sweep both hyperparameters to study their joint effect. Under the flipping label attack with 40 % malicious clients and $q=0.9$, we observe that *inner*

Table 2: FedLAW $\beta$-sweep with 40% flipping label attack ($q$=0.9). Reported values are mean test accuracy $\pm$ standard deviation across 5 runs.

| $\beta$ | MNIST Acc. (%) | CIFAR-10 Acc. (%) |
|---|---|---|
| $1 \times 10^{-2}$ | $87.67 \pm 0.97$ | $59.66 \pm 0.77$ |
| $6.3 \times 10^{-3}$ | $86.80 \pm 0.97$ | $57.83 \pm 1.80$ |
| $4.0 \times 10^{-3}$ | $86.54 \pm 0.46$ | $58.91 \pm 0.01$ |
| $1.6 \times 10^{-3}$ | $86.43 \pm 0.64$ | $59.12 \pm 0.42$ |
| $1.0 \times 10^{-3}$ | $86.27 \pm 2.69$ | $54.23 \pm 7.52$ |
| $6.3 \times 10^{-4}$ | $86.97 \pm 0.64$ | $58.03 \pm 0.24$ |
| $4.0 \times 10^{-4}$ | $83.59 \pm 5.31$ | $59.11 \pm 2.08$ |
| $1.0 \times 10^{-4}$ | $84.12 \pm 4.61$ | $55.17 \pm 0.16$ |
| $6.3 \times 10^{-5}$ | $82.49 \pm 4.40$ | $55.27 \pm 0.39$ |
| $4.0 \times 10^{-5}$ | $85.19 \pm 6.63$ | $55.50 \pm 1.43$ |
| $1.6 \times 10^{-5}$ | $83.67 \pm 5.00$ | $58.00 \pm 1.80$ |
| $1.0 \times 10^{-5}$ | $84.50 \pm 1.99$ | $53.90 \pm 4.44$ |

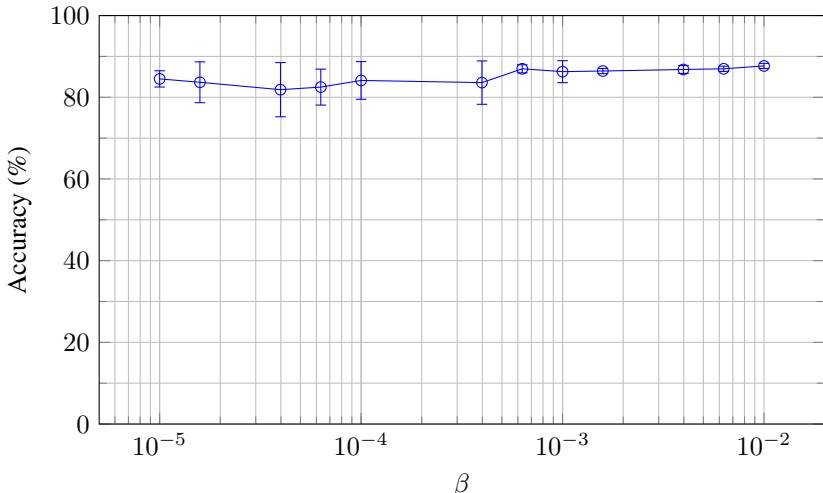

Figure 5: FedLAW sensitivity to $\beta$ on MNIST ($q$=0.9, 40% flipping label attack). Error bars denote $\pm1$ standard deviation across 5 runs.

*aggregation size* has limited impact, whereas performance is notably more sensitive to the *candidate pool size*. For the MNIST dataset, the best or near-best results are typically achieved when the candidate pool size is set to 40 or 50. It is important to note that these experiments violate Bulyan's theoretical guarantee, which requires the number of Byzantine clients $f$ to satisfy $f < (n - 3)/4$, where $n$ is the total number of clients (Guerraoui et al., 2018). Theoretical results guarantee that, under this assumption, the deviation of each aggregated coordinate from any honest one is bounded by $O(\sigma/\sqrt{d})$, where $\sigma$ is the variance among honest updates and $d$ is the model dimension. Since the assumption is violated in our setting, the theoretical bound does not formally apply. Nevertheless, as shown in Fig. 7, Bulyan still achieves strong empirical robustness in highly adversarial conditions.

**Choosing the Sparsity Level $s$.** In FedLAW, enforcing sparsity does not require exact knowledge of the number of Byzantine clients $b_f$. When an upper bound on $b_f$ is available, we set $s = n - b_f$ and choose the cap $t$ according to Lemma 5 with $t \geq 1/s$, selecting a specific value via validation. When $b_f$ is unknown, a practical strategy is to initialize $s = n$ and gradually reduce it across epochs, which implements a soft exclusion mechanism that avoids early over-pruning while still driving the weights of adversarial clients toward zero. In this case, $s$ can be treated as a tunable hyperparameter that controls the effective sparsity level and can be selected via cross-validation.

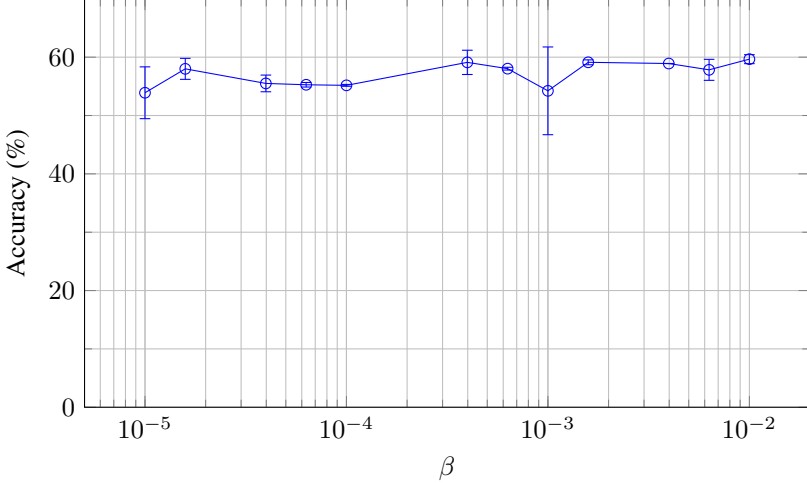

Figure 6: FedLAW sensitivity to $\beta$ on CIFAR-10 ($q=0.9$, 40% flipping label attack). Error bars denote ±1 standard deviation across 5 runs.

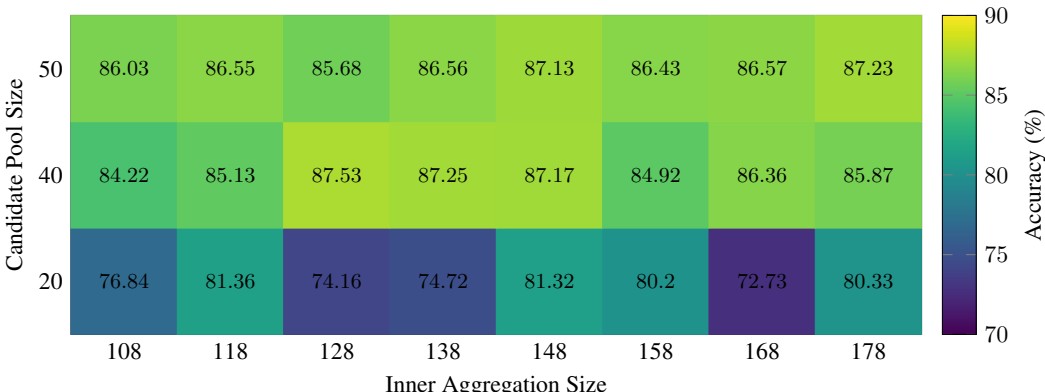

Figure 7: Bulyan sensitivity to inner aggregation size and candidate pool size on MNIST under the flipping label attack ($q=0.9$, 40 % malicious clients).

### I.3 FEDLAW: CLIENT-WEIGHT DYNAMICS UNDER FOUR ADVERSARIAL SETTINGS.

To investigate how FedLAW responds to various adversarial behaviors during training, we analyze the evolution of client aggregation weights in four distinct attack settings: *flipping label*, *inverse gradient*, *backdoor attack*, and a *double attack* scenario. We conduct the experiments on the MNIST dataset with a non-iid partitioning factor of $q = 0.9$ across 10 clients, of which 40% (i.e., 4 clients) are malicious. In the double attack setting, two adversaries apply inverse gradient manipulation while the other two send randomly perturbed global parameters. Each client trains a three-layer fully connected MLP using a batch size of 64 for three local epochs per round. The aggregation is performed using FedLAW with sparse weighting.

Figure 4 displays the per-client weight trajectories over 100 global training epochs. Grey curves denote benign clients, red curves indicate malicious clients executing single-strategy attacks, and in the double attack panel, blue curves represent global parameter attackers while red curves indicate inverse gradient attackers. Across all settings, FedLAW effectively distinguishes between benign and malicious behavior. Benign clients consistently receive stable, high weights, while malicious clients are rapidly down-weighted, either immediately (in flip-label and inverse gradient attacks) or gradually (in the backdoor and double attack settings). The results demonstrate FedLAW's ability to suppress diverse attack strategies in real-time during training.

### I.4 EVALUATION OF MALICIOUS CLIENT DETECTION

To comprehensively evaluate the performance of our method in detecting malicious clients, we report four standard classification metrics: *Precision*, *Recall*, *F1 Score*, and *Accuracy*. We define the counts of true positives (TP), false positives (FP), false negatives (FN), and true negatives (TN) using the following strategy, where $\mathcal{M}$ represents the set of malicious clients.

Formally, for each client $i$, we define the following indicator variables:

$$\text{TP}_i = \begin{cases} 1, & \text{if } i \in \mathcal{M} \text{ and } w_i \leq \varepsilon, \\ 0, & \text{otherwise}, \end{cases} \tag{147}$$

$$\text{FP}_i = \begin{cases} 1, & \text{if } i \notin \mathcal{M} \text{ and } w_i \leq \varepsilon, \\ 0, & \text{otherwise}, \end{cases} \tag{148}$$

$$\text{FN}_i = \begin{cases} 1, & \text{if } i \in \mathcal{M} \text{ and } w_i > \varepsilon, \\ 0, & \text{otherwise}, \end{cases} \tag{149}$$

$$\text{TN}_i = \begin{cases} 1, & \text{if } i \notin \mathcal{M} \text{ and } w_i > \varepsilon, \\ 0, & \text{otherwise}. \end{cases} \tag{150}$$

Here:

- **True Positives (TP):** Counts the number of *malicious clients* that are correctly identified as malicious by the method.
- **False Positives (FP):** Counts the number of *benign clients* that are incorrectly flagged as malicious.
- **False Negatives (FN):** Counts the number of *malicious clients* that are mistakenly identified as benign.
- **True Negatives (TN):** Counts the number of *benign clients* that are correctly identified as benign.

The total counts are computed as:

$$\text{TP} = \sum_{i=1}^{n} \text{TP}_i, \quad \text{FP} = \sum_{i=1}^{n} \text{FP}_i, \quad \text{FN} = \sum_{i=1}^{n} \text{FN}_i, \quad \text{TN} = \sum_{i=1}^{n} \text{TN}_i.$$

Using these counts, we compute:

- **Precision:**
$$\text{Precision} = \frac{\text{TP}}{\text{TP} + \text{FP}},$$
which measures the proportion of clients flagged as malicious that are actually malicious.
- **Recall:**
$$\text{Recall} = \frac{\text{TP}}{\text{TP} + \text{FN}},$$
which measures the proportion of actual malicious clients that are successfully detected.
- **F1 Score:**
$$\text{F1 Score} = 2 \times \frac{\text{Precision} \times \text{Recall}}{\text{Precision} + \text{Recall}},$$
providing a balance between precision and recall.

- **Accuracy:**

$$\text{Accuracy} = \frac{\text{TP} + \text{TN}}{n},$$

which measures the overall fraction of correctly classified clients (both malicious and benign), where $n$ is the total number of clients.

We set $\varepsilon = 10^{-4}$ to ensure that clients with very small aggregation weights are identified as malicious for evaluation. This threshold is appropriate, as fewer than 200 clients typically remain after excluding detected malicious ones. In the equal-weight case, benign clients receive weights above $5 \times 10^{-3}$, well above $\varepsilon$, allowing a clear distinction between benign and malicious clients.

Table 7 summarizes FedLAW's performance in detecting malicious clients on MNIST and CIFAR-10 under four attack types: Flipping Label, Inverse Gradient, Backdoor Attack, and Double Attack. We evaluate across different data heterogeneity levels ($q = 0.6$ and $q = 0.9$) and malicious client fractions (0.1 to 0.4), using Precision, Recall, F1 Score, and Accuracy. Results are reported as the mean ± standard deviation across five independent runs.

Recall and Accuracy are especially important: high Recall ensures most malicious clients are caught, which boosts Accuracy. This is critical when the malicious fraction is high. Precision helps avoid false positives, but its impact is less critical in practice when benign clients are the majority.

FedLAW consistently shows strong detection capabilities. For Flipping Label, Inverse Gradient, and Backdoor Attacks, often it achieves Recall values close to or equal to 1.0, often resulting in high Accuracy ($\geq 0.92$). CIFAR-10 generally yields better results than MNIST, likely due to its more complex features, which may help us in the detection procedure. Precision and F1 scores are also high (typically $\geq 0.9$), confirming robust overall performance.

The Double Attack poses a greater challenge due to its hybrid and dynamic nature. On CIFAR-10 with $q = 0.9$ and 40% malicious clients, FedLAW still secures solid Recall (0.899), but lower Precision (0.848) and F1 (0.872) pull Accuracy down to 0.897. On MNIST with $q = 0.6$ and 10% malicious clients, Recall falls to 0.7 and Accuracy to 0.940. At low malicious fractions, even minor misclassifications can significantly affect metrics, though their practical impact remains limited due to the low number of malicious. It is important to note that FedLAW's performance under the double attack is slightly lower compared to other attack types, reflecting the increased complexity of this scenario. The double attack is used specifically to assess the method's robustness in the most demanding settings.

In summary, FedLAW provides reliable and accurate detection of malicious clients across all four attack types and under highly heterogeneous data distributions, conditions that closely resemble real-world federated learning settings. Its consistently high Recall and Accuracy make it an effective and practical defense mechanism for secure federated learning systems.

Table 3: Test accuracy (%) on MNIST under five attack types, two non-IID levels ($q$) and varying fractions of malicious clients. Reported as mean ± std over five runs and are also depicted in Figure 2.a.

| Algorithm | Fraction of Malicious Clients | | | | | | | |
|---|---|---|---|---|---|---|---|---|
| | $q = 0.6$ | | | | $q = 0.9$ | | | |
| | 0.1 | 0.2 | 0.3 | 0.4 | 0.1 | 0.2 | 0.3 | 0.4 |
| **Attack: Flipping Label** | | | | | | | | |
| FedLAW | 92.63±0.07 | 92.71±0.05 | 92.39±0.07 | 92.22±0.20 | 89.60±0.24 | 89.55±0.57 | 88.30±0.28 | 87.45±0.26 |
| Bulyan | 92.06±0.04 | 91.86±0.32 | 91.68±0.20 | 91.14±0.45 | 89.01±0.06 | 87.59±0.95 | 87.03±0.66 | 87.53±0.35 |
| Bulyan-Bucketing | 92.18±0.24 | 91.89±0.35 | 91.69±0.36 | 91.22±0.28 | 88.61±0.37 | 87.41±1.09 | 85.99±0.38 | 85.36±1.37 |
| Krum | 86.08±0.21 | 86.29±0.80 | 86.37±0.56 | 86.43±0.24 | 76.46±0.84 | 76.72±1.26 | 76.16±0.96 | 75.55±0.60 |
| Trimmed Mean | 92.15±0.29 | 91.58±0.17 | 91.02±0.32 | 90.13±0.75 | 88.54±0.34 | 86.54±0.06 | 82.13±1.26 | 71.21±3.74 |
| CClip | 92.44±0.25 | 91.89±0.39 | 88.97±0.54 | 72.70±2.53 | 87.00±0.49 | 78.31±0.76 | 66.36±0.68 | 54.45±1.97 |
| CClip-Bucketing | 92.46±0.05 | 91.55±0.39 | 89.51±0.75 | 73.56±3.36 | 86.72±0.34 | 79.11±0.69 | 63.50±1.92 | 54.00±2.45 |
| RFA | 92.60±0.16 | 92.34±0.24 | 92.06±0.38 | 89.90±1.41 | 88.60±0.47 | 86.22±0.82 | 72.43±3.79 | 55.20±2.16 |
| RFA-Bucketing | 92.89±0.67 | 92.84±0.73 | 92.16±0.46 | 88.29±2.49 | 89.00±0.27 | 86.17±0.86 | 71.12±2.49 | 54.82±1.41 |
| CwMed | 92.09±0.28 | 91.64±0.20 | 91.03±0.14 | 90.25±0.23 | 88.12±0.45 | 86.09±0.32 | 82.68±0.54 | 72.82±3.87 |
| Huber Aggregator | 92.48±0.17 | 92.17±0.36 | 91.55±0.49 | 88.35±1.42 | 87.31±0.66 | 78.66±1.75 | 65.22±1.99 | 54.84±2.56 |
| FedAVG | 92.65±0.03 | 92.23±0.13 | 89.56±0.19 | 77.71±0.61 | 88.05±0.29 | 80.04±0.13 | 67.74±0.88 | 56.11±0.37 |
| **Attack: Inverse Gradient** | | | | | | | | |
| | | | | | | | | Continued on next page |

**Table 3 (continued) from previous page**

| Algorithm | Fraction of Malicious Clients | | | | | | | |
|---|---|---|---|---|---|---|---|---|
| | $q = 0.6$ | | | | $q = 0.9$ | | | |
| | 0.1 | 0.2 | 0.3 | 0.4 | 0.1 | 0.2 | 0.3 | 0.4 |
| FedLAW | 92.40±0.19 | 91.88±0.46 | 91.83±0.33 | 91.62±0.33 | 88.71±0.55 | 88.45±0.18 | 86.86±0.15 | 87.41±1.39 |
| Bulyan | 91.59±0.38 | 91.26±0.16 | 90.29±0.27 | 89.11±0.33 | 87.09±0.06 | 85.88±0.23 | 82.54±0.96 | 83.80±1.83 |
| Bulyan-Bucketing | 91.31±0.27 | 91.18±0.14 | 90.39±0.59 | 88.44±1.41 | 85.89±0.97 | 85.97±1.13 | 82.06±4.05 | 81.44±2.81 |
| Krum | 81.99±3.34 | 80.43±4.33 | 75.72±1.59 | 76.88±1.10 | 68.89±1.25 | 66.25±5.60 | 64.39±3.56 | 61.93±8.41 |
| Trimmed Mean | 91.38±0.32 | 90.63±0.08 | 88.49±1.00 | 84.35±1.52 | 84.49±0.75 | 77.76±2.30 | 66.53±3.91 | 52.38±1.55 |
| CClip | 84.98±3.19 | 73.78±1.31 | 10.00±0.00 | 10.00±0.00 | 10.00±0.00 | 10.00±0.00 | 10.00±0.00 | 10.00±0.00 |
| CClip-Bucketing | 83.02±1.36 | 73.73±0.18 | 10.00±0.00 | 10.00±0.00 | 10.00±0.00 | 10.00±0.00 | 10.00±0.00 | 10.00±0.00 |
| RFA | 91.82±0.29 | 90.70±0.15 | 87.85±4.97 | 60.88±3.61 | 84.92±1.08 | 76.15±1.00 | 65.85±2.18 | 10.00±0.00 |
| RFA-Bucketing | 92.18±0.04 | 90.32±0.13 | 89.14±0.88 | 59.05±0.26 | 84.54±0.87 | 76.69±0.18 | 67.39±0.94 | 10.00±0.00 |
| CwMed | 91.41±0.29 | 90.66±0.44 | 88.90±0.61 | 83.89±1.93 | 85.33±0.41 | 80.72±2.19 | 68.27±0.93 | 52.11±2.89 |
| Huber Aggregator | 91.71±0.32 | 84.81±3.37 | 65.26±1.16 | 57.09±1.55 | 82.08±1.34 | 73.28±1.32 | 64.00±0.94 | 55.40±1.30 |
| FedAVG | 84.16±1.05 | 74.06±0.44 | 10.00±0.00 | 10.00±0.00 | 10.00±0.00 | 10.00±0.00 | 10.00±0.00 | 10.00±0.00 |
| **Attack: Backdoor** | | | | | | | | |
| FedLAW | 92.32±0.14 | 92.64±0.30 | 92.19±0.13 | 92.46±0.27 | 89.69±0.42 | 89.17±1.77 | 88.81±0.71 | 87.88±1.16 |
| Bulyan | 92.15±0.07 | 91.73±0.22 | 69.23±1.95 | 59.66±3.32 | 88.79±0.04 | 83.28±2.40 | 51.79±8.79 | 33.89±1.80 |
| Bulyan-Bucketing | 92.11±0.16 | 91.62±0.27 | 90.17±0.09 | 88.54±0.44 | 88.27±0.19 | 85.22±1.69 | 79.39±4.89 | 72.17±3.84 |
| Krum | 27.60±3.99 | 28.16±0.97 | 15.83±4.34 | 22.64±6.54 | 15.21±0.35 | 16.36±2.40 | 18.68±2.33 | 16.90±2.69 |
| Trimmed Mean | 91.74±0.07 | 91.24±0.21 | 90.40±0.16 | 88.77±0.60 | 88.51±0.58 | 86.84±0.99 | 81.08±1.04 | 74.45±2.48 |
| CClip | 92.21±0.22 | 91.66±0.17 | 90.89±0.46 | 90.36±0.19 | 88.35±0.22 | 87.87±1.03 | 84.94±0.83 | 81.45±1.11 |
| CClip-Bucketing | 92.15±0.15 | 91.31±0.12 | 91.15±0.18 | 90.38±0.31 | 88.57±0.84 | 87.17±1.19 | 84.57±0.88 | 80.52±0.73 |
| RFA | 93.72±0.27 | 92.54±0.12 | 91.22±0.42 | 90.51±0.33 | 89.27±0.53 | 87.42±0.43 | 86.53±1.22 | 79.33±2.19 |
| RFA-Bucketing | 93.48±0.37 | 92.30±0.18 | 90.99±0.22 | 90.51±0.11 | 89.55±0.41 | 86.97±1.35 | 84.31±0.60 | 80.17±3.38 |
| CwMed | 91.61±0.26 | 91.02±0.30 | 90.40±0.33 | 88.44±0.69 | 87.78±0.48 | 86.23±0.41 | 81.17±2.18 | 67.89±4.39 |
| Huber Aggregator | 92.40±0.08 | 91.48±0.28 | 90.73±0.39 | 89.93±0.35 | 88.79±0.43 | 87.42±0.29 | 84.94±1.22 | 80.80±2.65 |
| FedAVG | 91.99±0.06 | 91.34±0.19 | 91.03±0.16 | 89.93±0.24 | 88.43±0.63 | 87.13±1.17 | 85.00±0.53 | 79.74±2.65 |
| **Attack: Double Attack** | | | | | | | | |
| FedLAW | 92.57±0.05 | 92.39±0.02 | 92.34±0.46 | 92.31±0.23 | 89.93±0.21 | 89.76±0.11 | 89.35±0.24 | 87.47±0.95 |
| Bulyan | 91.98±0.11 | 91.69±0.08 | 91.22±0.13 | 90.79±0.22 | 88.80±0.22 | 87.50±0.23 | 87.01±1.13 | 85.47±0.31 |
| Bulyan-Bucketing | 92.06±0.33 | 91.59±0.23 | 91.49±0.03 | 90.57±0.68 | 88.84±0.68 | 87.83±0.68 | 84.91±0.69 | 83.50±3.44 |
| Krum | 82.24±3.66 | 80.82±2.03 | 79.28±1.60 | 80.77±2.15 | 71.20±0.52 | 71.83±2.64 | 71.25±1.57 | 62.43±5.90 |
| Trimmed Mean | 92.00±0.04 | 91.61±0.40 | 90.68±0.67 | 89.74±0.26 | 88.32±0.45 | 85.02±1.05 | 82.01±2.83 | 76.10±1.22 |
| CClip | 92.08±0.12 | 91.67±0.08 | 90.78±0.62 | 82.99±4.99 | 89.12±0.29 | 80.84±1.60 | 68.12±3.87 | 10.00±0.00 |
| CClip-Bucketing | 92.27±0.18 | 91.72±0.21 | 90.47±0.38 | 86.45±4.14 | 87.97±1.15 | 82.27±2.20 | 74.48±2.68 | 10.00±0.00 |
| RFA | 94.37±1.39 | 94.29±0.21 | 94.13±0.17 | 93.58±0.30 | 91.59±0.50 | 83.74±4.85 | 85.23±2.65 | 57.15±1.60 |
| RFA-Bucketing | 94.94±0.23 | 94.78±0.25 | 94.07±1.53 | 91.50±5.75 | 91.13±1.46 | 89.97±0.14 | 76.23±5.72 | 60.54±4.33 |
| CwMed | 91.92±0.22 | 91.03±0.10 | 90.77±0.58 | 89.63±0.56 | 87.58±0.54 | 86.18±0.86 | 82.78±2.90 | 76.71±2.63 |
| Huber Aggregator | 91.97±0.31 | 91.28±0.51 | 90.08±1.04 | 85.00±2.90 | 86.06±1.28 | 78.47±2.42 | 66.62±0.95 | 57.96±0.35 |
| FedAVG | 92.29±0.21 | 91.62±0.11 | 90.83±0.40 | 85.68±6.05 | 89.42±0.46 | 86.79±1.64 | 71.49±6.16 | 68.28±3.67 |
| **Attack: Lie Attack** | | | | | | | | |
| FedLAW | 92.72±0.04 | 92.24±0.02 | 90.70±0.08 | 86.88±4.02 | 90.15±0.05 | 89.51±0.11 | 86.30±0.14 | 70.10±2.17 |
| Bulyan | 92.16±0.02 | 91.44±0.56 | 89.41±1.36 | 50.15±33.52 | 88.80±0.37 | 86.16±1.92 | 58.92±22.98 | 21.23±13.09 |
| Bulyan-Bucketing | 92.21±0.16 | 91.88±0.03 | 90.40±0.05 | 86.94±0.05 | 89.14±0.16 | 87.53±0.68 | 81.73±2.54 | 29.98±2.07 |
| Krum | 92.32±0.17 | 90.92±0.40 | 88.06±0.53 | 39.53±11.36 | 89.49±0.33 | 83.84±1.01 | 26.56±1.91 | 11.62±0.23 |
| Trimmed Mean | 92.21±0.25 | 92.07±0.20 | 90.66±0.01 | 87.58±0.23 | 88.94±0.11 | 88.42±0.15 | 81.78±0.49 | 56.70±4.70 |
| CClip | 92.57±0.09 | 92.56±0.16 | 92.09±0.33 | 90.60±0.12 | 89.90±0.42 | 88.85±0.18 | 86.47±1.21 | 81.63±2.68 |
| CClip-Bucketing | 92.76±0.14 | 92.76±0.07 | 92.11±0.14 | 90.64±0.09 | 89.91±0.27 | 89.13±0.26 | 87.96±1.15 | 82.69±1.09 |
| RFA | 92.25±0.10 | 91.70±0.18 | 90.94±0.12 | 89.15±0.09 | 89.69±0.44 | 86.76±0.28 | 82.25±2.92 | 55.50±3.14 |
| RFA-Bucketing | 92.52±0.19 | 91.74±0.12 | 91.03±0.21 | 89.29±0.59 | 89.13±0.47 | 87.35±0.95 | 82.16±1.63 | 57.39±7.07 |
| CwMed | 92.31±0.14 | 91.56±0.15 | 90.12±0.15 | 87.06±0.33 | 87.96±0.41 | 86.36±1.03 | 79.07±2.32 | 56.03±2.68 |
| Huber Aggregator | 92.45±0.13 | 92.04±0.02 | 92.09±0.30 | 90.69±0.12 | 90.08±0.44 | 89.34±0.48 | 86.19±1.03 | 68.83±14.50 |
| FedAVG | 92.67±0.11 | 92.61±0.21 | 91.91±0.03 | 90.16±0.01 | 90.06±0.69 | 88.54±1.10 | 86.92±0.88 | 84.22±2.34 |

Table 4: Test accuracy (%) on CIFAR-10 under five attack types, two non-IID levels ($q$) and varying fractions of malicious clients. Reported as mean $\pm$ std over five runs and are also depicted in Figure 2.b.

| Algorithm | Fraction of Malicious Clients | | | | | | | |
|---|---|---|---|---|---|---|---|---|
| | $q = 0.6$ | | | | $q = 0.9$ | | | |
| | 0.1 | 0.2 | 0.3 | 0.4 | 0.1 | 0.2 | 0.3 | 0.4 |
| **Attack: Flipping Label** | | | | | | | | |
| FedLAW | 74.59±0.54 | 73.21±0.09 | 72.43±0.23 | 70.52±0.16 | 66.86±0.50 | 64.85±0.24 | 62.39±0.51 | 58.86±2.22 |
| Bulyan | 73.53±0.71 | 70.55±0.75 | 68.60±0.53 | 62.20±1.97 | 64.93±0.57 | 61.84±0.92 | 60.00±0.94 | 53.57±10.37 |
| Bulyan-Bucketing | 70.52±3.66 | 71.56±0.68 | 67.17±3.08 | 45.86±12.96 | 59.48±6.86 | 58.33±6.53 | 58.56±6.54 | 47.82±5.42 |
| Krum | 27.66±4.51 | 24.77±2.19 | 27.19±0.45 | 24.99±0.14 | 21.23±0.40 | 17.64±2.69 | 11.94±0.00 | 15.97±0.18 |
| Trimmed Mean | 73.12±0.03 | 71.72±0.03 | 67.39±0.30 | 51.18±0.74 | 62.24±0.33 | 59.27±0.29 | 53.02±1.97 | 43.45±2.09 |
| CClip | 72.42±0.44 | 66.19±1.13 | 60.12±5.46 | 39.98±8.15 | 61.93±0.91 | 55.58±1.91 | 47.93±2.25 | 42.22±1.57 |

Table 4 (continued) from previous page

| Algorithm | Fraction of Malicious Clients | | | | | | | |
|---|---|---|---|---|---|---|---|---|
| | $q = 0.6$ | | | | $q = 0.9$ | | | |
| | 0.1 | 0.2 | 0.3 | 0.4 | 0.1 | 0.2 | 0.3 | 0.4 |
| CClip-Bucketing | 72.31±0.37 | 66.89±0.87 | 55.85±6.59 | 41.97±9.59 | 61.65±0.81 | 54.95±1.17 | 49.94±2.21 | 41.34±2.04 |
| RFA | 72.61±0.75 | 69.70±2.29 | 64.16±3.89 | 50.96±4.01 | 62.66±1.07 | 57.76±0.66 | 51.42±2.16 | 42.11±0.80 |
| RFA-Bucketing | 73.81±0.63 | 71.45±1.69 | 67.51±4.59 | 53.38±1.80 | 63.10±0.78 | 57.10±1.05 | 49.06±2.86 | 40.19±0.62 |
| CwMed | 73.50±0.42 | 71.03±0.31 | 68.66±1.03 | 52.44±2.28 | 63.91±0.41 | 59.80±1.26 | 53.40±1.68 | 44.92±3.49 |
| Huber Aggregator | 73.50±0.54 | 70.80±1.04 | 66.18±0.25 | 55.77±1.04 | 62.66±0.86 | 57.13±0.24 | 51.06±5.03 | 38.95±1.61 |
| FedAVG | 72.19±0.64 | 67.63±0.61 | 52.23±7.76 | 51.21±9.39 | 61.70±0.73 | 54.91±0.78 | 48.59±1.52 | 41.24±1.78 |
| **Attack: Inverse Gradient** | | | | | | | | |
| FedLAW | 74.45±0.58 | 73.16±0.63 | 72.05±0.55 | 70.32±0.24 | 66.26±0.67 | 64.41±0.37 | 62.95±1.13 | 59.38±1.14 |
| Bulyan | 72.82±0.66 | 70.56±0.30 | 67.62±0.73 | 65.98±0.11 | 64.55±1.31 | 60.46±1.38 | 57.53±1.79 | 56.24±1.19 |
| Bulyan-Bucketing | 72.42±0.35 | 71.50±0.26 | 68.01±0.78 | 52.84±3.32 | 64.13±0.34 | 61.57±1.11 | 59.58±1.28 | 47.20±0.63 |
| Krum | 29.68±0.71 | 30.11±0.98 | 29.15±2.62 | 24.48±1.44 | 19.23±3.51 | 17.59±2.46 | 21.32±3.50 | 17.03±1.57 |
| Trimmed Mean | 71.05±0.25 | 67.18±0.39 | 59.22±0.86 | 42.44±3.84 | 61.56±1.73 | 55.27±1.55 | 48.16±1.34 | 38.17±1.37 |
| CClip | 71.40±0.29 | 67.61±0.47 | 67.09±1.48 | 10.00±0.00 | 61.17±0.73 | 55.01±1.27 | 53.53±5.57 | 10.00±0.00 |
| CClip-Bucketing | 71.44±0.61 | 67.39±1.09 | 67.20±0.63 | 10.00±0.00 | 62.89±1.58 | 56.77±0.31 | 52.09±3.76 | 10.00±0.00 |
| RFA | 70.81±0.61 | 65.37±1.74 | 10.00±0.00 | 10.00±0.00 | 60.98±2.00 | 10.00±0.00 | 10.00±0.00 | 10.00±0.00 |
| RFA-Bucketing | 71.37±1.07 | 66.71±1.17 | 10.00±0.00 | 10.00±0.00 | 61.26±1.18 | 10.00±0.00 | 10.00±0.00 | 10.00±0.00 |
| CwMed | 71.93±0.78 | 68.08±0.40 | 60.48±1.74 | 42.89±1.86 | 62.08±0.36 | 56.74±1.67 | 49.29±2.31 | 34.10±4.47 |
| Huber Aggregator | 72.00±0.78 | 67.56±0.16 | 65.55±1.39 | 10.00±0.00 | 61.96±1.68 | 56.37±0.76 | 47.93±2.93 | 10.00±0.00 |
| FedAVG | 67.83±1.58 | 10.00±0.00 | 10.00±0.00 | 10.00±0.00 | 10.00±0.00 | 10.00±0.00 | 10.00±0.00 | 10.00±0.00 |
| **Attack: Backdoor** | | | | | | | | |
| FedLAW | 74.21±0.05 | 72.89±0.14 | 71.77±0.11 | 70.10±0.38 | 66.53±0.22 | 64.73±0.90 | 63.07±0.43 | 59.90±0.17 |
| Bulyan | 73.05±0.27 | 66.66±1.16 | 47.46±0.07 | 30.93±0.32 | 62.85±0.21 | 54.52±0.84 | 35.38±2.69 | 18.62±2.53 |
| Bulyan-Bucketing | 71.61±0.34 | 69.73±0.51 | 67.06±0.72 | 60.39±0.48 | 63.53±0.33 | 58.58±1.21 | 53.70±0.91 | 48.04±0.84 |
| Krum | 25.36±1.17 | 16.47±0.35 | 15.18±1.60 | 12.12±3.10 | 13.20±3.32 | 13.76±5.08 | 9.79±0.29 | 10.06±0.08 |
| Trimmed Mean | 74.69±0.39 | 73.11±0.29 | 70.72±1.24 | 66.07±0.76 | 65.66±0.20 | 64.05±0.18 | 59.45±1.63 | 52.92±2.76 |
| CClip | 73.61±0.48 | 72.94±0.47 | 71.32±0.24 | 68.59±0.50 | 65.91±0.29 | 64.13±0.94 | 60.60±1.18 | 56.54±0.99 |
| CClip-Bucketing | 74.16±0.50 | 73.09±0.31 | 71.24±0.29 | 69.33±0.74 | 65.94±0.65 | 63.45±0.21 | 60.30±0.80 | 56.31±0.42 |
| RFA | 74.15±0.51 | 73.00±0.15 | 71.90±0.47 | 69.87±0.66 | 66.22±0.41 | 64.00±0.65 | 62.05±0.49 | 59.04±0.63 |
| RFA-Bucketing | 73.80±0.63 | 72.56±0.49 | 71.67±0.46 | 69.69±0.45 | 66.10±0.73 | 63.70±1.25 | 62.39±0.16 | 59.69±0.50 |
| CwMed | 73.83±0.44 | 72.07±0.33 | 69.79±0.64 | 65.50±2.24 | 65.44±0.64 | 63.66±1.28 | 57.91±1.28 | 52.14±0.59 |
| Huber Aggregator | 74.07±0.14 | 71.73±0.25 | 69.15±0.65 | 65.30±1.46 | 65.12±0.70 | 62.13±0.89 | 60.26±0.30 | 55.30±0.30 |
| FedAVG | 74.36±0.18 | 72.88±0.37 | 71.24±0.38 | 69.71±0.28 | 66.35±0.23 | 63.21±0.68 | 60.62±1.92 | 55.94±0.86 |
| **Attack: Double Attack** | | | | | | | | |
| FedLAW | 74.09±0.31 | 72.89±0.28 | 71.63±0.29 | 70.54±0.70 | 66.45±0.08 | 64.95±0.14 | 62.89±0.57 | 59.98±1.25 |
| Bulyan | 73.56±0.53 | 70.60±0.18 | 69.44±0.34 | 69.01±0.36 | 64.86±0.36 | 62.14±1.28 | 60.15±1.44 | 59.02±1.02 |
| Bulyan-Bucketing | 72.85±0.11 | 72.09±0.31 | 71.20±0.41 | 64.51±0.23 | 64.47±0.19 | 64.10±0.51 | 61.77±0.55 | 53.64±1.19 |
| Krum | 25.41±1.33 | 29.35±2.74 | 23.43±2.81 | 26.55±1.57 | 18.94±1.51 | 18.02±3.86 | 17.60±3.54 | 17.95±2.79 |
| Trimmed Mean | 72.72±0.67 | 69.81±1.93 | 67.26±0.72 | 62.76±0.06 | 63.30±0.13 | 60.16±0.11 | 56.54±1.20 | 50.95±1.11 |
| CClip | 72.49±0.29 | 70.71±0.35 | 67.77±0.96 | 63.34±0.78 | 63.27±0.86 | 59.31±1.02 | 52.72±0.48 | 46.01±2.03 |
| CClip-Bucketing | 72.81±0.82 | 69.91±0.32 | 67.43±1.01 | 63.02±1.23 | 63.27±1.46 | 59.48±0.88 | 53.14±1.73 | 45.99±1.49 |
| RFA | 58.99±2.02 | 64.97±2.37 | 55.80±3.36 | 10.00±0.00 | 66.19±2.71 | 59.29±1.40 | 46.31±5.51 | 10.00±0.00 |
| RFA-Bucketing | 64.33±3.61 | 70.75±3.30 | 57.14±3.17 | 10.00±0.00 | 66.90±1.67 | 60.26±2.12 | 10.00±0.00 | 10.00±0.00 |
| CwMed | 73.22±0.61 | 70.79±0.47 | 66.95±0.67 | 62.99±0.77 | 64.65±0.89 | 61.84±0.79 | 57.28±0.39 | 52.12±1.44 |
| Huber Aggregator | 73.11±0.05 | 70.17±0.06 | 67.61±0.47 | 63.06±0.86 | 64.06±0.51 | 60.20±2.26 | 53.12±1.82 | 46.73±1.73 |
| FedAVG | 71.98±0.53 | 68.73±0.15 | 65.74±0.82 | 58.39±1.36 | 62.11±1.10 | 56.41±1.07 | 46.83±1.30 | 39.48±1.05 |
| **Attack: Lie Attack** | | | | | | | | |
| FedLAW | 74.20±0.30 | 71.96±0.26 | 60.91±0.37 | 41.56±3.28 | 66.49±0.19 | 61.79±0.50 | 46.22±1.76 | 28.34±0.44 |
| Bulyan | 73.52±0.20 | 50.18±0.61 | 31.19±2.30 | 10.00±0.01 | 65.57±0.63 | 40.99±1.13 | 13.82±3.44 | 10.00±0.00 |
| Bulyan-Bucketing | 73.19±0.27 | 64.14±0.51 | 42.53±2.28 | 27.45±1.14 | 64.99±0.21 | 51.49±0.31 | 32.16±1.41 | 11.69±2.99 |
| Krum | 65.43±0.76 | 45.04±1.09 | 28.70±1.52 | 10.00±0.00 | 52.68±0.60 | 35.39±1.33 | 13.32±3.13 | 10.01±0.02 |
| Trimmed Mean | 74.66±0.25 | 69.01±0.48 | 45.98±1.21 | 27.65±0.40 | 67.00±0.68 | 57.22±0.41 | 35.47±1.48 | 10.00±0.00 |
| CClip | 74.07±0.17 | 72.78±0.17 | 63.91±0.02 | 42.54±0.27 | 66.35±0.14 | 64.08±0.59 | 49.48±0.25 | 34.11±1.04 |
| CClip-Bucketing | 74.10±0.64 | 72.80±0.32 | 64.31±0.18 | 43.05±0.74 | 66.75±0.67 | 64.39±0.01 | 52.47±0.18 | 31.59±1.50 |
| RFA | 74.27±0.12 | 65.25±0.46 | 50.53±0.62 | 33.52±2.06 | 66.18±0.06 | 52.98±1.13 | 39.49±1.08 | 22.14±1.86 |
| RFA-Bucketing | 74.47±0.04 | 66.18±0.48 | 50.92±0.15 | 35.05±1.12 | 66.18±0.54 | 54.56±1.37 | 40.71±0.27 | 25.56±0.28 |
| CwMed | 72.04±0.08 | 58.56±0.77 | 43.48±0.81 | 21.37±2.06 | 60.80±0.19 | 47.00±2.13 | 31.23±3.98 | 10.00±0.01 |
| Huber Aggregator | 74.33±0.10 | 68.08±0.18 | 51.65±0.29 | 36.16±1.98 | 67.08±0.91 | 61.23±0.14 | 45.58±1.24 | 26.49±2.58 |
| FedAVG | 74.30±0.06 | 73.51±0.20 | 64.04±0.88 | 44.70±2.87 | 66.54±0.99 | 64.30±0.41 | 51.71±0.80 | 33.04±2.57 |

## I.5 COMPUTATIONAL COMPLEXITY

**Communication Overhead between Server and Clients:** As demonstrated in Table 5, state-of-the-art Byzantine-robust federated learning methods typically do not update aggregation weights, incurring zero communication rounds for this step. In contrast, our proposed method, FedLAW, optimizes the aggregation weights $\mathbf{w}$ alongside the global model parameters $\boldsymbol{\theta}$, requiring an additional 20 communication rounds for $\mathbf{w}$ updates. Since Byzantine attacks are assumed to occur early in

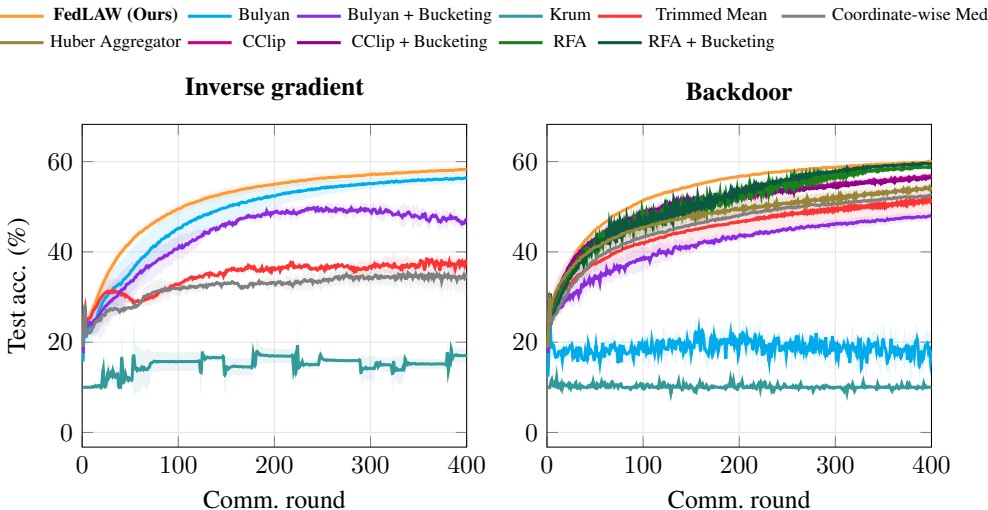

Figure 8: Convergence on CIFAR-10 under two adversarial settings ($q = 0.9$, 40% malicious clients; see Section 5 for details). Each panel shows the average test accuracy over 5 independent runs; shaded regions denote $\pm 1$std. FedLAW uses two communication rounds per model update, so 400 rounds correspond to 200 global epochs. All other methods run for 400 global epochs. We report results for one data poisoning attack (backdoor) and one model attack (inverse gradient), which are representative of the broader attack space. Methods excluded from the inverse gradient plot did not converge.

training and $\mathbf{w}$ converges quickly, we restrict its updates to the first 20 rounds, ensuring a fixed and limited communication overhead.

To ensure a fair comparison with existing methods and maintain a standardized benchmark, we align the number of communication rounds for updating $\boldsymbol{\theta}$ with those used by state-of-the-art approaches in our numerical study. This setup allows for a direct evaluation of FedLAW's performance under equivalent conditions for $\boldsymbol{\theta}$ updates. However, as highlighted in Remark 1 and illustrated in Figure 8, FedLAW achieves the target accuracy with fewer $\boldsymbol{\theta}$-update rounds compared to other methods, owing to the joint optimization of $\mathbf{w}$ and $\boldsymbol{\theta}$. This flexibility enhances convergence speed, effectively offsetting the additional 20 rounds for $\mathbf{w}$ updates and potentially reducing the total communication cost in practical settings.

Empirically, we observe that the learned weights change most during the early phase (roughly the first 10–20% of communication rounds); after this, they remain essentially stable, so fixing $\mathbf{w}$ has negligible impact on robustness or accuracy while keeping FedLAW's long-term communication cost close to that of FedAvg. When continued adaptivity is desired (e.g., under non-stationary data or late-onset attacks), $\mathbf{w}$ can be updated beyond this warm-up phase or periodically refreshed, at the price of a modest additional communication cost.

Finally, we note that the CIFAR-10 dataset requires more $\boldsymbol{\theta}$-update rounds than MNIST due to its higher complexity, which necessitates additional training effort to achieve robust convergence across all methods.

**Memory Complexity of Server-Side Weight Update**: In our method, the server updates both the global model parameters and the aggregation weights. The weight update involves computing the vector

$$\mathbf{h}_k = \mathbf{w}_k + \beta\alpha\mathbf{G}_k^T\tilde{\mathbf{G}}_{k+1}\mathbf{w}_k - \beta\mathbf{f}(\tilde{\boldsymbol{\theta}}_{k+1}) \tag{151}$$

where $\mathbf{w}_k \in \mathbb{R}^n$ are the aggregation weights, $\mathbf{G}_k = [\nabla_\theta f_1(\boldsymbol{\theta}_k), \cdots, \nabla_\theta f_n(\boldsymbol{\theta}_k)] \in \mathbb{R}^{d\times n}$, $\tilde{\mathbf{G}}_{k+1} = [\nabla_\theta f_1(\tilde{\boldsymbol{\theta}}_{k+1}), \cdots, \nabla_\theta f_n(\tilde{\boldsymbol{\theta}}_{k+1})] \in \mathbb{R}^{d\times n}$ are gradient matrices, and $\mathbf{f}(\tilde{\boldsymbol{\theta}}_{k+1}) \in \mathbb{R}^n$ denotes the losses of all clients. The vector $\mathbf{h}_k$ is then projected onto the sparse unit-capped simplex.

To analyze the memory requirements for computing $\mathbf{h}_k$, we summarize the necessary data stored on the server in Table 6.

Table 5: Number of communication rounds for updating $\theta$ and $\mathbf{w}$.

| Algorithm | Dataset | Rounds for $\theta$ | Rounds for w |
|---|---|---|---|
| FedLAW (ours) | MNIST | 200 | 20 |
| | CIFAR10 | 400 | 20 |
| Bulyan | MNIST | 200 | 0 |
| | CIFAR10 | 400 | 0 |
| Krum | MNIST | 200 | 0 |
| | CIFAR10 | 400 | 0 |
| Trimmed Mean | MNIST | 200 | 0 |
| | CIFAR10 | 400 | 0 |
| FedAVG | MNIST | 200 | 0 |
| | CIFAR10 | 400 | 0 |

| Symbol | Shape | Purpose | Memory Cost |
|---|---|---|---|
| $\mathbf{w}_k$ | $n$ | Aggregation weights | $\mathcal{O}(n)$ |
| $\mathbf{G}_k$ | $d \times n$ | Previous round gradients | $\mathcal{O}(dn)$ |
| $\tilde{\mathbf{G}}_{k+1}$ | $d \times n$ | Current round gradients | $\mathcal{O}(dn)$ |
| $\mathbf{f}(\tilde{\theta}_{k+1})$ | $n$ | client losses | $\mathcal{O}(n)$ |
| Intermediate $\mathbf{z} = \tilde{\mathbf{G}}_{k+1}\mathbf{w}_k$ | $d$ | Matrix-vector product scratch | $\mathcal{O}(d)$ |
| Projection buffer | $n$ | Sparse simplex projection | $\mathcal{O}(n)$ |

Table 6: Memory components required for computing aggregation weights.

The dominant memory cost arises from storing the two gradient matrices $\mathbf{G}_k$ and $\tilde{\mathbf{G}}_{k+1}$, each of size $\mathcal{O}(dn)$. The other components contribute lower-order terms, resulting in an overall memory complexity of $\mathcal{O}(dn)$.

Note that forming the product $\mathbf{G}_k^T \tilde{\mathbf{G}}_{k+1}\mathbf{w}_k$ naively would require constructing an $\mathcal{O}(n^2)$ matrix. However, this can be avoided by computing it in two steps:

1. Compute $\mathbf{z} = \tilde{\mathbf{G}}_{k+1}\mathbf{w}_k$ with cost $\mathcal{O}(dn)$.

2. Then compute $\mathbf{G}_k^T \mathbf{z}$ with cost $\mathcal{O}(dn)$.

In summary, the memory required for server-side weight updates scales linearly with both the number of model parameters and the number of clients. Therefore, the total memory requirement per round is $\mathcal{O}(dn)$.

**Computational Complexity of Projection onto the sparse unit capped simplex**: According to (10), the projection onto the sparse unit-capped simplex involves the following steps:

- **Sparsity enforcement:** $\mathbf{h}_\lambda = P_{L_s}(\mathbf{h}_k)$, selecting the top-$s$ elements from $\mathbf{h}_k \in \mathbb{R}^n$. This step can be performed in $\mathcal{O}(n \min(s, \log n))$ (Kyrillidis et al., 2013).

- **Support selection:** $\mathcal{S}^* = \text{supp}(\mathbf{h}_\lambda)$, identifying the indices of the top-$s$ elements, which requires $\mathcal{O}(n)$ time.

- **Unit-capped simplex projection:** $\mathbf{w}_{k+1\mathcal{S}^*} = \mathcal{P}_{\Delta_t^+}(\mathbf{h}_{\lambda\mathcal{S}^*})$, $\quad \mathbf{w}_{k+1(\mathcal{S}^*)^\complement} = 0$. This projection onto the unit-capped simplex has a computational complexity of $\mathcal{O}(s^2)$ (Wang & Lu, 2015).

The total complexity of the projection step is $\mathcal{O}(n \min(s, \log n) + s^2)$.

### I.6 EMPIRICAL SIZE OF $\varepsilon_k$.

In this part, we explicitly measured $\varepsilon_k$ in Theorem 2 under a high malicious fraction and extreme heterogeneity (malicious client fraction $0.4$, non-IID parameter $q = 0.9$, backdoor attack). For each communication round $k$ over the course of training (up to $400$ epochs), we computed $\varepsilon_k$ and aggregated the results over five random seeds.

| Attack Type | q | Frac. Mal. | MNIST | | | | CIFAR-10 | | | |
|---|---|---|---|---|---|---|---|---|---|---|
| | | | Prec. | Rec. | F1 | Acc. | Prec. | Rec. | F1 | Acc. |
| Flipping Label | 0.6 | 0.4 | $0.990 \pm 0.010$ | $0.998 \pm 0.006$ | $0.994 \pm 0.006$ | $0.995 \pm 0.005$ | $0.976 \pm 0.055$ | $1.000$ | $0.987 \pm 0.029$ | $0.989 \pm 0.025$ |
| | | 0.3 | $0.990 \pm 0.022$ | $1.000$ | $0.995 \pm 0.011$ | $0.997 \pm 0.007$ | $0.971 \pm 0.065$ | $1.000$ | $0.984 \pm 0.035$ | $0.990 \pm 0.023$ |
| | | 0.2 | $0.963 \pm 0.051$ | $1.000$ | $0.980 \pm 0.027$ | $0.992 \pm 0.011$ | $0.959 \pm 0.091$ | $1.000$ | $0.977 \pm 0.051$ | $0.990 \pm 0.023$ |
| | | 0.1 | $0.982 \pm 0.041$ | $1.000$ | $0.990 \pm 0.021$ | $0.998 \pm 0.005$ | $0.964 \pm 0.101$ | $1.000$ | $0.979 \pm 0.059$ | $0.995 \pm 0.014$ |
| | 0.9 | 0.4 | $0.921 \pm 0.052$ | $0.948 \pm 0.072$ | $0.934 \pm 0.060$ | $0.946 \pm 0.049$ | $0.954 \pm 0.048$ | $0.950 \pm 0.054$ | $0.952 \pm 0.051$ | $0.961 \pm 0.041$ |
| | | 0.3 | $0.963 \pm 0.045$ | $0.966 \pm 0.047$ | $0.965 \pm 0.046$ | $0.979 \pm 0.028$ | $0.937 \pm 0.058$ | $0.987 \pm 0.014$ | $0.960 \pm 0.026$ | $0.975 \pm 0.017$ |
| | | 0.2 | $0.970 \pm 0.021$ | $0.975 \pm 0.025$ | $0.972 \pm 0.023$ | $0.989 \pm 0.009$ | $0.932 \pm 0.073$ | $0.965 \pm 0.029$ | $0.947 \pm 0.044$ | $0.978 \pm 0.019$ |
| | | 0.1 | $0.972 \pm 0.041$ | $1.000$ | $0.986 \pm 0.021$ | $0.997 \pm 0.005$ | $0.948 \pm 0.064$ | $0.939 \pm 0.082$ | $0.944 \pm 0.073$ | $0.989 \pm 0.014$ |
| Inverse Gradient | 0.6 | 0.4 | $0.880 \pm 0.045$ | $0.874 \pm 0.052$ | $0.877 \pm 0.049$ | $0.902 \pm 0.039$ | $0.956 \pm 0.063$ | $0.973 \pm 0.061$ | $0.964 \pm 0.060$ | $0.971 \pm 0.048$ |
| | | 0.3 | $0.824 \pm 0.166$ | $0.823 \pm 0.165$ | $0.824 \pm 0.165$ | $0.895 \pm 0.098$ | $0.975 \pm 0.037$ | $1.000$ | $0.987 \pm 0.019$ | $0.992 \pm 0.012$ |
| | | 0.2 | $0.913 \pm 0.140$ | $0.917 \pm 0.138$ | $0.915 \pm 0.139$ | $0.965 \pm 0.056$ | $0.963 \pm 0.082$ | $1.000$ | $0.980 \pm 0.045$ | $0.991 \pm 0.012$ |
| | | 0.1 | $0.913 \pm 0.194$ | $0.930 \pm 0.157$ | $0.921 \pm 0.177$ | $0.983 \pm 0.039$ | $0.968 \pm 0.092$ | $1.000$ | $0.981 \pm 0.053$ | $0.996 \pm 0.012$ |
| | 0.9 | 0.4 | $0.899 \pm 0.072$ | $0.904 \pm 0.064$ | $0.902 \pm 0.068$ | $0.922 \pm 0.054$ | $0.997 \pm 0.006$ | $0.997 \pm 0.006$ | $0.997 \pm 0.006$ | $0.998 \pm 0.005$ |
| | | 0.3 | $0.833 \pm 0.071$ | $0.833 \pm 0.071$ | $0.833 \pm 0.071$ | $0.899 \pm 0.043$ | $0.969 \pm 0.048$ | $0.997 \pm 0.008$ | $0.982 \pm 0.029$ | $0.989 \pm 0.018$ |
| | | 0.2 | $0.825 \pm 0.071$ | $0.825 \pm 0.071$ | $0.825 \pm 0.071$ | $0.929 \pm 0.029$ | $0.943 \pm 0.087$ | $0.985 \pm 0.034$ | $0.963 \pm 0.061$ | $0.984 \pm 0.026$ |
| | | 0.1 | $0.707 \pm 0.033$ | $0.857 \pm 0.131$ | $0.774 \pm 0.073$ | $0.953 \pm 0.008$ | $0.901 \pm 0.140$ | $0.990 \pm 0.022$ | $0.939 \pm 0.088$ | $0.986 \pm 0.021$ |
| Backdoor Attack | 0.6 | 0.4 | $0.979 \pm 0.018$ | $1.000$ | $0.989 \pm 0.009$ | $0.991 \pm 0.008$ | $0.968 \pm 0.068$ | $1.000$ | $0.983 \pm 0.037$ | $0.985 \pm 0.033$ |
| | | 0.3 | $0.980 \pm 0.030$ | $1.000$ | $0.990 \pm 0.015$ | $0.994 \pm 0.010$ | $1.000$ | $1.000$ | $1.000$ | $1.000$ |
| | | 0.2 | $0.970 \pm 0.023$ | $1.000$ | $0.985 \pm 0.012$ | $0.994 \pm 0.005$ | $0.963 \pm 0.090$ | $1.000$ | $0.979 \pm 0.050$ | $0.991 \pm 0.023$ |
| | | 0.1 | $0.900 \pm 0.039$ | $1.000$ | $0.947 \pm 0.022$ | $0.989 \pm 0.005$ | $1.000$ | $1.000$ | $1.000$ | $1.000$ |
| | 0.9 | 0.4 | $0.997 \pm 0.006$ | $1.000$ | $0.998 \pm 0.003$ | $0.999 \pm 0.003$ | $0.966 \pm 0.064$ | $1.000$ | $0.981 \pm 0.035$ | $0.984 \pm 0.031$ |
| | | 0.3 | $0.988 \pm 0.016$ | $1.000$ | $0.994 \pm 0.008$ | $0.996 \pm 0.005$ | $0.972 \pm 0.061$ | $1.000$ | $0.985 \pm 0.033$ | $0.990 \pm 0.022$ |
| | | 0.2 | $0.959 \pm 0.035$ | $1.000$ | $0.979 \pm 0.018$ | $0.991 \pm 0.008$ | $0.968 \pm 0.095$ | $1.000$ | $0.981 \pm 0.056$ | $0.991 \pm 0.027$ |
| | | 0.1 | $0.915 \pm 0.085$ | $1.000$ | $0.954 \pm 0.048$ | $0.990 \pm 0.011$ | $1.000$ | $1.000$ | $1.000$ | $1.000$ |
| Double Attack | 0.6 | 0.4 | $0.874 \pm 0.010$ | $0.874 \pm 0.010$ | $0.874 \pm 0.010$ | $0.899 \pm 0.008$ | $0.855 \pm 0.093$ | $0.897 \pm 0.130$ | $0.875 \pm 0.111$ | $0.899 \pm 0.088$ |
| | | 0.3 | $0.825 \pm 0.026$ | $0.825 \pm 0.026$ | $0.825 \pm 0.026$ | $0.896 \pm 0.015$ | $0.856 \pm 0.042$ | $0.909 \pm 0.070$ | $0.881 \pm 0.054$ | $0.927 \pm 0.032$ |
| | | 0.2 | $0.807 \pm 0.028$ | $0.800 \pm 0.025$ | $0.803 \pm 0.026$ | $0.921 \pm 0.011$ | $0.826 \pm 0.029$ | $0.960 \pm 0.065$ | $0.887 \pm 0.041$ | $0.952 \pm 0.017$ |
| | | 0.1 | $0.696 \pm 0.037$ | $0.707 \pm 0.027$ | $0.701 \pm 0.031$ | $0.940 \pm 0.008$ | $0.681 \pm 0.015$ | $0.990 \pm 0.022$ | $0.807 \pm 0.018$ | $0.952 \pm 0.005$ |
| | 0.9 | 0.4 | $0.804 \pm 0.133$ | $0.810 \pm 0.132$ | $0.807 \pm 0.132$ | $0.845 \pm 0.106$ | $0.848 \pm 0.100$ | $0.899 \pm 0.141$ | $0.872 \pm 0.119$ | $0.897 \pm 0.095$ |
| | | 0.3 | $0.804 \pm 0.039$ | $0.804 \pm 0.039$ | $0.804 \pm 0.039$ | $0.882 \pm 0.025$ | $0.848 \pm 0.129$ | $0.888 \pm 0.162$ | $0.867 \pm 0.145$ | $0.920 \pm 0.086$ |
| | | 0.2 | $0.756 \pm 0.039$ | $0.750 \pm 0.043$ | $0.753 \pm 0.041$ | $0.901 \pm 0.016$ | $0.776 \pm 0.121$ | $0.823 \pm 0.164$ | $0.799 \pm 0.141$ | $0.918 \pm 0.057$ |
| | | 0.1 | $0.697 \pm 0.133$ | $0.697 \pm 0.133$ | $0.697 \pm 0.133$ | $0.939 \pm 0.027$ | $0.764 \pm 0.082$ | $0.970 \pm 0.045$ | $0.852 \pm 0.055$ | $0.966 \pm 0.015$ |

Table 7: Precision, recall, F1 score, and accuracy of FEDLAW under different attack types, values of $q$, and fractions of malicious clients on MNIST and CIFAR-10. Results are reported as the mean $\pm$ standard deviation over 5 independent runs.

- **CIFAR-10** (400 global epochs), the average value of $\varepsilon_k$ over training is approximately $0.010$, with median $0.003$. The per-round mean $\varepsilon_k$ always lies in the interval $[0.001, 0.065]$; about $96.5\%$ of the rounds satisfy $\varepsilon_k < 0.05$ and all rounds satisfy $\varepsilon_k < 0.1$.

- **MNIST** (200 global epochs), the average $\varepsilon_k$ is approximately $0.040$, with median $0.034$. Here the per-round mean $\varepsilon_k$ lies in $[0.028, 0.155]$; around $85.5\%$ of the rounds satisfy $\varepsilon_k < 0.05$ and $99\%$ satisfy $\varepsilon_k < 0.1$. Individual runs can exhibit a short transient in the very first rounds where $\varepsilon_k$ reaches up to $\approx 0.33$, but this effect quickly decays and does not persist.

Overall, these measurements show that, even under a high malicious fraction and extreme data heterogeneity, $\varepsilon_k$ remains on the order of $10^{-2}$–$10^{-1}$ throughout training (e.g., means $\approx 0.01$ on CIFAR-10 and $\approx 0.04$ on MNIST), with almost all rounds below 0.05–0.1. This is one to two orders of magnitude smaller than the typical training loss values (which are $\mathcal{O}(1)$), and is thus consistent with treating $\varepsilon_k$ as a lower-order term in our analysis.

### I.7 ADDITIONAL EXPERIMENTS ON CIFAR-100

We further evaluate FedLAW on a more challenging image-classification benchmark with many classes. We consider CIFAR-100 (100 classes) with $n = 200$ clients, using the same model architecture and training protocol as in the CIFAR-10 experiments. The data are split in a strongly non-IID manner using two heterogeneity levels, $q = 0.6$ and $q = 0.9$, and we vary the fraction of malicious clients from $10\%$ to $40\%$. As representative attacks, we consider flipping label, inverse gradient, backdoor, double attack, and Lie attack, applied to all aggregators under comparison. The detailed results are summarized in Table 8.

Under the flipping label attack with $q = 0.6$, FedLAW remains stable as the malicious fraction increases, decreasing moderately from $42.27\% \pm 0.27$ (10%) to $37.50\% \pm 0.15$ (40%), while several robust baselines degrade substantially and some collapse close to random guessing ($\approx 1\%$ on 100

classes). For the more extreme non-IID setting $q = 0.9$, all methods degrade, and FedLAW drops from $32.12\% \pm 0.70$ (10%) to $21.49\% \pm 0.81$ (40%).

Under the inverse-gradient attack with $q = 0.6$, FedLAW consistently attains test accuracies around 38–43% across all malicious fractions, while several robust baselines degrade substantially and, in some cases, collapse close to random guessing ($\approx 1\%$ on 100 classes). For example, at 40% malicious clients and $q = 0.6$, FedLAW reaches $38.02\% \pm 0.12$, whereas several baselines suffer dramatic degradation and, in some cases, even collapse to accuracies close to random guessing ($\approx 1\%$ on 100 classes). For the more extreme non-IID setting $q = 0.9$, the inverse-gradient attack is significantly more destructive for all methods; FedLAW remains competitive for moderate malicious fractions (up to 20%), but performance deteriorates for very high malicious rates (e.g., 30–40%).

Under the backdoor attack, FedLAW remains accurate and stable across all configurations. For $q = 0.6$, its accuracy stays around 38–43% even at 40% malicious clients, and for $q = 0.9$ it remains best or near-best among all baselines. For the double attack, FedLAW remains stable for $q = 0.6$ (about 38–42% across all malicious fractions) and is best or near-best for $q = 0.9$ up to 30% malicious clients, but drops at 40% malicious clients.

The Lie attack is the most challenging setting for all methods: accuracies drop sharply as the malicious fraction increases, and several baselines collapse to near-random performance. FedLAW also degrades substantially under Lie at high malicious rates (e.g., from $40.94\% \pm 0.30$ at 10% to $7.51\% \pm 0.30$ at 40% for $q = 0.6$, and to $\approx 1\%$ at 40% for $q = 0.9$). Overall, these experiments indicate that FedLAW continues to provide strong robustness on a harder, high-class dataset with strong data heterogeneity and large fractions of adversarial clients.

Table 8: Test accuracy (%) on CIFAR-100 under five attack types, two non-IID levels ($q$) and varying fractions of malicious clients. Reported as mean $\pm$ std over five runs.

| Algorithm | Fraction of Malicious Clients | | | | | | | |
|---|---|---|---|---|---|---|---|---|
| | $q = 0.6$ | | | | $q = 0.9$ | | | |
| | 0.1 | 0.2 | 0.3 | 0.4 | 0.1 | 0.2 | 0.3 | 0.4 |
| **Attack: Flipping Label** | | | | | | | | |
| FedLAW | 42.27±0.27 | 40.60±0.32 | 39.42±0.25 | 37.50±0.15 | 32.12±0.70 | 30.74±0.68 | 25.25±1.22 | 21.49±0.81 |
| Bulyan | 41.32±0.09 | 38.46±0.09 | 36.35±0.22 | 34.12±0.24 | 27.35±0.04 | 14.77±0.24 | 7.52±0.65 | 5.98±0.17 |
| Bulyan-Bucketing | 40.61±0.49 | 40.31±0.28 | 37.85±0.06 | 31.83±0.08 | 24.38±0.13 | 23.15±0.25 | 21.51±0.12 | 19.06±0.59 |
| Krum | 4.73±0.95 | 1.00±0.00 | 4.50±0.11 | 1.00±0.00 | 1.00±0.00 | 1.00±0.00 | 1.00±0.00 | 1.00±0.00 |
| Trimmed Mean | 40.71±0.21 | 38.65±0.73 | 35.14±0.33 | 28.04±0.52 | 30.10±0.20 | 25.83±0.36 | 23.62±0.39 | 18.55±1.76 |
| CClip | 40.88±0.13 | 38.27±0.49 | 34.36±0.33 | 28.11±0.86 | 31.58±0.94 | 28.26±0.42 | 24.52±0.53 | 19.81±1.13 |
| CClip-Bucketing | 41.01±0.52 | 38.45±0.28 | 34.36±0.58 | 28.41±0.18 | 31.25±0.76 | 27.15±0.19 | 24.87±0.88 | 20.17±0.63 |
| RFA | 41.58±0.27 | 39.00±0.39 | 34.92±0.70 | 28.11±0.88 | 31.85±0.79 | 30.07±0.52 | 24.79±0.43 | 20.95±1.38 |
| RFA-Bucketing | 41.54±0.24 | 39.07±0.55 | 35.20±0.35 | 27.86±0.67 | 31.33±0.54 | 28.71±1.01 | 24.92±0.50 | 19.70±1.16 |
| CwMed | 40.93±0.25 | 38.26±0.20 | 34.40±0.43 | 28.47±0.36 | 25.19±0.66 | 24.60±0.55 | 21.53±0.72 | 18.22±0.38 |
| Huber Aggregator | 40.95±0.24 | 39.45±0.24 | 35.63±0.10 | 27.89±0.57 | 31.09±0.25 | 27.81±0.43 | 23.32±0.66 | 18.20±0.35 |
| FedAVG | 40.92±0.27 | 38.25±0.12 | 33.91±0.34 | 26.90±0.30 | 31.12±0.18 | 27.20±0.55 | 23.75±0.48 | 19.70±0.53 |
| **Attack: Inverse Gradient** | | | | | | | | |
| FedLAW | 42.51±0.05 | 41.39±0.49 | 40.05±0.27 | 38.02±0.12 | 30.51±0.31 | 27.48±1.41 | 21.45±1.08 | 18.42±1.89 |
| Bulyan | 36.00±0.96 | 35.19±0.51 | 34.87±0.28 | 32.04±0.25 | 7.33±0.57 | 5.49±0.73 | 3.86±0.12 | 2.61±0.17 |
| Bulyan-Bucketing | 39.28±0.21 | 38.53±0.27 | 35.06±0.41 | 25.58±0.07 | 22.77±0.32 | 20.95±0.58 | 17.66±1.06 | 12.90±0.90 |
| Krum | 4.29±1.40 | 4.90±1.90 | 3.64±0.00 | 4.21±0.36 | 1.00±0.00 | 1.00±0.00 | 1.00±0.00 | 1.00±0.00 |
| Trimmed Mean | 37.90±0.35 | 33.13±0.18 | 27.78±0.40 | 20.45±0.53 | 28.04±0.13 | 22.61±0.83 | 19.12±1.65 | 12.92±3.06 |
| CClip | 37.33±0.07 | 31.84±0.62 | 29.47±0.80 | 23.40±0.09 | 27.08±0.19 | 1.00±0.00 | 1.00±0.00 | 1.00±0.00 |
| CClip-Bucketing | 38.16±0.41 | 32.62±1.29 | 28.99±0.78 | 23.80±1.25 | 28.33±0.93 | 1.00±0.00 | 1.00±0.00 | 1.00±0.00 |
| RFA | 39.21±0.48 | 34.19±1.02 | 30.14±0.53 | 1.00±0.00 | 31.31±1.10 | 1.00±0.00 | 1.00±0.00 | 1.00±0.00 |
| RFA-Bucketing | 39.10±0.83 | 35.19±0.52 | 30.03±0.31 | 1.00±0.00 | 29.50±0.28 | 1.00±0.00 | 1.00±0.00 | 1.00±0.00 |
| CwMed | 39.07±0.50 | 33.43±0.19 | 27.40±0.34 | 21.23±0.35 | 22.60±0.52 | 19.80±0.82 | 16.70±0.88 | 11.19±1.59 |
| Huber Aggregator | 39.25±0.69 | 34.16±0.20 | 29.86±0.18 | 1.00±0.00 | 29.32±0.09 | 26.92±0.00 | 1.00±0.00 | 1.00±0.00 |
| FedAVG | 37.56±0.25 | 31.46±0.43 | 1.00±0.00 | 1.00±0.00 | 1.00±0.00 | 1.00±0.00 | 1.00±0.00 | 1.00±0.00 |
| **Attack: Backdoor** | | | | | | | | |
| FedLAW | 43.43±0.64 | 41.95±0.41 | 40.02±0.11 | 38.22±0.53 | 31.92±0.12 | 30.83±0.36 | 28.51±0.61 | 26.37±0.68 |
| Bulyan | 39.61±0.15 | 31.59±0.24 | 7.63±0.26 | 1.82±0.29 | 25.57±0.80 | 1.29±0.08 | 1.00±0.00 | 1.00±0.00 |
| Bulyan-Bucketing | 39.08±0.31 | 36.74±0.26 | 33.21±0.55 | 27.77±0.87 | 22.27±0.67 | 12.17±1.63 | 4.52±0.46 | 1.44±0.10 |
| Krum | 1.00±0.00 | 1.00±0.00 | 1.00±0.00 | 1.00±0.00 | 1.00±0.00 | 1.00±0.00 | 1.00±0.00 | 1.00±0.00 |
| Trimmed Mean | 42.32±0.10 | 40.87±0.10 | 37.14±0.62 | 30.07±2.64 | 30.20±0.30 | 25.23±0.61 | 12.31±2.13 | 2.14±0.08 |
| CClip | 41.88±0.41 | 40.21±0.54 | 37.38±1.55 | 32.73±3.21 | 30.99±0.46 | 28.58±0.87 | 25.58±2.64 | 21.30±1.65 |
| CClip-Bucketing | 41.85±0.20 | 40.63±0.33 | 37.44±1.00 | 34.15±2.59 | 31.48±0.98 | 29.23±1.20 | 26.76±1.32 | 21.73±2.15 |
| RFA | 42.18±0.24 | 40.87±0.27 | 39.65±0.39 | 37.50±0.31 | 32.72±0.29 | 31.09±0.33 | 28.33±0.54 | 25.00±1.08 |
| RFA-Bucketing | 42.46±0.75 | 41.23±0.25 | 39.73±0.65 | 37.49±0.27 | 31.75±0.28 | 29.24±0.54 | 28.12±0.59 | 25.47±0.89 |
| | | | | | | | | Continued on next page |

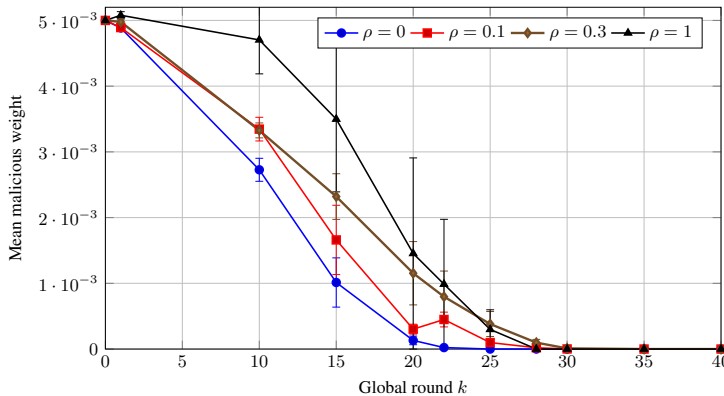

Figure 9: Evolution of the average malicious aggregation weight under the mimic–misaligned attack on MNIST with a three-layer network, 200 clients, $q = 0.9$, and $40\%$ malicious clients, for different values of the loss under-reporting parameter $\rho$. Curves show the mean over five runs, with error bars indicating $\pm$ one standard deviation. Increasing $\rho$ raises malicious weights slightly in the early rounds but only mildly delays their collapse to zero.

**Table 8 (continued) from previous page**

| Algorithm | Fraction of Malicious Clients | | | | | | | |
|---|---|---|---|---|---|---|---|---|
| | $q = 0.6$ | | | | $q = 0.9$ | | | |
| | 0.1 | 0.2 | 0.3 | 0.4 | 0.1 | 0.2 | 0.3 | 0.4 |
| CwMed | 41.28±0.55 | 39.25±0.16 | 32.76±1.20 | 28.47±0.24 | 21.12±1.14 | 12.85±1.55 | 3.50±2.24 | 2.14±1.05 |
| Huber Aggregator | 42.45±0.36 | 40.29±0.08 | 36.52±1.21 | 31.16±2.91 | 31.32±0.12 | 28.20±0.85 | 25.09±0.95 | 20.60±0.56 |
| FedAVG | 41.39±0.06 | 39.14±0.30 | 37.66±0.32 | 35.92±0.88 | 30.91±0.30 | 28.80±0.37 | 26.78±1.02 | 24.35±0.91 |
| **Attack: Double Attack** | | | | | | | | |
| FedLAW | 42.13±0.23 | 41.28±0.15 | 39.98±0.19 | 38.02±0.16 | 35.44±0.23 | 31.24±1.26 | 29.30±1.16 | 20.39±0.74 |
| Bulyan | 39.92±0.16 | 35.01±0.85 | 29.28±0.13 | 24.79±0.51 | 26.66±0.24 | 15.78±0.50 | 7.62±0.54 | 8.63±1.55 |
| Bulyan-Bucketing | 39.79±0.14 | 37.85±0.10 | 34.20±0.35 | 29.89±0.46 | 25.32±0.94 | 23.63±0.10 | 23.15±0.11 | 20.86±0.37 |
| Krum | 5.03±0.07 | 3.89±1.05 | 3.40±0.80 | 5.22±0.16 | 1.00±0.00 | 1.00±0.00 | 1.00±0.00 | 1.00±0.00 |
| Trimmed Mean | 39.74±0.42 | 36.55±0.23 | 33.23±0.39 | 29.17±0.78 | 29.18±0.20 | 25.91±1.21 | 21.63±0.44 | 18.29±1.27 |
| CClip | 40.33±0.47 | 37.12±0.06 | 34.60±0.36 | 30.74±0.19 | 30.46±0.35 | 27.31±0.33 | 24.19±0.46 | 1.00±0.00 |
| CClip-Bucketing | 40.90±0.21 | 37.86±0.19 | 33.95±0.33 | 30.25±0.27 | 30.87±0.11 | 27.33±0.37 | 23.71±0.41 | 1.00±0.00 |
| RFA | 40.79±0.71 | 33.73±0.88 | 1.00±0.00 | 1.00±0.00 | 35.61±0.93 | 31.66±0.76 | 29.11±0.45 | 1.00±0.00 |
| RFA-Bucketing | 39.01±0.52 | 33.61±0.47 | 1.00±0.00 | 1.00±0.00 | 35.42±0.30 | 31.23±1.18 | 27.46±0.60 | 1.00±0.00 |
| CwMed | 40.46±0.75 | 37.53±0.76 | 33.84±0.09 | 29.25±0.52 | 25.34±0.38 | 23.17±0.84 | 20.01±2.17 | 16.81±1.31 |
| Huber Aggregator | 40.23±0.32 | 37.29±0.01 | 34.21±0.10 | 29.75±0.19 | 29.94±0.34 | 26.63±0.15 | 22.93±0.38 | 1.00±0.00 |
| FedAVG | 39.43±0.35 | 34.02±0.19 | 29.07±0.69 | 23.64±0.64 | 29.14±0.94 | 25.04±0.85 | 18.95±0.53 | 1.00±0.00 |
| **Attack: Lie Attack** | | | | | | | | |
| FedLAW | 40.94±0.30 | 36.86±0.71 | 20.33±0.43 | 7.51±0.30 | 29.74±0.73 | 16.59±1.41 | 8.85±1.02 | 1.00±0.00 |
| Bulyan | 41.10±0.50 | 12.26±0.14 | 1.00±0.00 | 1.00±0.00 | 26.91±1.07 | 1.00±0.00 | 1.00±0.00 | 1.00±0.00 |
| Bulyan-Bucketing | 40.48±0.16 | 23.07±1.05 | 7.34±0.54 | 1.00±0.00 | 22.02±0.23 | 1.43±0.63 | 1.00±0.00 | 1.00±0.00 |
| Krum | 27.68±0.23 | 8.97±0.84 | 1.00±0.00 | 1.00±0.00 | 1.00±0.00 | 1.00±0.00 | 1.00±0.00 | 1.00±0.00 |
| Trimmed Mean | 42.44±0.22 | 30.50±0.32 | 8.07±0.52 | 1.00±0.00 | 30.18±0.57 | 9.63±1.14 | 1.00±0.00 | 1.00±0.00 |
| CClip | 42.35±0.37 | 37.21±7.22 | 23.07±8.44 | 9.34±0.81 | 32.16±2.08 | 23.89±7.91 | 14.42±0.59 | 1.00±0.00 |
| CClip-Bucketing | 41.84±0.72 | 37.00±6.36 | 22.59±8.34 | 5.80±4.22 | 31.90±1.49 | 23.35±8.43 | 9.50±7.36 | 1.00±0.00 |
| RFA | 41.58±0.32 | 30.81±0.74 | 16.88±0.31 | 1.00±0.00 | 29.69±0.47 | 15.70±1.05 | 1.24±0.24 | 1.00±0.00 |
| RFA-Bucketing | 42.52±0.49 | 31.32±0.39 | 17.09±0.65 | 1.01±0.02 | 30.34±0.73 | 15.96±0.13 | 1.32±0.53 | 1.00±0.00 |
| CwMed | 35.37±0.16 | 14.52±0.54 | 5.18±0.27 | 1.00±0.00 | 15.94±0.18 | 2.57±0.27 | 1.00±0.00 | 1.00±0.00 |
| Huber Aggregator | 42.33±0.44 | 28.39±0.32 | 11.96±0.38 | 1.00±0.01 | 31.60±0.84 | 17.91±3.04 | 1.03±0.02 | 1.00±0.00 |
| FedAVG | 43.08±0.28 | 40.85±0.12 | 28.25±0.58 | 9.01±0.94 | 32.46±0.33 | 29.16±0.78 | 16.06±0.64 | 1.00±0.00 |

## I.8 Empirical Study of Mimic−Misaligned Attack

We evaluate FedLAW under a model-poisoning "mimic–misaligned" attack on the MNIST dataset using a three-layer network, with 200 clients, highly non-IID data ($q = 0.9$), and a malicious fraction of $0.4$. For each malicious client $j$, we replace its update by $-g_j + \eta u_j$, where $u_j$ is a random unit vector and $\eta = 0.1$, and we under-report its loss as $\hat{f}_j = \mu_f - \rho \sigma_f$ with $\rho \in \{0, 0.1, 0.3, 1\}$, where $\mu_f$ and $\sigma_f$ denote the mean and standard deviation of the honest clients' losses. We restrict to $\rho \leq 1$, which keeps the cheated losses within one standard deviation of the honest mean, and thus in a regime where an attacker remains plausibly stealthy. Larger values (e.g., $\rho \geq 1.5$ or 2) correspond to losses

more than 1.5–2 standard deviations below the mean and can be easily filtered by a simple $z$-score or MAD-based preprocessing at the server. All other training hyperparameters follow the configuration in Table 1.

Figure 9 shows the evolution of the mean malicious aggregation weight over the global rounds. In all cases, the mean weight rapidly decays from the initial uniform value (approximately $5 \times 10^{-3}$) to below $10^{-4}$ and then remains essentially at zero. Increasing $\rho$ raises malicious weights slightly in the early rounds and only delays this collapse by a few rounds (from roughly round 22 for $\rho = 0$ to about round 28 for $\rho = 0.3$ and $\rho = 1$), but does not prevent the malicious weights from converging to zero. This is consistent with the Layer 2 analysis: mild loss under-reporting can temporarily compensate for the misalignment penalty, but as the weights adapt and reinforce misaligned clients' contribution to the aggregated gradient, their misalignment signal grows and their weights are eventually driven to zero.

