# OpenReview forum: "Byzantine-Robust Federated Learning with Learnable Aggregation Weights"
_ICLR.cc/2026/Conference — ICLR 2026 Poster_

### Official Review · Reviewer_YUPN · 2025-10-30

**Soundness:** 3
**Presentation:** 3
**Contribution:** 3
**Rating:** 6
**Confidence:** 3

**Summary:**

This paper uses an alternating minimization algorithm with strong convergence guarantees to address Byzantine attacks in FL. This method treats aggregate weights as learnable parameters and optimizes them jointly with global model parameters. Furthermore, the paper conducts experiments on various datasets and attack scenarios, and provides theoretical and convergence analysis of the method.

**Strengths:**

1. The paper is highly clear. The methods section provides sufficient proof and comprehensive theoretical analysis.
2. The appendix provides detailed information and experimental results supplementing the main text. The paper is highly comprehensive and logically organized. The experimental setting includes five different attack methods and various heterogeneous conditions.
3. The paper proposes learnable aggregation weights, providing a new approach to addressing the convergence problem of equal-weighted average damage.

**Weaknesses:**

1. Whether learnable aggregation weights increase the weight of byzantine clients and thus increase attack risk is lacking theoretical analysis and experimental proof.
2. The experimental datasets used are MNIST and CIFAR10. Both datasets contain only 10 classes, which cannot demonstrate the effectiveness of the method on more complex dataset.
3. In the main text, the method and theoretical analysis section of the paper lacks a figure explaining the pipeline, which increases the reading difficulty.

**Questions:**

1. I'd like to know the specific experimental results of the algorithm on more complex datasets, such as CIFAR100 and TinyImageNet, which contain more classes.
2. Would learnable aggregation weights exacerbate attacks by byzantine clients tricking the server into learning higher weights? I recommend that the paper provide theoretical analysis of this scenario.

---

> ### Author Response · Authors · 2025-11-21
> **Rebuttal by Authors**
>
> We thank the Reviewer for the positive evaluation and for the constructive comments, which helped us improve the manuscript. In particular, we have now
>
> * Extended the numerical study for the CIFAR-100 dataset (Appendix I.7).
>
> * Added Appendix C with a dynamics analysis of FedLAW and a tailored mimic attack with numerical evaluation (Figure 9).
>
> * Added the empirical study of the $\varepsilon_k$ term.
>
> We hope these clarifications and additional experiments help address your concerns and will be taken into account in your overall evaluation.
>
> ---
>
> **Q1)** Thank you for this suggestion. To evaluate our method on more complex datasets with many classes, we have added new experiments on CIFAR-100 (100 classes) under two non-IID levels ($q=0.6,0.9$) and varying fractions of malicious clients, considering two representative attacks: an inverse-gradient model-poisoning attack and a backdoor data-poisoning attack (see App. I.7 and Tab. 8). In this setting (larger label space, strong non-IID, and up to $40\\%$ malicious clients), absolute accuracies are lower than in benign CIFAR-10, and several robust baselines degrade to around $1\%$ test accuracy, which is close to random guessing on 100 classes. By contrast, FedLAW maintains test accuracies around $38$–$43\\%$ for $q=0.6$ and remains best or near-best across all baselines, indicating that the method continues to be effective on a harder, high-class benchmark.
>
> **Q2)**
> We thank the reviewer for raising this important question.
>
> Formally, Theorem 2 provides a Byzantine-resilience guarantee for our aggregator $\tilde{F}$: for {any} subset of Byzantine clients with $|\mathcal{H}^\complement|\le b_f$ (and arbitrary behavior), the mean of aggregated gradient remains close to the honest mean, and the resulting bias is explicitly bounded in Eq. (13). This implies Byzantine gradients cannot move the aggregate direction arbitrarily far from the honest mean.
>
> To address your comment more directly, we have added a new section, “Theoretical Analysis: A Multi-Layered Defense Against Strategic Attacks” (App. C). Building on Theorem 2, we formalize three structural safeguards consistent with Assumption 1
> (i) loss preprocessing to remove naive “small-loss” cheaters;
> (ii) As also mentioned in Assumption 1 in the paper, $\ell_2$ norm of Byzantine gradients is bounded by $\ell_2$ norm of honest gradients.
> (iii) weight capping $\mathbf{w}\in\Delta^+_{t,\ell_0}$ with $w_i \le t = 1/(n-b_f)$, which strictly bounds the total mass any subset of $b_f$ clients can receive.
>
> On top of these constraints, we analyze the core score
>
> $$
> s_{ji,k+1} =
> \alpha \langle\mathbf{b}_{k+1,j}- \tilde{\mathbf{v}}\_{k+1,i} , \tilde{F} \rangle + f_i(\tilde{{\mathbf{\theta}}}\_{k+2}) - \hat{f}\_{j} (\tilde{{\mathbf{\theta}}}\_{k+2})
> $$
>
> which compares a Byzantine client $j$ against an honest client $i$. The most challenging scenario is a  {mimic attack}, where the adversary tries to cancel the negative alignment term by slightly under-reporting its loss. Our new analysis shows that:
> (i) if the attacker lies too much on the loss, it is removed by Layer 1 loss preprocessing; and
> (ii) if it lies only slightly and misalignment byzantine gradient with honest, then by Theorem 2 the aggregated gradient $\tilde{F}$ remains aligned with the honest mean, so an effective attack satisfies
>
> $$
> \langle\mathbf{b}_{k+1,j}- \tilde{\mathbf{v}}\_{k+1,i} , \tilde{F} \rangle  < 0
> $$
>
> and the score $s_{ji,k+1}$ becomes negative over iterations, causing $w_{k+1,j}$ to decrease rather than grow. An adversary may try to weaken our defense by sending Byzantine gradients aligned with the average honest gradient, making the above inequality nearly vanish (very close to zero). However, since $n \gg b_f$, each client’s weight is capped by $t = 1/(n - b_f)$, which prevents any single client from moving the model far, and these updates closely follow the honest direction (as in LIE attack), so they cannot substantially deviate the global model.
> In other words, any attack that is strong enough to significantly affect the model produces either a loss gap or a misalignment signal that FedLAW uses to down-weight that client.
>
> We complement this analysis with a numerical experiment. In the revised manuscript, Appendix I.8 reports a complementary experiment on a model-poisoning ``mimic--misaligned'' attack under highly non-IID MNIST, where we track the average aggregation weight of malicious clients over rounds. Across all tested stealth levels, this average weight quickly decays from its initial uniform value to zero (see Appendix I.8 for details).
>
> **W3) Adding a pipeline figure:** Thanks for your helpful suggestion. In the final version, we will add figures in the paper illustrating the FedLAW pipeline and its theoretical components to improve readability. We have also added an empirical study in Appendix I.6, measuring per-client losses under high heterogeneity to evaluate the loss heterogeneity term in Theorem 2.

---

### Official Review · Reviewer_LuYw · 2025-10-31

**Soundness:** 2
**Presentation:** 2
**Contribution:** 3
**Rating:** 2
**Confidence:** 3

**Summary:**

This paper proposes a Byzantine robust federated learning algorithm where aggregation weights are learnt jointly with the model parameters via a nested optimization formulation. A Byzantine resilience analysis is provided, along with a convergence analysis. Experiments on MNIST and CIFAR-10 datasets show that the method outperforms baselines robust aggregations under varying heterogeneity scenarios, and for different Byzantine attacks.

**Strengths:**

- The method seems novel and provides a new avenue for research on Byzantine robustness.
- The algorithm intuition is clearly described, and many details are given on how to implement the algorithm in practice.
- The added computation and communication complexities are discussed in details, which is appreciated.
- The theoretical analysis is provided for the both the cases when only the sent gradients are corrupted and when also the sent loss evaluations are corrupted (even though only the latter truly matters).

**Weaknesses:**

1. It seems to me there is something conceptually wrong with the proof, precisely when decomposing the error in part E3. The bound between $F$ and $\tilde{F}$ assumes that the same byzantine clients will be selected by the aggregation for either the mini-batch or full gradients. This is wrong.
2. It also seems that the proof uses the exact minimum of equation (6), whereas the algorithm provides an approximation through the first order decomposition.
3. The theoretical results are not compared with baseline methods. Previous works such as [1] show robust methods that achieve the lower bounds, it is not clear how the presented method improves on those.
4. I do not agree with Remark 1 of the authors. Comparing the performance based on the total number of communication rounds is not fair.
5. Assumption (C) on Stochastic Gradient Model is not standard. Why isn’t the standard upper bound on the variance of the SGD noise (which seems to be used as inter-client variance in D1) enough for the guarantees ?
6. The Byzantine resilience guarantees are given probabilistically, which is highly non-standard in the literature. Compared to state-of-the-art Byzantine robust methods, no fundamental randomization is added by the FedLAW algorithm that would justify probabilistic bounds.
7. 321  “in practice $\epsilon_k$ is typically very small even under a high heterogeneity” the authors do not provide any justification for this statement.
8. 320 “Similarly, assuming bounded heterogeneity is a standard prerequisite for any non-IID analysis.” Can the authors please provide references for this ? The Magnitude Heterogeneity part does not look standard.
9. In Table 7, the accuracy scores on CIFAR-10 seem abnormally high (reaching even 100% with backdoor attacks).

**Minor issues**

- Theorem 2 does not specify the choice of s and t (which are specified only in the proof in the appendix)
- Shouldn’t the sparse unit capped operator be noted $\Delta_{t,s}$ instead ?

**Questions:**

1. Is there any reason why the probabilistic framework is necessary for the resilience results ?
2. It is not clear to me what the $g_i$s represent. The expectation on $v_{i,t}$ is taken with respect to what ?
3. In 170, the authors claim that the method conceptually favors clients whose gradients align with the descent direction of $f_i(\theta_k - \alpha  G_k w)$. I believe this intuition needs to be explained further, as it is not clear why this should be the case ? Why does the descent direction of one single client matter ?
4. In the experiments, why is the performance in the case of no defense almost the same as the other robust aggregation rules for many attacks and heterogeneity levels (and sometimes it is even better than some defenses)?
5. As multiple local steps are shown to improve the performance in FL (including Byzantine robustness), can the method be extended to support multiple local steps ?

**Minor questions**
- Isn’t it possible to link $L_w$, the smoothness coefficient of $\Phi_k$ to $L_{max}$ ?
- What is the point of Theorem 7 ?
- 270 $G_k^i$ is not defined. $v_{k,i}$ is defined in 253 as the full batch gradient

---

> ### Author Response · Authors · 2025-11-21
> **Rebuttal by Authors (1/3)**
>
> Thank you for your thorough and constructive feedback. We have carefully addressed your comments and substantially improved the paper's theoretical foundation. Specifically, we now:
>
> * Extended the proof in Section F3 of the paper (E3 in the previous version) for any batch size
>
> * Clarified the need for the probabilistic framework and the stochastic gradient model.
>
> * Added an empirical study of the $\varepsilon_k$ term and Remark 2 in the paper for probabilistic byzantine analysis.
>
> Before addressing each point, we highlight two key themes in your comments: the probabilistic nature of our guarantees and the scope of our threat model. We hope that our clarifications and revisions will encourage the reviewer to increase the score.
>
> ---
>
> **(i) On the Probabilistic Framework:**
>
>  Our goal is to analyze Byzantine resilience in practical FL with client-side mini-batch SGD, where inherent stochasticity makes purely deterministic analyses inadequate. In standard Byzantine analyses, terms such as $\\|\mathbb{E}\\{\tilde{\mathbf{F}}\\}-\mathbf{g}\\|_2$ are deterministic under full-batch gradients, since the expectation is taken over the full data distribution. In practice, mini-batch sampling introduces an additional source of randomness: taking the expectation over both data and sampling would again yield deterministic quantities, but at the cost of ignoring the variability observed during training.
>
> Instead, we keep the expectation over the full-batch data distribution and explicitly model mini-batch sampling as a random effect, which is crucial in adversarial settings where strategic attackers react to specific mini-batch realizations rather than expected gradients (see Appendix C, Eqs. (31)–(33), where the attack success explicitly depends on the honest clients’ mini-batch gradients). Our bounds capture this stochastic variability through a concentration scale $\varepsilon_S$, with the deterministic regime recovered by setting $\varepsilon_S = 0$. To highlight this connection, we added Remark 2 after Theorem 2.
>
>
> **(ii) Two attack surfaces.**
>
> Because FedLAW uses both per-client losses and gradients at the server, we rigorously analyze Byzantine resilience in two scenarios: (a) model attacks (e.g., gradient inversion), where only gradients are replaced; and (b) data-poisoning attacks (e.g., label flipping), where both losses and gradients are corrupted. Methods that rely solely on gradients expose only a single attack surface, whereas our formulation handles both.
>
> ----
>
> **W1) Section E3 (now is F.3):** We thank the reviewer for this insightful point. In the original proof, we used identical supports because, for large batch sizes $B$, it can be shown that the support selected by the mini-batch objective coincides with that of the full-batch objective. To address the reviewer’s concern and cover arbitrary $B$, we have extended the analysis in Section F.3 to remove the use of support equivalence between the mini-batch and full-batch optimizations due to large batch size.
>
> **Technical approach:** We clarify that in the extended version, we redefine the theoretical full-batch aggregator $F$ (line 1329) by replacing only the mini-batch gradients in $\tilde F$ with their full-batch counterparts, while keeping the same weights as in $\tilde F$. This way, $F$ and $\tilde F$ share the same support, and $F$ is used purely as an analytical tool for error decomposition. Next, we show that
>
> $$
> \mathbb{E}_{\text{batch}}\\{\\|\tilde{\mathbf{v}}_i - {\mathbf{b}}_j\\|_2^2\\}
> \geq
> \{\\|{\mathbf{v}}_i - {\mathbf{b}}_j\\|_2^2\}
> $$
>
> $$
> \mathbb{E}_{\text{batch}}\\{\\|\tilde{\mathbf{v}}_i - \tilde{\mathbf{v}}_j\\|_2^2\\}
> \leq
> \\|\mathbf{v}_i - \mathbf{v}_j\\|_2^2 + 2\varepsilon_S^2.
> $$
>
> $$
> \mathbb{E}_{\text{batch}}\\{\\|\tilde{\mathbf{v}}_i\\|_2^2\\}
> \geq
> \\|\mathbf{v}_i\\|_2^2
> $$
>
> **Result:** In Section F.3, the proof is unchanged apart from changes to obtain the above bounds and improve clarity; these are [1302-1309] and [1478-1565] in the revised manuscript. The bound in Theorem 2 remains identical, with only a minor adjustment to $C_{\text{het}}$ in equation (13): the term $\frac{2\varepsilon^2_S b_f}{n-b_f}$ is replaced by $\frac{2\varepsilon^2_S n}{n-b_f}$. This conclusion holds **for any choice of clients** selected by the mini-batch optimization and does not rely on a large batch size.
>
> Additionally, to improve the readability of the proof in Section F, we have added a brief roadmap [1229-1235] at the beginning of the section that outlines the main steps of the argument.

---

> ### Author Response · Authors · 2025-11-21
> **Rebuttal by Authors (2/3)**
>
> **W2) On the use of the exact minimum of (6):**
> The Byzantine resilience of our algorithm rests on two results: Theorem 3 shows that, under the step-size schedule in Assumption 1, the weights in Algorithm 2 converge to a critical point of (6), and Theorem 2 proves that optimal solution of (6) is Byzantine-resilient. Hence, FedLAW’s learned weights converge to a solution whose aggregated gradient is provably Byzantine-resilient. Figures 1 and 4 empirically confirm this by showing that Algorithm 2 drives attacker weights to (near) zero.
>
> **W3) Theoretical Comparison:** Thank you for the suggestion. In the final version, we will include a theoretical comparison between FedLAW and existing Byzantine-robust methods, explicitly linking their assumptions and guarantees to our convergence result in Theorem 3.
> Regarding reference [1], we could not find it in your discussion; since you refer to lower bounds, we are aware of [2], against whose bucketing method we already compare numerically in Section 5, where our algorithm performs better.
>
>
> **W4) Communication overhead:**
> Introducing $\mathbf{w}$ as a decision variable in (6) adds extra flexibility, which accelerates
> convergence of the global model parameters. Figure~8 provides numerical evidence by plotting test accuracy versus the total number of communication rounds. This observation was also emphasized by Reviewer 8DwE, who requested a standardized comparison.
>
> **W5) Assumption C:** Our analysis targets the mini-batch SGD setting used in practice. A variance-only bound on SGD noise is insufficient for our guarantees because we need high-probability, per-client control to (i) bound the sampling deviation that defines $\varepsilon_S$ and (ii) ensure stability of the support selected by (6). Assumption 1(C) (a per-sample deviation/tail bound) enables Lemma 6’s Hoeffding concentration, yielding the explicit $\varepsilon_S$ terms used in Theorems 2 and strengthening the analysis beyond in-expectation bounds.
>
>
> **W6) On probabilistic Byzantine guarantees.**
> We agree that FedLAW itself introduces no additional randomness. Our probabilistic bounds model the inherent stochasticity of practical FL usage mini-batch SGD. Deterministic guarantees in prior work often assume full-batch gradients; our analysis instead covers the realistic stochastic case and therefore states high-probability results. Probabilistic analyses are also not without precedent for robust aggregation (e.g., [3, Thm.~4] which discusses the convergence guarantee).
>
> **W7) On the size of $\varepsilon_k$ under heterogeneity.**
> Thank you for the careful reading. Our claim relies on the different sensitivities of loss and gradient heterogeneity: even when honest client gradients are highly diverse under non-IID data, their scalar losses often remain similar, so the inter-honest loss dispersion $\varepsilon_k$ stays small($\varepsilon_k$ measures loss heterogeneity among honest clients; it is typically much smaller than loss gaps involving Byzantine clients.). We added an empirical study in Appendix I.6 measuring per-client losses under high heterogeneity for MNIST and CIFAR-10, which confirms that $\varepsilon_k$ is small; the text after Theorem 2 (lines 340–341) was updated to reference this result.
>
> **W8) Gradients Heterogeneity:** Measuring and incorporating heterogeneity in the theory is standard in non-IID FL (e.g., [3] uses a metric $\Omega$ in Theorem 4). In our work, gradient heterogeneity is captured by $H_k$ and $K_k$. Because FedLAW optimizes client weights, the objective (see the first equation in §4.1) depends both on gradient differences and on their magnitudes, so we need a dispersion measure ($H_k$) and a magnitude-based measure ($K_k$). These quantities enter our bounds explicitly: when $H_k$ or $K_k$ is large, the upper bound in Theorem 2 increases accordingly.
>
> **W9) Table 7:** In Table 7, we report FedLAW’s malicious-client detection accuracy (as defined in Section I.4). Because most clients are benign, accuracy $(\text{TP} + \text{TN})/n$ is dominated by True Negatives, and FedLAW’s very low False Positive rate leads to very high accuracy.
>
> This is particularly true for data poisoning attacks (like a backdoor) on a complex dataset like CIFAR-10. As detailed in Section B.2, FedLAW's use of both loss and gradient signals provides strong evidence of these attacks, allowing it to achieve perfect or near-perfect detection (i.e., zero False Positives and zero False Negatives), resulting in the reported 100\% accuracy in some cases. Since accuracy is insufficient under class imbalance, we also report Precision, Recall, and F1: for subtler attacks (e.g., inverse gradient), accuracy stays high due to few False Positives, but Recall drops, reflecting the harder detection task.

---

> > ### Author Response · Authors · 2025-11-21
> > **Rebuttal by Authors (3/3)**
> >
> > **W10) Hyperparamerts $s$ and $t$ in Theorem2:** In Theorem 2, the number of Byzantine clients is denoted by $b_f$. According to Proposition 2 in the paper, to exactly exclude the $b_f$ clients, there is only one possible choice, namely $s = n - b_f$ and $t = \frac{1}{n - b_f}$, as mentioned following equation~(48) in the paper.
> > \vspace{5mm}
> >
> > **W11)** We utilized the notation $\Delta^{+}_{t,\ell_0}$, where $t$, $+$, and $\ell_0$ respectively denote the capped, nonnegativity, and sparsity constraints in equation~(2) of the paper.
> >
> > ---
> >
> >
> > **Q1) On the probabilistic setting.**
> > This point has been addressed above.
> >
> > **Q2) What do $g_{k,i}$ and the expectations refer to?**
> >
> > Here $\mathcal{D}_i$ is the client’s underlying data distribution.
> >
> > The
> > full-batch (empirical) gradient
> >  over client $i$’s finite dataset $D_i\subset\mathcal{D}_i$ is
> >
> > $$
> > v_{k,i}\=\\frac{1}{|D_i|}\sum_{z\in D_i}\nabla f_i(\theta_k;z).
> > $$
> >
> > Thus, the expectation involving $v_{k,i}$ is taken with respect to the randomness of the dataset draw $D_i\sim\mathcal{D}_i^{|D_i|}$:
> >
> > $$
> > \mathbb{E}\_{D_i}\\{v_{k,i}\,\\}\=g_{k,i}.
> > $$
> >
> >  We have revised Sections 2.2 and 2.1 of the paper to include this point and defining the notations $\mathbb{E}\\{.\\}$ and $\mathbb{E}_{\text{batch}}\\{.\\}$, respectively.
> >
> > **Q3) Descent Direction of a Single in (6):** We thank the reviewer for the careful reading. Our premise is not that a single client’s descent direction matters, but that Eq.(6) is designed to identify the most {internally coherent client subgroup}. Specifically, for any candidate weight vector $\mathbf{w}$, the objective first forms a collective descent direction $d(\mathbf{w}) = G_k \mathbf{w}$, and then evaluates how effective this direction is for the same subgroup via the post-update loss $\sum_i w_i f_i(\theta_k - \alpha d(\mathbf{w})).$
> > The $\arg\min_{\mathbf{w}}$ thus searches for a subgroup whose proposed direction works well for its own members. Benign clients, which tend to have mutually aligned gradients, form a coherent subgroup and achieve a low objective value, whereas including a malicious outlier distorts $d(\mathbf{w})$, making it a poor descent direction for the benign majority and thereby penalizing that choice of $\mathbf{w}$. In this way, the optimization implicitly searches for self-consistent subgroups and robustly isolates the benign cluster.
> >  In response to your comment, we have added this clarification to Appendix B.2 and now explicitly refer readers in the main text to Section~B.2 for further details.
> >
> > **Q4) Experiment:** We thank the reviewer for this important observation. It reflects the well-known robustness–utility trade-off, which is especially severe in heterogeneous FL. In non-IID settings, benign client gradients are naturally diverse, and simple defenses (e.g., Krum, Trimmed Mean) may wrongly flag them as malicious and discard them [2,3]. These false positives can hurt performance more than a weak attack, making “No Defense” (FedAvg) look competitive (see also Fig. 5 in [4]) and motivating more advanced robust FL methods [2].
> >
> > **Q5) Multiple Local steps:** As indicated in the FedLAW pseudocode in Algorithm~1 of the paper, our method includes local training for $E$ epochs, where $E$ is provided as an input parameter to our implementation.
> >
> > **Q6) smoothness coefficient:** Thanks for your comment. We addressed this point in the paper before Section 5 and its proof is provided in Lemma 17.
> >
> > **Q7) Point of Theorem 7:** Thanks for your careful reading. The paper has been revised based on your point.
> >
> > **Q8)** Thank you for the comment. In our paper, $G_{k,i}$ denotes an arbitrary distribution for $v_{k,i}$. Since we never use $G_{k,i}$ directly—and only rely on its mean and variance—we removed this notation from Assumption 1 to simplify the presentation.
> >
> > ---
> >
> > [2] Sai Praneeth Karimireddy, Lie He, and Martin Jaggi. Byzantine-robust learning on heterogeneous datasets via bucketing. In International Conference on Learning Representations (ICLR), 2022.
> >
> > [3] Pillutla, Krishna, Sham M. Kakade, and Zaid Harchaoui. "Robust aggregation for federated learning." IEEE Transactions on Signal Processing 70 (2022): 1142-1154.
> >
> > [4] Valadi, Viktor, et al. "{FedVal}: Different good or different bad in federated learning." 32nd USENIX Security Symposium (USENIX Security 23). 2023.
> >
> > [5] M. Chen, N. Shlezinger, H.V. Poor, Y.C. Eldar, and S. Cui,  Communication-efficient federated learning, PNAS, 2021.

---

### Official Review · Reviewer_8DwE · 2025-10-31

**Soundness:** 4
**Presentation:** 3
**Contribution:** 4
**Rating:** 8
**Confidence:** 4

**Summary:**

This paper proposes FedLAW, a Byzantine-robust federated learning method that treats client aggregation weights as learnable parameters, jointly optimized with the global model. The key contributions are:

A novel optimization problem formulation incorporating adaptive aggregation weights with sparsity constraints to exclude malicious clients.

An alternating minimization algorithm with theoretical convergence guarantees under adversarial settings.

Comprehensive theoretical analyses demonstrating Byzantine resilience and convergence properties.

Extensive experiments on MNIST and CIFAR-10 under various attack scenarios and non-IID data settings, showing FedLAW outperforms state-of-the-art methods, especially under high heterogeneity and malicious client ratios.

**Strengths:**

The paper presents a genuinely novel approach to Byzantine-robust federated learning by treating aggregation weights as learnable parameters. This isn't just a minor tweak to existing methods—it represents a meaningful shift in how we approach the aggregation problem. The theoretical foundation is particularly impressive, providing not just convergence guarantees but also a detailed Byzantine resilience analysis that clearly explains why the method works. What makes the contribution stand out is how well the empirical results support the theory; the method maintains strong performance even under challenging conditions like 40% malicious clients and high data heterogeneity, which is exactly where many existing methods struggle. The writing is clear and the figures effectively illustrate the method's behavior, especially the weight evolution plots that show how it dynamically identifies and suppresses malicious clients.

**Weaknesses:**

While the method is compelling, it does come with some practical trade-offs. The two-round communication per update is a noticeable overhead, and while the authors argue that faster convergence might compensate for this, the paper doesn't provide a conclusive analysis of the total communication cost compared to alternatives. Some of the theoretical assumptions, like the bounded gradient deviation and heterogeneity bounds, feel somewhat idealistic—in real-world non-IID settings, these assumptions might not hold as neatly. The experimental validation, while thorough on standard datasets, leaves me wondering how the method would scale to more complex problems or different data domains. The hyperparameter selection also seems non-trivial, particularly the sparsity level, which appears to require some knowledge of the malicious client ratio.

**Questions:**

Communication Efficiency: Could the two-round communication be optimized? Can you provide a standardized comparison of total communication rounds versus accuracy?

Practicality of Assumptions: How realistic are assumptions C1 and D1 in real non-IID settings? Are there methods to verify or relax them?

Scalability: How does FedLAW perform with a very large number of clients (e.g., >1000)? Are there distributed optimization strategies to improve efficiency?

Hyperparameter Tuning: Does the selection of $s$ and $t$ rely on prior knowledge of the malicious client ratio? Can these parameters be adapted dynamically?

Integration with Privacy Techniques: Can FedLAW be combined with differential privacy or cryptographic methods to enhance privacy protection?

---

> ### Author Response · Authors · 2025-11-21
> **Rebuttal by Authors (1/2)**
>
> Thank you for your thorough review and positive assessment of our work. We appreciate that you found the learnable weights to be a novel and meaningful contribution, and we are especially grateful for your valuable feedback. Your questions are insightful and touch on key practical considerations for our method.
> We would like to address each of your questions below.
>
> ---
> **W1) Experiments on more complex datasets.**
>
> We thank the reviewer for highlighting this point. In the revised manuscript, we added experiments on CIFAR-100 (100 classes) with two non-IID levels ($q=0.6,0.9$), varying malicious fractions, and both model- and data-poisoning attacks (App. I.7, Tab. 8). In this harder setting, several robust baselines collapse to around $1$% test accuracy (near random), whereas FedLAW remains best or near-best. These results indicate that FedLAW scales to more complex datasets, partly addressing the reviewer’s concern about scalability.
>
> **Q1) Communication efficiency.**
>
> This is a practical and important question.
>
> - **Total cost comparison:** Our main comparison (Figure~8) already plots accuracy against total communication rounds. The $x$-axis accounts for FedLAW's two rounds per update, compared with the baselines' one. Under this metric, FedLAW achieves higher accuracy for the same total communication budget, as the learned weights accelerate convergence, thereby compensating for the extra communication rounds.
> - **Optimization:** Based on our numerical study, the aggregation weights $w$ tend to stabilize quickly in practice. For example, when attacks occur in the early stages of training, we observe that FedLAW typically requires only the initial rounds (e.g., the first 10--20\% of communication rounds) to meaningfully adjust the weights; thereafter, the weights remain essentially stable, and fixing them at that point is highly effective. As a result, FedLAW's long-term communication cost is very close to that of FedAvg, with only a small, fixed upfront overhead. This behavior is an implementation choice rather than an inherent limitation of the algorithm. In settings where continued adaptivity is preferred, FedLAW can keep updating (or periodically refreshing) $w$, trading a modest additional communication cost for more frequent reweighting.
>
> We thank the reviewer for highlighting this communication-efficiency trade-off, which we now make explicit in the revised manuscript (Appendix~I.5).
>
> **Q2) Practicality of assumptions C1 and D1.**
>
> Thank you for raising this important point. We agree that the practical status of C1 and D1 should be made more explicit.
>
> - **Assumption C1 (bounded gradient deviation):**
>     Assumption~C1 requires that single-sample gradients do not deviate arbitrarily from their population counterparts, i.e.,
>     $\\|\nabla f_i(\theta_k; z) - v_{k,i}\\| \le R_k$ for all $k,i,z$, which in particular implies a uniform bound on their norms.
>     Such bounded-gradient conditions are standard in non-asymptotic analyses of stochastic gradient methods [1,2].
>     In practice, they are routinely approximated in federated learning systems: feature and label normalization is applied, and, crucially, per-example or per-client gradient-norm clipping is used for stability, robustness, or differential privacy [3,4].
>     Thus, C1 is not purely idealized: its validity can be approximately verified during training by monitoring gradient norms, and enforced in practice via this standard use of clipping.
>
> - **Assumption D1 (heterogeneity bounds):** This assumption introduces explicit, data-dependent heterogeneity quantities (directional and magnitude heterogeneity, variance, and loss heterogeneity). These play the same conceptual role as the client heterogeneity/gradient dissimilarity terms that appear in FedAvg-style and robust FL analyses, but in a more decomposed form. Importantly, our theory does not require these quantities to be small; they enter as constants in the bounds, so larger heterogeneity yields a correspondingly larger, yet still explicit, guarantee. In practice, such heterogeneity terms can be approximated at the server side from mini-batch gradients by periodically computing cross-client gradient dispersion and empirical variance on a subset of rounds. Our experiments deliberately use extreme non-IID splits ($q = 0.9$, i.e., up to $90\%$ of a client’s data in a single label), where FedLAW remains stable and effective under multiple attacks. This empirical behavior indicates that C1 and D1 are not overly restrictive in realistic non-IID settings. We also added an empirical study in Appendix I.6 measuring per-client losses under high heterogeneity, which confirms that $\varepsilon_k$ in D1 is small and is now referenced after Theorem~2. In the final version, we will add further numerical studies to evaluate these assumptions more thoroughly.

---

> ### Author Response · Authors · 2025-11-21
> **Rebuttal by Authors (2/2)**
>
> **Q3) Scalability to many clients.**
>
> This is a great point. Our method is designed for the {cross-silo setting}, where the number of clients $n$ is in the tens to low thousands and its scalability is practical:
> - **Computation:** The server-side weight update has complexity $\mathcal{O}(n \min(s,\log n) + s^2)$, which is independent of the model dimension $d$. This cost is minor compared to the $\mathcal{O}(dn)$ cost of processing gradients.
> - **Memory:** The $\mathcal{O}(dn)$ memory requirement is comparable to many other robust aggregators (such as Krum and Bulyan) that must inspect individual client updates.
>
> Adapting our persistent weight approach to the cross-device setting (massive $n$, sparse participation) is an interesting direction for future work, and we will clarify the cross-silo scope in the paper.
>
> **Q4) Hyperparameter tuning and dependence on the malicious ratio.**
>
> This is an important point. Like most Byzantine-robust FL methods (e.g., Krum, Bulyan, bucketing), our main experiments assume a known upper bound $b_f$ to ensure a fair comparison. Under this assumption, we set $s = n - b_f$ and choose the cap $t$ according to Lemma 5 (Supplementary), with $t \ge 1/s$. In practice, we use $t = 1/(s – 10)$, tuned on a validation set (see Table 1 and Section I.2).
>
> Our framework does not fundamentally require the exact $b_f$. When $b_f$ is unknown, $s$ and $t$ can be treated as standard hyperparameters and selected via validation, and one can also adopt simple adaptive heuristics (e.g., starting from $s = n$ and gradually decreasing $s$ over epochs). In all cases, $s$ acts as an upper bound on the number of active clients: for example, setting $s = 10$ still allows the method to effectively neutralize several Byzantine clients by assigning them negligible weights through the optimization of (6).
>
> Additionally, we note that our sensitivity analysis (Appendix I.2) shows that FedLAW is highly robust to the choice of $\beta$, which reduces the overall tuning burden.
>
> We thank the reviewer for highlighting this, and we added clarification in Section I.2 of the revised paper.
>
> **Q5) Integration with privacy techniques.**
>
> We thank the reviewer for this insightful question regarding FedLAW’s interaction with privacy mechanisms.
>
> - **Differential privacy (DP):** FedLAW operates on client updates, so it is compatible with client-side differential privacy: clients can clip and add DP noise locally before sending updates, and the server can apply FedLAW to these noised gradients. In our analysis, such noise appears as an additional stochastic perturbation and can be absorbed into the variance/deviation terms (e.g., $\sigma_k$, $R_k$). A detailed utility--privacy trade-off (e.g., explicit rates as a function of the privacy budget) is an interesting direction for future work.
> - **Cryptographic methods:** FedLAW shares a known trade-off with other robust methods (such as Krum). It is not compatible with standard secure aggregation, which hides all individual updates, because FedLAW (like Krum) must inspect per-client statistics to identify and down-weight malicious clients.
>
>
> ---
>
>
> [1] F. Bach and E. Moulines.
>     ``Non-asymptotic analysis of stochastic approximation algorithms for machine learning.''
>     In Advances in Neural Information Processing Systems (NeurIPS), vol.~24, 2011.
>
> [2] V. V. Mai and M. Johansson.
>     ``Stability and Convergence of Stochastic Gradient Clipping: Beyond Lipschitz Continuity and Smoothness.''
>     In Proceedings of the 38th International Conference on Machine Learning (ICML), PMLR 139:7325--7335, 2021.
>
> [3] M. Abadi, A. Chu, I. Goodfellow, H. B. McMahan, I. Mironov, K. Talwar, and L. Zhang.
>     ``Deep Learning with Differential Privacy.''
>     In Proceedings of the 2016 ACM SIGSAC Conference on Computer and Communications Security (CCS), pp.~308--318, 2016.
>
> [4] X. Zhang, X. Chen, M. Hong, S. Wu, and J. Yi.
>     ``Understanding Clipping for Federated Learning: Convergence and Client-Level Differential Privacy.''
>     In Proceedings of the 39th International Conference on Machine Learning (ICML), PMLR 162:26048--26067, 2022.

---

> > ### Comment · Reviewer_8DwE · 2025-11-24
> > **Thanks for your response**
> >
> > Thank you very much for your detailed response. I will keep my score. Moreover, I would suggest to compare with the following related works:
> >
> > Shiyuan Zuo et al. Federated learning resilient to byzantine attacks and data heterogeneity. IEEE Transactions on Mobile Computing, 2025.
> >
> > Puning Zhao et al. High dimensional distributed gradient descent with arbitrary number of byzantine attackers. arXiv:2307.13352.

---

> > > ### Author Response · Authors · 2025-11-24
> > > **Response to Reviewer 8DwE**
> > >
> > > Dear Reviewer 8DwE,
> > >
> > > Thank you very much for your response and for the valuable suggestions. We will carefully consider these works and incorporate comparisons in the final version of the manuscript.
> > >
> > >  Best regards,
> > >
> > >  The authors

---

### Official Review · Reviewer_oBSd · 2025-11-09

**Soundness:** 4
**Presentation:** 3
**Contribution:** 3
**Rating:** 8
**Confidence:** 4

**Summary:**

This paper studies proposes an algorithm that can learn the aggregation weights for benign clients in Byzantine attack environment. The idea is novel and is effective in enhancing accuracy after data in Byzantine attackers are ignored. The contributions include both new algorithm in robust federated learning and theoretical analyses for Byzantine resilience and algorithm convergence.

**Strengths:**

1. The paper studies Byzantine adversarial tolerance, which is an important problem in federated learning.
2. The paper is well written and easy to follow.
3. The theoretical analysis is rigorous and the experiments are extensive to support the effectiveness of the proposed algorithm.

**Weaknesses:**

1. Figure 1 is not clear, with confusing color to denote different algorithms.
2. How to enforce sparsity is not discussed, i.e., how to determine how many clients are malicious?

**Questions:**

1. Did you consider removing the malicious client identify step and automatically learning/assigning low weights to Byzantine clients?

---

> ### Author Response · Authors · 2025-11-21
> **Rebuttal by Authors**
>
> We thank the reviewer for the positive and encouraging assessment. We are grateful for your time and constructive feedback, which has helped us further improve the paper. Below, we address your comments in detail.
>
> ---
>
> **W1)** We thank the reviewer for pointing this out. We have revised Figure 1 to improve its clarity by using more distinct colors for different algorithms.
>
> **W2)** This is an important point. Most Byzantine FL methods (e.g., Krum, Bulyan, bucketing) assume a reasonable upper bound of $b_f$. We follow this convention in our experiments to ensure fair comparison. Under this assumption, we set $ s = n - b_f $, and choose the cap $t$ according to Lemma 5 (Supplementary), with $ t \geq 1/s $. For example, we use $ t = 1/(s – 10) $, tuned via a validation set (see Table 1 and Section I.2 for details).
>
> Our method can also operate when $b_f$ is unknown. A practical strategy is to initialize $ s = n $ and gradually reduce it across epochs. This soft exclusion approach avoids early over-pruning while maintaining robustness. Notably, $s$ acts as an upper bound on sparsity—e.g., setting $ s = 10 $ still allows the algorithm to exclude up to seven Byzantine clients by learning appropriate weights through cost minimization.
>
> Finally, when $b_f$ is unknown, $s$ can be treated as a tunable hyperparameter and selected via cross-validation. We thank the reviewer for highlighting this, and we added clarification in Section~I.2 of the revised paper.
>
>
> **Q1)** Thanks for your question. Yes, this point exactly captures the core contribution of our work.
>
> In FedLAW, we formulate an optimization problem in which assigning low weights to Byzantine clients is an automatic outcome of (6): clients whose updates (gradients and losses) worsen the objective value are penalized by the optimization, and thus receive very small (often zero) weights at the solution.

---

### Author Response · Authors · 2025-11-21
**General response (1/2)**

Dear Area Chair,

Since the rebuttal and discussion period was cut short, we wanted to briefly summarize the main contributions of our paper and how the latest revision addressed the key reviewer concerns. The details are in our responses to the individual reviewers and the updated manuscript, but we hope this note helps you quickly see where the paper stands.

First of all, we are grateful for the insightful, constructive, and overall positive feedback from the reviewers. We are pleased that they recognize the genuine novelty of jointly learning aggregation weights and the global model. In particular, LuYw notes that the method *provides a new avenue for research on Byzantine robustness,* and 8DwE highlights it as a *meaningful shift in how robust aggregation is approached*. We appreciate that reviewers oBSd, 8DwE, and YUPN acknowledge the rigorous theoretical foundations, and we value their constructive comments on the theoretical analysis. We are also happy that the overall clear presentation and comprehensive experimental study are well received, and thank the suggestions for further refinement.

In response to the questions and concerns raised, we have substantially clarified and strengthened the paper. Most notably, we have:

**1) In response to the second question of the Reviewer YUPN, we have added Appendix C with a formal analysis of FedLAW's weight updates under malicious clients, where we also considered a tailored mimic attack. In addition, we included in Appendix I.8 a numerical evaluation (Fig. 9) of the weight trajectories under a mimic attack:**


We use three constraints that limit Byzantine influence  a priori:
(i) loss preprocessing to remove naive “small-loss” cheaters;  (ii) gradient-norm clipping, which enforces Assumption 1 in the paper, namely that the $\ell_2$ norm of byzantine gradients is bounded by the $\ell_2$ norm of honest gradients.  (iii) weight capping $\mathbf{w}\in\Delta^+_{t,\ell_0}$ with $w_i \le t = 1/(n-b_f)$, which strictly bounds the total mass any subset of $b_f$ clients can receive. These are consistent with Assumption 1 and used in Theorem 2.

Under these constraints, we formally analyze how the malicious clients may choose their strategy to trick the server into assigning them higher weights.
The analysis is based on investigating a *core score* that compares how the weight update of a Byzantine client behaves relative to that of an honest client.

Our analysis shows that, if the adversary under-reports its loss too aggressively, it is detected and removed by the first-layer loss preprocessing.

Second, if it only under-reports slightly but still keeps its gradient direction misaligned with that of the honest clients, then Theorem 2 guarantees that the aggregated gradient remains aligned with the honest mean.  Therefore, in an effective attack the Byzantine gradient will be misaligned with the aggregated gradient, resulting in a negative core score for the Byzantine client over the iterations, causing its aggregation weight to decrease.

Third, if the adversary instead aligns its gradients with the honest mean to avoid this signal, its effect is further limited by the cap $t = 1/(n-b_f)$, and its updates closely follow the honest direction, so they cannot substantially deviate the global model. So, in a *mimic* attack, the adversary slightly under-reports its loss in order to compensate for the effect of its misalignment with the aggregated update compared to honest gradients.

To summarize, any attack that is strong enough to significantly affect
the model produces either a loss gap or a misalignment signal that FedLAW uses to down-weight those clients.

Finally, Appendix I.8 provides a complementary experiment under a model-poisoning *mimic-misaligned* attack on highly non-IID MNIST, where the average aggregation weight of malicious clients quickly decays from its initial uniform value to zero across all tested stealth levels.

**2) Added Remark 2 after Theorem 2, to clarify the need for the probabilistic framework and the stochastic gradient model, which Reviewer LuYw asked about:**


Our analysis targets practical FL with client-side mini-batch SGD, where the extra randomness from mini-batch random sampling makes a purely deterministic treatment inadequate. We therefore keep expectations over the full data distribution but model mini-batch sampling explicitly as a random effect. This is crucial in adversarial settings where strategic attackers react to specific mini-batch realizations rather than expected gradients (see Appendix C, Eqs. (31)–(33), where the attack success explicitly depends on the honest clients’ mini-batch gradients). Our bounds capture this stochastic variability through a concentration scale $\varepsilon_S$, with the deterministic regime recovered by setting $\varepsilon_S = 0$. To highlight this connection, we added Remark 2 after Theorem 2.

---

> ### Author Response · Authors · 2025-12-02
> **General response (2/2)**
>
> **3) Extended the proof in Section F3 of the paper for any batch size that Reviewer LuYw suggested:**
>
>
>  In the original proof, we used identical supports because, for large batch sizes $B$, it can be shown that the support selected by the mini-batch objective coincides with that of the full-batch objective. To address the reviewer’s concern and cover arbitrary $B$, we have extended the analysis in Section F.3 to remove the use of support equivalence between the mini-batch and full-batch optimizations due to large batch size.
>
> **Technical approach:** We clarify that in the extended version, we redefine the theoretical full-batch aggregator $F$ (line 1329) by replacing only the mini-batch gradients in $\tilde F$ with their full-batch counterparts, while keeping the same weights as in $\tilde F$. This way, $F$ and $\tilde F$ share the same support, and $F$ is used purely as an analytical tool for error decomposition.
>
> **Result**: In Section F.3, the proof is unchanged apart from changes to obtain the above bounds and improve clarity; these are [1302-1309] and [1478-1565] in the revised manuscript. The bound in Theorem 2 remains identical, with only a minor adjustment to $C_{\text{het}}$ in equation (13): the term $\frac{2\varepsilon^2_S b_f}{n-b_f}$ is replaced by $\frac{2\varepsilon^2_S n}{n-b_f}$. This conclusion holds \textbf{for any choice of clients} selected by the mini-batch optimization and does not rely on a large batch size.
>
>
>  **4) Added a roadmap (lines 1228-1235) and further guidance through our main theoretical proofs in F3.**
>
>
>  **5) Added a numerical study for the CIFAR-100 dataset (Appendix I.7)**.
>
>
> To address Reviewers YUPN and 8DwE’s comment for validation on a more complex dataset, we added CIFAR-100 experiments (100 classes) with two non-IID levels ($q=0.6,0.9$), varying malicious fractions, and both model- and data-poisoning attacks (App. I.7, Tab. 8). In this harder setting, several robust baselines drop to $\approx 1\\%$ test accuracy (near random), while FedLAW remains best or near-best, indicating that it scales to more complex datasets. We believe that this study at least partially addresses the scalability concern.
>
>
> **6) Added a paragraph (lines 2667–2672) clarifying that our method does not require exact knowledge of the number of Byzantine clients and explaining how to choose the sparsity level $s$ and capped $t$, addressing the comments from Reviewers oBSd and 8DwE.**
>
>
> **7) Added Appendix I.6 in the revised paper to empirically evaluate the size of $\varepsilon_k$ in Assumption 1, addressing the comment of Reviewer LuYw.**
>
>
> For reviewer comments not mentioned above, please refer to the individual rebuttals to each reviewer. In the final version of the paper, we will also (i) add a pipeline figure in the main text to illustrate the FedLAW workflow and its theoretical components, (ii) include additional simulations on CIFAR-100, and (iii) provide a more direct comparison between our theoretical analysis and that of other robust FL methods.
>
> In summary, we believe that we have addressed all questions and concerns raised by the reviewers.

---

### Meta-Review · Program_Chairs · 2026-01-07

**Summary:**

This paper proposes FedLAW, a Byzantine-robust federated learning framework that jointly learns aggregation weights and model parameters under adversarial settings. The reviewers generally agree that the idea of learning aggregation weights is novel and that the paper contains strong theoretical analysis and provides experiments on standard benchmarks. Several reviewers also appreciate the clarity of presentation and the rigor of the convergence and robustness proofs. The rebuttal and revision address a number of technical and presentation issues (e.g., proof completeness, clarification of assumptions, and additional experiments).

**Reviewer Concerns:**

Concerns adequately addressed in the rebuttal:
+ The authors clarified several theoretical points raised by reviewers.
+ Additional experiments were provided (e.g., CIFAR-100) to partially address scalability and dataset complexity concerns.
+ Presentation issues (e.g., figure clarity, proof roadmap, explanation of sparsity and hyperparameters) were improved in the revised manuscript.


Outstanding concerns that remain significant:
- Although the paper evaluates against several known Byzantine attacks, the method is primarily tested against relatively static adversaries. More state-of-the-art adaptive attacks, which explicitly react to the aggregation mechanism and learned weights, are not comprehensively evaluated. While the rebuttal introduces additional discussion and limited analysis of mimic-style attacks, this does not fully substitute for systematic validation against strong adaptive adversaries that are now standard in the literature.
- The method requires additional communication rounds, careful hyperparameter tuning (e.g., sparsity level, weight caps), and assumptions about bounded gradients and heterogeneity. While the authors argue these are reasonable, the rebuttal does not convincingly demonstrate robustness when these assumptions are violated or only approximately satisfied. In particular, cross-device federated learning scenarios with many transient clients and limited participation are not studied.
- Despite adding CIFAR-100 experiments, the empirical evaluation is still limited to image classification benchmarks under controlled conditions. It remains unclear how well the approach generalizes to diverse data modalities, more heterogeneous client behaviors, or large-scale deployments. This concern was raised by multiple reviewers and is only partially addressed by the rebuttal.


Overall, while the authors have made a genuine effort to respond constructively, the rebuttal does not fully resolve these central concerns about robustness, practicality, and generalization.

**Reviewer Scores:**

The reviewer who initially gave a low score (2) is unlikely to substantially revise their assessment. Reviewers who were initially positive may maintain similar scores.

---

### Decision · Program_Chairs · 2026-01-26

Accept (Poster)